**Technical Report**

# Novel mouse models based on intersectional genetics to identify and characterize plasmacytoid dendritic cells

**Michael Valente** [1,2] ✉, **Nils Collinet**[1], **Thien-Phong Vu Manh**[1], **Dimitri Popoff**[1], **Khalissa Rahmani**[1], **Karima Naciri**[1], **Gilles Bessou**[1], **Rejane Rua**[1], **Laurine Gil**[1], **Cyrille Mionnet**[1], **Pierre Milpied** [1], **Elena Tomasello** [1,3] ✉ **& Marc Dalod** [1,3] ✉

Plasmacytoid dendritic cells (pDCs) are the main source of type I interferon (IFN-I) during viral infections. Their other functions are debated, due to a lack of tools to identify and target them in vivo without affecting pDC-like cells and transitional DCs (tDCs), which harbor overlapping phenotypes and transcriptomes but a higher efficacy for T cell activation. In the present report, we present a reporter mouse, pDC-Tom, designed through intersectional genetics based on unique *Siglech* and *Pacsin1* coexpression in pDCs. The pDC-Tom mice specifically tagged pDCs and, on breeding with *Zbtb46*[GFP] mice, enabled transcriptomic profiling of all splenic DC types, unraveling diverging activation of pDC-like cells versus tDCs during a viral infection. The pDC-Tom mice also revealed initially similar but later divergent microanatomical relocation of splenic IFN⁺ versus IFN⁻ pDCs during infection. The mouse models and specific gene modules we report here will be useful to delineate the physiological functions of pDCs versus other DC types.

Host survival from viral infections depends on IFN-I, exerting both antiviral and immunoregulatory functions[1]. However, dysregulated IFN-I production fuels immunopathology in autoimmune diseases and certain viral infections[1,2]. Hence, identifying the cellular sources of IFN-I and their molecular regulation is important to design treatments to boost or dampen IFN-I responses depending on the pathophysiological context.

The pDCs are specialized in rapid and high-level production of IFN-I in response to viruses[1,3]. They engulf virus-derived material into endosomes equipped with toll-like receptor 9 (TLR9) for sensing unmethylated CpG DNA and TLR7 for single-stranded RNA. TLR7/9 activates an MyD88-to-IRF7 signaling cascade, leading to IFN-I production. Recently, new DC types sharing surface markers and gene expression with pDCs were identified, including pDC-like cells and transitional DCs (tDCs), which can contaminate pDC populations and confound their characterization[4–6].

Only a small fraction of pDCs produces IFN-I during viral infections[7]. How this process is regulated remains enigmatic. We do not know precisely when, where and how pDCs sense and sample virus-derived material, and how this shapes host antiviral defense[3]. Answering these questions has been hampered by the lack of mutant mouse models enabling specific and penetrant targeting of pDCs[8]. This bottleneck was caused by lack of a gene expressed in pDCs with high enough specificity to target them by classic knock-out or knock-in approaches[9]. In the *Siglech*-based deleter/reporter mice, other DC populations and macrophage subsets are targeted[10]. Moreover, activated pDCs downregulate SiglecH[7]. Hence, *Siglech*–green fluorescent protein (GFP) mice are not suitable for reliably identifying pDCs in vivo[10], even if this was attempted by selecting cells with high GFP intensity and plasmacytoid morphology[11]. Transgenic Siglech-Cre mice additionally suffer from a low pDC-targeting efficacy[12]. In *Itgax*[Cre];*Tcf4*[flox/-] mice, tDCs and pDC-like cell development is compromised[6]; macrophage and

[1]Aix-Marseille University, CNRS, INSERM, CIML, Centre d'Immunologie de Marseille-Luminy, Turing Center for Living Systems, Marseille, France. [2]Present address: Veracyte, Luminy biotech entreprises, Marseille, France. [3]These authors jointly supervised this work: Elena Tomasello, Marc Dalod. ✉e-mail: Michael.Valente@veracyte.com; tomasell@ciml.univ-mrs.fr; dalod@ciml.univ-mrs.fr

B cell subsets coexpressing *Tcf4* and *Itgax*[6,13] might also be affected. BDCA2-DTR mice[14] are the most trusted model for pDC depletion; however, they should be used with caution because their serial injection with diphtheria toxin causes artefactual chronic IFN-I production and severe immunopathology, with one dose sufficient to induce IFN-I (ref. [15]). Thus, there is an unmet scientific need for mutant mouse models allowing specific and penetrant targeting of pDCs without technical artefacts[8].

## Results

### The pDC-Tom mice allow specific pDC detection by flow cytometry

We generated mice knocked in for Cre expression from *Siglech*, which is highly expressed by mouse pDCs[3]. We crossed *Siglech*[iCre] and *Rosa26*[LoxP-STOP-LoxP(LSL)-RFP] mice[16] to generate S-RFP mice for fate-mapping *Siglech*-targeted cells (Fig. 1a). Over 95% of splenic pDCs were red fluorescent protein positive (RFP[+]) (Fig. 1b and Extended Data Fig. 1a,b). Variable proportions of myeloid and lymphoid lineages expressed RFP (Fig. 1b and Extended Data Fig. 1a,b), consistent with *Siglech* expression[10,17]. We reasoned that enhanced specificity could be achieved by harnessing intersectional genetics, driving expression of a reporter under the control of two genes coexpressed only in pDCs. We aimed for activation by *Siglech*-driven Cre of a conditional fluorescent reporter cassette knocked in a gene exclusively expressed by pDCs within *Siglech* fate-mapped cells. We selected *Pacsin1*, expressed exclusively in pDCs within hematopoietic cells[18] and promoting their IFN-I production[19]. We generated *Pacsin1*[LoxP-STOP-LoxP-tdTomato] (*Pacsin1*[LSL-tdT]) mice, knocked in with a floxed cassette for tdTomato (tdT) conditional expression. We crossed them with *Siglech*[iCre] mice, to generate pDC-Tom mice (Fig. 1c). In splenocytes from pDC-Tom mice, tdT was exclusively expressed in pDCs (Fig. 1d and Extended Data Fig. 1a,c). The tdT[+] cells expressed neither lineage markers nor CD11b (Extended Data Fig. 1d). The CD45[+] tdT[+] cells isolated from different organs were CD11c[int] and BST2[high] (Fig. 1e), as expected for pDCs[20]. CD45[+]tdT[+] cells coexpressed Ly6D, B220, SiglecH and CCR9 (Fig. 1f), a combination specific to pDCs. Thus, tdT expression in pDC-Tom mice is sufficient to specifically and unambiguously identify most pDCs.

### The pDC-Tom mice allow refining pDC gating strategies

Defining pDCs as coexpressing CD11c, BST2 and SiglecH can lead to contamination by conventional DCs (cDCs), pDC-like cells[4] or tDCs[5,6,21]. Moreover, on inflammation, such as during mouse cytomegalovirus (MCMV) infection, the expression of these markers is altered, causing a phenotypic convergence of pDCs and cDCs[7]. Hence, we harnessed pDC-Tom mice to define a gating strategy allowing unequivocal pDC identification both at steady state and during infection. We defined pDCs as tdT[+] cells and used HyperFinder for unsupervised computational generation of a gating strategy to identify them, based on surface markers without using tdT. We included Ly6D, which is selectively expressed on pDCs and B cells, discriminating them from cDCs and tDCs[4,5]. At steady state, splenic pDCs were identified as Bst2[high]Ly6D[+]B220[+]CD19[-]CCR9[+]SiglecH[+] cells (Fig. 1g). During MCMV infection, they were identified as Ly6D[+]CX₃CR1[low/int]CD19[-]CCR9[high]B220[high]BST2[high] cells (Fig. 1h). Hence, current identification of pDCs as lin[-]CD11b[-]CD11c[low-to-int]BST2[high] cells[20] can be improved by addition of positivity for Ly6D or CCR9 and eventual exclusion of CX₃CR1[high] cells. SiglecH is not a good marker postinfection (p.i.), because it is downregulated (Fig. 1i), especially on IFN-I-producing pDCs[7]. We thus propose identifying pDCs as Ly6D[high]BST2[high]CD19[-]B220[+]CD11b[-]CD11c[+] cells.

### In pDC-Tom mice, tdT soars in pDCs' immediate precursors

We analyzed bone marrow cells to investigate tdT expression along the myeloid[17,22–25] and lymphoid[4,5] ontogeny paths proposed for pDCs (Fig. 2a). Within differentiated cells, tdT was expressed exclusively in pDCs (Fig. 2b). The tdT was also detected, at a lower mean fluorescence intensity (MFI), in pDCs' immediate precursors, the CD11c[+] pre-pDCs (Fig. 2c and Extended Data Fig. 2a).

Upstream along the myeloid path, very low tdT levels were detected in SiglecH[+] pre-DCs, but none in SiglecH[-] pre-DCs, irrespective of Ly6C expression (Fig. 2d and Extended Data Fig. 2a). This is consistent with SiglecH[+]Ly6C[+/-] pre-DCs giving rise to cDCs and pDCs, whereas SiglecH[-]Ly6C[+] and SiglecH[-]Ly6C[-] pre-DCs generate mostly cDC2s or cDC1s, respectively[17]. The common DC progenitor (CDP), monocyte and DC progenitor (MDP) and common myeloid progenitor (CMP) were tdT[-] (Fig. 2e and Extended Data Fig. 2a).

Upstream along the lymphoid path, low tdT levels were detected in CD127[+]SiglecH[+]Ly6D[+] progenitors (Fig. 2g and Extended Data Fig. 2b), consistent with these cells giving rise to pDCs[4]. Very low tdT levels were detected in the Ly6D[+]SiglecH[-] progenitor and none upstream (Fig. 2f–g and Extended Data Fig. 2b).

Thus, in pDC-Tom mice, tdT expression is exclusively induced in late bone marrow precursors committed to the pDC lineage, with a strong increase in CD11c[+] pre-pDCs and maximal level in differentiated pDCs.

### ZeST mice distinguish pDCs, pDC-like cells and tDCs

Refined, flow cytometry phenotypic keys can discriminate pDCs from pDC-like cells and tDCs at steady state[6]. This remains challenging in inflammation and in tissues by immunohistofluorescence. Within hematopoietic cells, *Zbtb46* is specifically expressed in the cDC lineage including in pre-cDCs[18], in pDC-like cells, as confirmed in *Zbtb46*[GFP] mice[4,26], and in tDCs[6]. Therefore, to rigorously identify pDC-like cells and tDCs, we generated *Zbtb46*[GFP];*Siglech*[iCre];*Pacsin1*[LSL-tdT] (ZeST) mice (Fig. 3a).

In ZeST mice, most of the lin[-], SiglecH[+] or CD11c[high] cells expressed GFP or tdT in an almost mutually exclusive manner (Fig. 3b). Within tdT[+] or GFP[+] cells, we identified type 1 cDCs (cDC1s) as XCR1[+] and type 2 cDCs (cDC2s) as CD11b[+] (Fig. 3b). Within the XCR1[-]CD11b[-]BST2[high]SiglecH[+] gate, we identified pDCs as Ly6D[+]GFP[-] and pDC-like cells as GFP[+] according to their original definition[4] but considering only the major Ly6D[-] fraction. The small fraction coexpressing Ly6D and GFP (Zbtb46[+]Ly6D[+] cells) was studied separately to determine whether they encompassed pDCs. The tDCs were originally characterized as Lin[-]CD11b[-]XCR1[-]SiglecH[-/high]CD11c[low-to-high]CX3CR1[+] and split into two fractions: CD11c[low]Ly6C[+] cells, harboring higher levels of SiglecH and Tcf4, more similar to pDCs, versus CD11c[high]Ly6C[-] cells, high for Zbtb46, more similar to cDC2s[6]. Comparing phenotypic markers and genes between CD11c[low]Ly6C[+] tDCs[6] and pDC-like cells[4] suggested that they were the same population. Thus, to avoid overlap between the gates for pDC-like cells and tDCs, we focused on the SiglecH[-/low] tDCs, identified similarly to their original definition[6] as CD11b[-], XCR1[-], SiglecH[-/low], BST2[-/low]CX3CR1[+] cells, within singlet, live, nonautofluorescent, Lin[-], SiglecH[+] or CD11c[high], tdT[+] or GFP[+] cells (Fig. 3b). The SiglecH[-/low] tDCs were all Ly6D[-] (Fig. 3b–c) and expressed higher levels of CD11c than pDCs and pDC-like cells (Fig. 3c); hence we called them CD11c[high] tDCs. Only very few of the Lin[-], CD11c[high] or SiglecH[+] cells expressed neither GFP nor tdT, a fraction of which were Ly6D[+]SiglecH[high] and thus probably tdT[-] pDCs (Fig. 3b).

To validate the identity of the DC types gated manually, we characterized their phenotype further (Fig. 3c). As expected, beyond being Ly6D[+]SiglecH[+], both tdT[-] and tdT[+] pDCs were CCR9[high]CD11c[int]. A fraction of Zbtb46[+]Ly6D[+] cells expressed lower levels of tdT and CCR9, and higher levels of CD11c, compared with tdT[+] pDCs. CD11c[high] tDCs were GFP[high]tdT[-]. The pDC-like cells were GFP[+]tdT[-]CCR9[-/low]. The surface phenotype of these populations was not modified during MCMV infection (Fig. 3c and Extended Data Fig. 3). Hence, tdT was expressed only in cells harboring a phenotype of bona fide pDCs, whereas GFP was mainly expressed in cDC1s, cDC2s, pDC-like cells and CD11c[high] tDCs.

Next, we performed an unsupervised analysis of the phenotypic relationships for all the Lin[-], CD11c[high] or SiglecH[+] cells. Onto a *t*-distributed stochastic neighbor embedding (t-SNE) representation of the data, we projected the populations identified through manual

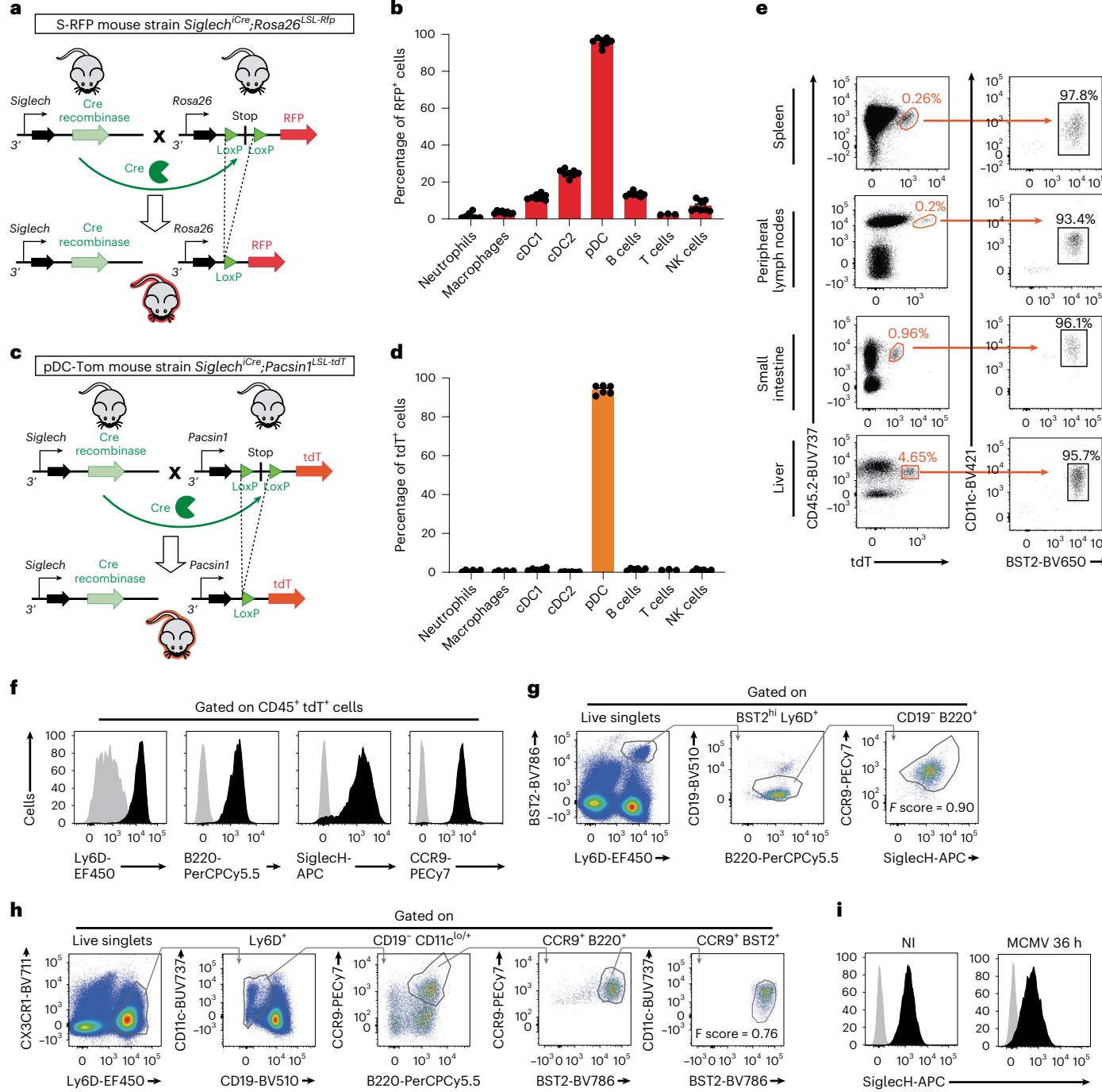

**Fig. 1 | The pDC-Tom mice allow specific and unambiguous identification of pDCs in different organs. a**, Scheme illustrating the strategy followed to generate S-RFP mice. LoxP is the sequence recognized by Cre recombinase. 'Stop' corresponds to a transcriptional stop sequence. **b**, Splenocytes isolated from S-RFP mice were stained with fluorescently labeled antibodies to identify the indicated myeloid and lymphoid cell populations and analyzed for RFP expression by flow cytometry. The data shown (mean ± s.e.m.) are pooled from two independent experiments (n = 8). **c**, Scheme illustrating the strategy followed to generate pDC-Tom mice. **d**, Splenocytes isolated from pDC-Tom mice stained as in **b** to analyze tdT expression by flow cytometry. The data shown (mean ± s.e.m.) are pooled from two independent experiments (n = 6). **e**, Single-cell suspensions of indicated organs isolated from pDC-Tom mice stained with indicated fluorescent antibodies and analyzed by flow cytometry. **f**, Splenocytes from **e** analyzed for the expression of indicated markers on CD45⁺tdT⁺ cells. Gray histograms correspond to negative controls (fluorescence − 1). Black histograms correspond to the signal obtained on staining with the indicated antibody. For **e** and **f**, the data shown are from one mouse representing seven animals for the spleen and five animals for the peripheral lymph nodes (LNs), liver and small intestine. **g**,**h**, The HyperFinder plugin of the FlowJo software was applied to define an unsupervised gating strategy to identify pDCs from uninfected (**g**) or 36-h MCMV-infected (**h**) pDC-Tom mice. **i**, SiglecH expression (black histograms) shown on the pDCs as defined by the automated gating strategies computed for uninfected animals (**g**) or 36-h MCMV-infected mice (**h**). The negative controls (fluorescence − 1) are shown as gray histograms.

gating as defined in Fig. 3b. This analysis highlighted three main cell clusters, corresponding to cDC2s, cDC1s and pDCs (Fig. 3d). The other DC types formed a continuum between pDCs and cDC2s, with tdT⁻ pDCs and Zbtb46⁺Ly6D⁺ cells close to pDCs, versus pDC-like cells and CD11c^high tDCs close to cDC2s, consistent with previous observations[6].

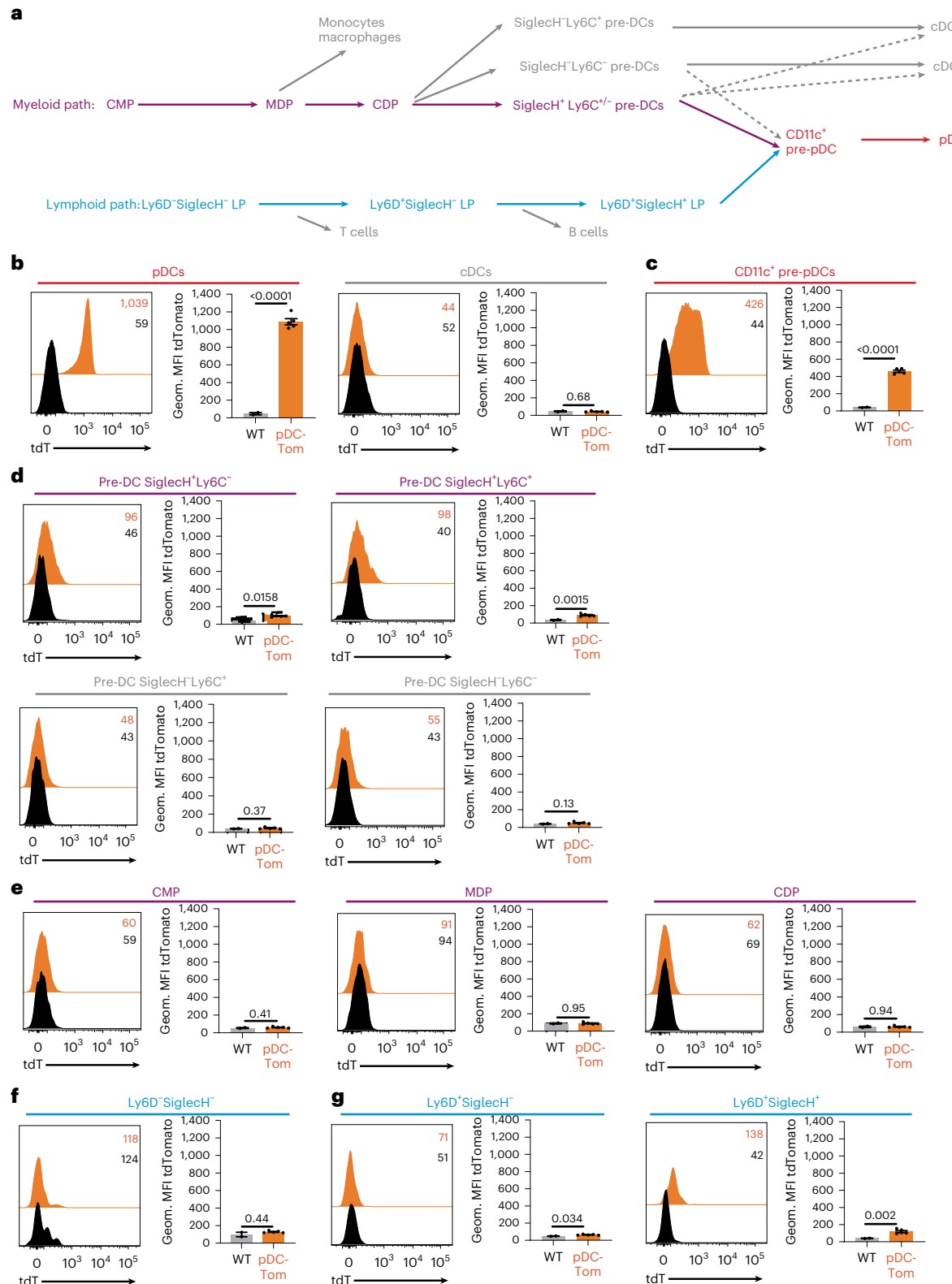

**Fig. 2 | Expression of tdT is detectable in late bone marrow precursors selectively committed to the pDC lineage. a**, Scheme of the previously proposed ontogenic paths for pDC differentiation along the myeloid (top, magenta) or lymphoid (bottom, cyan) lineages. Cells of these lineages diverging from the pDC main differentiation path are shown in gray. CD11c⁺ pre-pDCs and terminally differentiated pDCs, which are common to both paths, are shown in red. **b**–**g**, Bone marrow cells isolated from pDC-Tom animals, stained with fluorescently labeled antibodies and analyzed by flow cytometry. The expression of tdT (orange histograms) was evaluated in bone marrow pDCs and cDCs (**b**),

CD11c⁺ pre-pDCs (**c**) and different progenitors along the myeloid (**d**, pre-DC and **e**, early progenitors) or lymphoid (**f**, Ly6D⁻ progenitors, and **g**, Ly6D⁺ progenitors) ontogenic paths. C57BL/6 mice were used as negative controls (black histograms). WT, wild-type. The fluorescence histograms shown (left) are from one mouse representing five pDC-Tom animals from two independent experiments. The bar graphs (right) correspond to the results of all five animals, with data shown as mean ± s.e.m. Statistical analysis was by two-sided, unpaired Student's *t*-test: *$P < 0.05$, **$P < 0.01$, ***$P < 0.001$, ****$P < 0.0001$. Geom., Geometric.

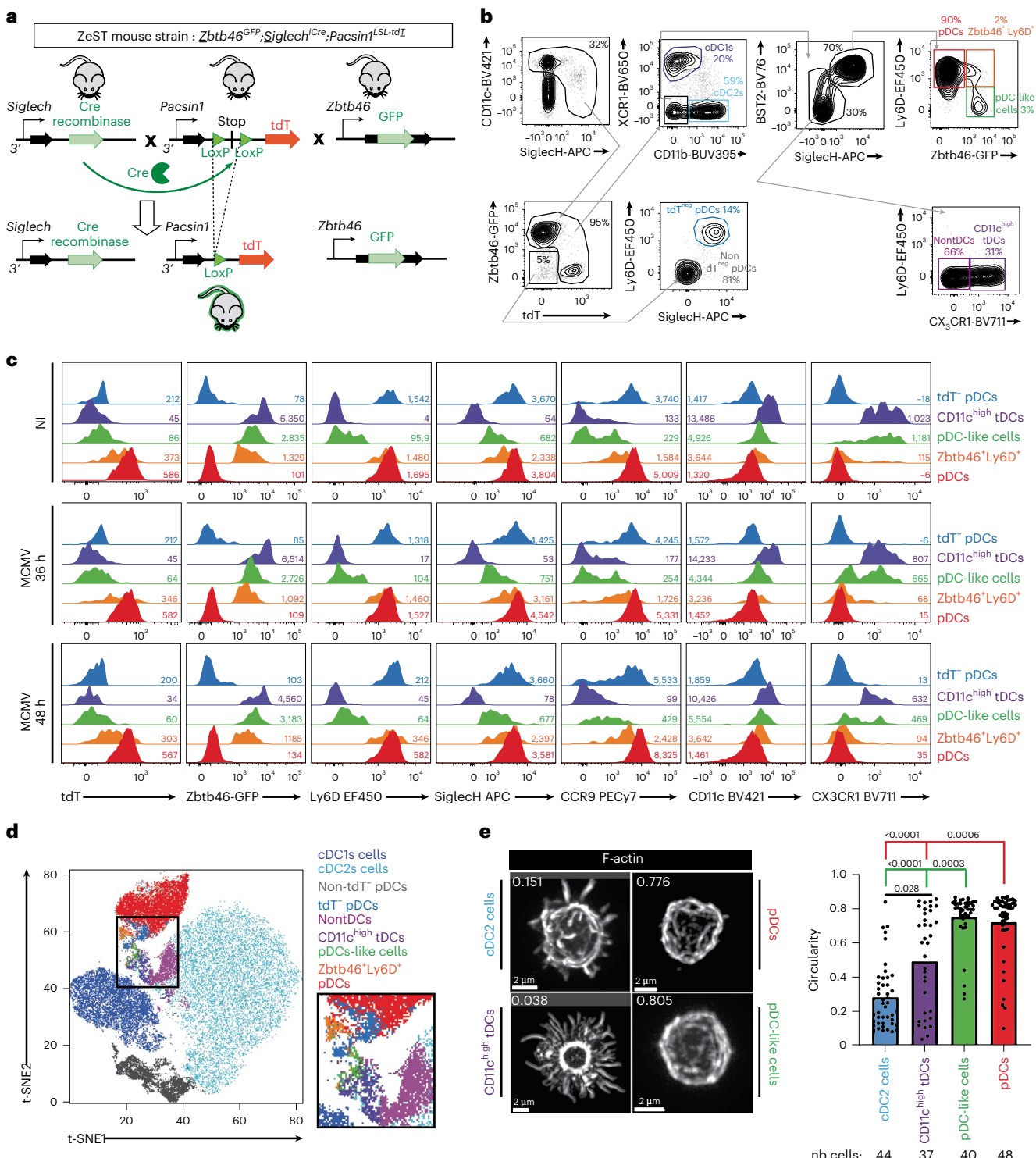

**Fig. 3 | ZeST mice allow unambiguous discrimination of pDCs from tDCs and pDC-like cells. a**, Strategy for generation of ZeST mice. **b,c**, Splenocytes from ZeST mice stained with fluorescently labeled antibodies and analyzed by flow cytometry. **b**, Gating strategy followed to identify cDC1s, cDC2s, CD11c^high tDCs, pDC-like cells, pDCs, Zbtb46+Ly6D+ cells and tdT− pDCs. The first dot plot showing CD11c versus SiglecH expression was gated on singlets, live (LiveDead−), nonautofluorescent, Lineage(CD19,CD3,Ly6G,NK1.1)− cells. **c**, Expression of indicated fluorescent proteins or cell-surface markers on each of the cell populations gated as in **b**, from splenocyte suspensions from uninfected versus 36-h or 48-h MCMV-infected pDC-Tom mice. **d**, Projection of the cell types identified in **b**, according to the color key shown on the upper right of the panel, on the t-SNE space calculated for all cells expressing high levels of CD11c or positive for SiglecH. The data shown are from one ZeST mouse representing

at least ten uninfected animals for **b**–**d** and for seven MCMV-infected animals at 36 h p.i. or eight MCMV-infected animals at 48 h p.i. for **c. e**, Quantitative and unbiased assessment of the cellular morphology of cDC2s, tDCs, pDC-like cells and pDCs sorted from the spleen of uninfected ZeST mice according to the gating strategy shown in **b**. One representative confocal microscopy image of each DC type is shown on the left. The distribution of the circularity indices for individual cells across DC types is shown as dots on the right, with the overlaid color bars showing the mean circularity indices of each DC type. The data shown are from two independent experiments, each performed with one mouse, with 37–44 individual cells analyzed for each DC type, as shown below the graph. nb cells, number of indiviudal cells analyzed. The Kruskal–Wallis test was used for the statistical analysis: *P < 0.05, **P < 0.01, ***P < 0.001, ****P < 0.0001.

We sorted cDC2s, CD11c[high] tDCs, pDC-like cells and pDCs from steady-state mouse spleens to examine their morphology (Fig. 3e). The cDC2s harbored many pseudopods or dendrites, translating into a low circularity index. Most pDC-like cells and pDCs harbored a round morphology, translating into a high circularity index. The CD11c[high] tDC population was morphologically heterogeneous, with a bimodal distribution of circularity indices, half of the cells being dendritic, like cDC2s, and the other half round, like pDCs. Overall, quantitative and unbiased analysis of cellular morphology supported success in high-degree purification of the DC types.

Finally, to better discriminate the tdT signal from autofluorescence, and analyze more cell-surface markers in ZeST mice, we harnessed spectral flow cytometry (Extended Data Figs. 4 and 5). Unsupervised cell clustering based on all surface markers (Extended Data Fig. 4a,b), without considering tdT and GFP, showed that most GFP[+] cells were cDC2s or cDC1s (Extended Data Fig. 4c). They also encompassed lymphoid cells and a cluster of myeloid cells, but with a low MFI (Extended Data Fig. 4d). The pDCs represented 81.5 ± 6% of tdT[+] splenocytes (Extended Data Fig. 4e). The individual contribution of other phenotypic cell clusters to the tdT[+] gate was very small and their tdT MFI below that of pDCs, barely above background (Extended Data Fig. 4f).

Complementary analysis by supervised identification of cell types via manual gating allowed study of both tDC populations: the Ly6C[−] versus Ly6C[+] fractions of Lin[−]CD11b[−]XCR1[−], CD11c[+] or BST2[+], CX3CR1[+]CD26[+] cells (Extended Data Fig. 5a–d). The populations phenotypically defined as pDC-like cells[4] or Ly6C[+] tDCs[6] largely overlapped, as was the case for CD11c[high] tDCs and Ly6C[−] tDCs (Extended Data Fig. 5b–e). Most Lin[−]CD11b[−]XCR1[−], CD11c[+] or BST2[+], Ly6D[−]CX3CR1[−] cells were CD11c[+]CD26[+]GFP[+]CD64[−] (Extended Data Fig. 5f), indicating DC lineage. They encompassed major histocompatibility complex II (MHC-II)[−/low] and MHC-II[high] cells putatively corresponding to pre-DCs versus differentiated DCs, respectively. A decrease in the absolute numbers of most of DC types was observed in the spleen at 48 h p.i. (Supplementary Table). This analysis confirmed the high specificity and penetrance of GFP expression in cDC1s and cDC2s, and both tDC populations, as well as the high specificity and penetrance of tdT expression in pDCs (Extended Data Fig. 5g).

## ScRNA-seq confirms DC-type identification in ZeST mice

We harnessed ZeST mice to perform single-cell RNA sequencing (scRNA-seq) for the five DC types, on index sorting from the spleen of animals either infected or not infected with MCMV, using the gating strategy shown in Fig. 3b and the FB5P-seq (FACS-based 5′-end scRNA-seq) protocol[27,28].

We first analyzed 343 splenocytes isolated from noninfected (NI) mice. They were clustered and annotated for cell types (Supplementary Table) by Seurat, based on gene expression profiles (Extended Data Fig. 6a), and by Rphenograph, based on surface marker expression (Extended Data Fig. 6b,c). DC-type assignment to individual cells was rather consistent between these two strategies, but suggested heterogeneity of Seurat clusters II and III (Extended Data Fig. 6d). Therefore, to unambiguously and reliably assign a DC type to individual cells in an unbiased manner, we used a combinatorial strategy: to be selected, a cell had to belong to the expected intersection between the Rphenograph and Seurat clusters (colored cells in Extended Data Fig. 6d) and to be enriched for the corresponding DC-type-specific transcriptomic signatures[4,29] as tested with single-cell connectivity Map (sgCMap; Extended Data Fig. 6e). A final cell type was assigned to 205 cells: 103 pDCs, 26 pDC-like cells, 19 CD11c[high] tDCs, 23 cDC2s and 34 cDC1s (Extended Data Fig. 6e,f and Supplementary Table).

Our next objective was to assign a DC type to the 951 cells from the whole dataset, from both NI and MCMV-infected mice (Extended Data Fig. 7a and Supplementary Table). We aimed at achieving a robust DC-type assignment, irrespective of cell states and infection conditions, by combining transcriptomic and phenotypic analyses. Based on the intersection of the clustering with Seurat (Extended Data Fig. 7b) versus Rphenograph (Extended Data Fig. 7c,d), a final DC type was assigned to 851 cells (colored cells in Extended Data Fig. 7e): 310 pDCs, 170 pDC-like cells, 146 CD11c[high] tDCs, 103 cDC2s and 122 cDC1s, with 100 cells left unannotated (Fig. 4a). We confirmed DC-type assignment by a complementary method: we generated cell type-specific transcriptomic signatures from the dataset focused on cells from NI mice (Supplementary Table) and used them for sgCMap analysis of the whole dataset (Extended Data Fig. 7f). All the cells sorted as pDCs, and most of the tdT[−] pDCs, were computationally assigned to pDCs (Extended Data Fig. 7e,f). The assignment to cDC1s was also consistent between cell sorting and deductive re-annotation. Cell clustering on sgCMap scores tended to confirm the distinction between cDC2s and CD11c[high] tDCs, although many cells had a null score for the 'tDC_vs_cDC2' sgCMap signature, emphasizing the proximity between these two DC types (Extended Data Fig. 7f). As expected, Zbtb46[+]Ly6D[+] cells were mostly assigned to pDC-like cells. Some cells sorted as pDC-like cells were in time assigned to CD11c[high] tDCs (Extended Data Fig. 7e,f). Not only CD11c[high] tDCs but also pDC-like cells were CX3CR1[+/high] (Fig. 4b). Akin to cDCs, CD11c[high] tDCs were GFP[high], whereas pDC-like cells expressed SiglecH, BST2, Ly6D and CCR9 to levels intermediate between those of pDCs (high) and cDCs (low). The pDCs and pDC-like cells shared high expression of *Siglech* (Fig. 4c,d), *Tcf4* and *Runx2* (Fig. 4c). CD11c[high] tDCs and pDC-like cells shared high expression of *Crip1*, *Lgals3* and *Vim* (Fig. 4c), previously reported to discriminate pDC-like cells from pDCs[4]. The pDC-like cells and CD11c[high] tDCs shared with cDCs higher expression of *Zbtb46*, *Spi1*, *Slamf7* and *S100a11*, compared with pDCs (Fig. 4c). Contrary to cDC2 cells, a fraction of pDC-like cells and CD11c[high] tDCs expressed *Cd8a* (Fig. 4c,d), as reported for tDCs[6,21,29]. The pDC-like cells, CD11c[high] tDCs and cDC2s specifically expressed *Ms4a4c* (Fig. 4c,d) and *Ms4a6c* (Fig. 4c). CD11c[high] tDCs and cDC2s selectively shared expression of *Ms4a6b* (Fig. 4d) and *S100a4* (Fig. 4c). CD11c[high] tDCs expressed higher levels of certain cDC genes than pDCs and pDC-like cells, including *Batf3*, *Rogdi* and *Cyria* (Fig. 4c). The pDC-like cells expressed very high levels of *Ly6c2* (Fig. 4c,d). Hence, the pDC-like cells characterized in the present report were confirmed to align with both the originally described pDC-like cells[4] and the CD11c[low]Ly6C[+] tDCs[6].

All cDC1s expressed the XCR1 protein (Fig. 4b). However, a fraction was low/negative for *Xcr1* and other genes specific of steady-state cDC1s (*Gpr141b*, *Tlr3*, *Cadm1* and *Naaa*; Fig. 4c), consistent with DC-type activation decreasing the expression of many of the genes used to identify them at steady state[30–32], preventing use of individual genes for reliable DC-type identification in scRNA-seq datasets[33].

As expected, only pDCs expressed rearranged immunoglobulin (Ig) genes, *Ccr9*, *Klk1* and *Cox6a2* (Fig. 4c) as well as the tdT fluorescent reporter above autofluorescence levels (Fig. 4e). The tdT[−] pDCs were largely overrepresented in our scRNA-seq dataset because we enriched them for characterization. As we did not use the tdT signal in our analysis, these results confirm the specificity of pDC-Tom mice for pDC identification; they also confirmed proper identification of pDC-like cells and CD11c[high] tDCs in ZeST mice, and allowed refining of their characterization through side-by-side pangenomic transcriptomic profiling.

## The pDC-Tom mice allow mapping of pDC microanatomical localization

We harnessed the pDC-Tom mice to determine the microanatomical localization of pDCs in various organs. In the spleen, a relatively high density of tdT[+] cells was observed in the T cell zone (TCZ) within the white pulp (WP) (Fig. 5a). Scattered tdT[+] cells were detected in the red pulp (RP), delimited by CD169[+] metalophilic marginal zone (MZ) macrophages and densely populated by F4/80[+] RP macrophages

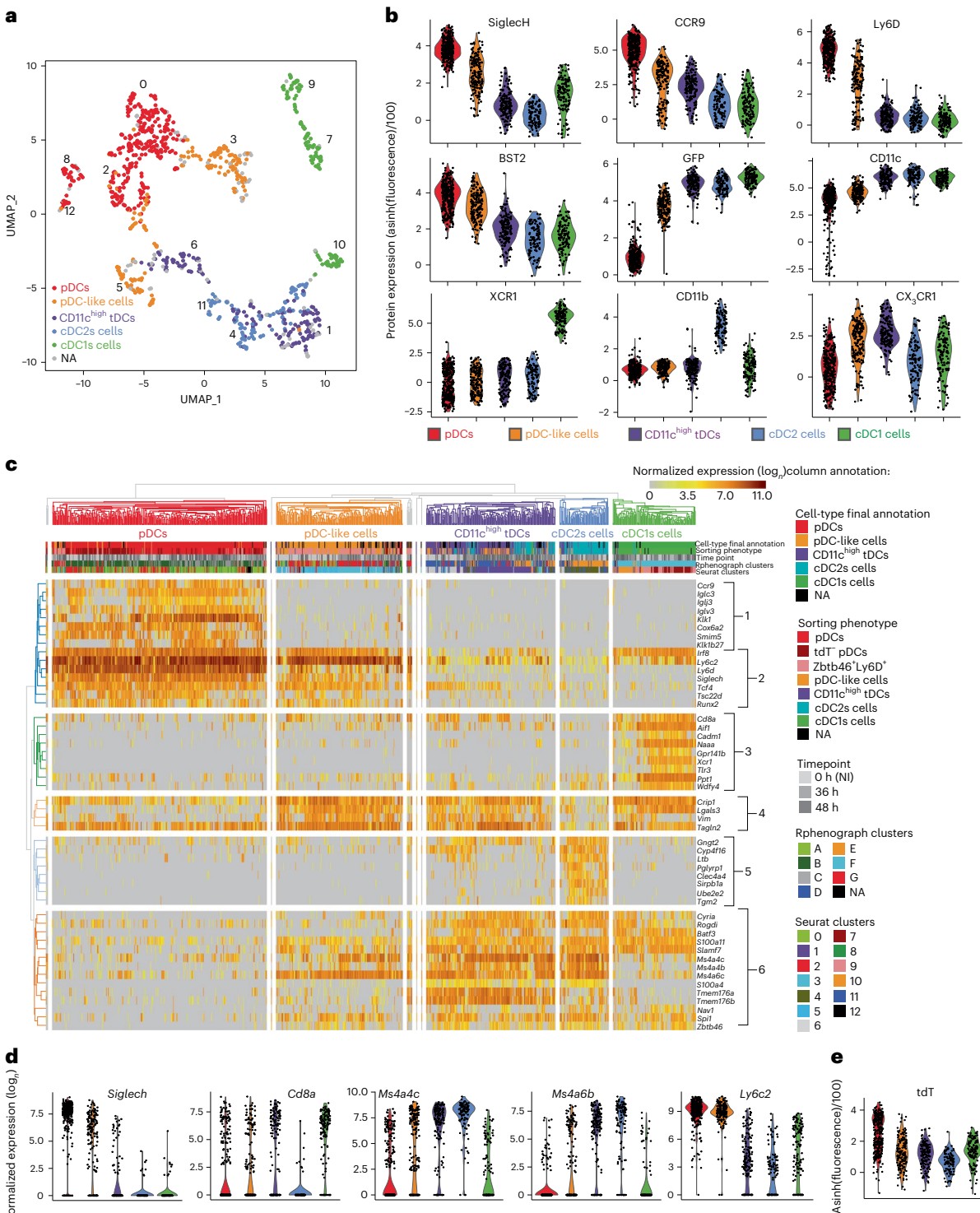

**Fig. 4 | ScRNA-seq confirmed proper identification of pDCs, pDC-like cells and tDCs in ZeST mice. a,** UMAP dimensionality reduction for DC types isolated from the spleens of eight ZeST mice (three uninfected, three MCMV infected for 36 h and two infected for 48 h; Extended Data Fig. 7a). Cells were index sorted into the five DC types studied (Fig. 3b) and used for scRNA-seq. As indicated by the color code, 851 individual cells were reassigned by deduction to a DC-type identity (cDC1s, cDC2s, pDCs, pDC-like cells or CD11c^high tDCs), based on combined analysis of their phenotypic and transcriptomic characteristics, as assessed, respectively, by Rphenograph clustering (Extended Data Fig. 7c) and Seurat clustering (numbers on the UMAP; Extended Data Fig. 7b), with confirmation via a single-cell enrichment analysis for DC-type-specific signatures generated from prior analysis of the cells from uninfected mice only (Extended Data Figs. 6 and 7); 100 cells were left nonannotated (NA). **b,** Violin

plots showing expression of phenotypic markers across DC types. **c,** Heatmap showing messenger RNA expression levels of selected genes (rows) with hierarchical clustering using Euclidean distance, across all 951 cells (columns) annotated for (1) cell type final annotation as shown in **a**, (2) sorting phenotype, (3) time point after MCMV infection, (4) belonging to Rphenograph clusters and (5) belonging to Seurat clusters. Six gene groups are shown: (1) genes specifically expressed at high levels in pDCs, (2) genes with shared selective expression in pDCs and pDC-like cells, (3) cDC1-specific genes, (4) genes previously reported to be expressed at higher levels in pDC-like cells over pDCs, (5) cDC2-specific genes and (6) genes expressed selectively at higher levels in CD11c^high tDCs and cDC2 or cDC1 or pDC-like cells. **d,** Violin plots showing mRNA expression levels of selected genes across DC types. **e,** Violin plots showing tdT expression across DC types.

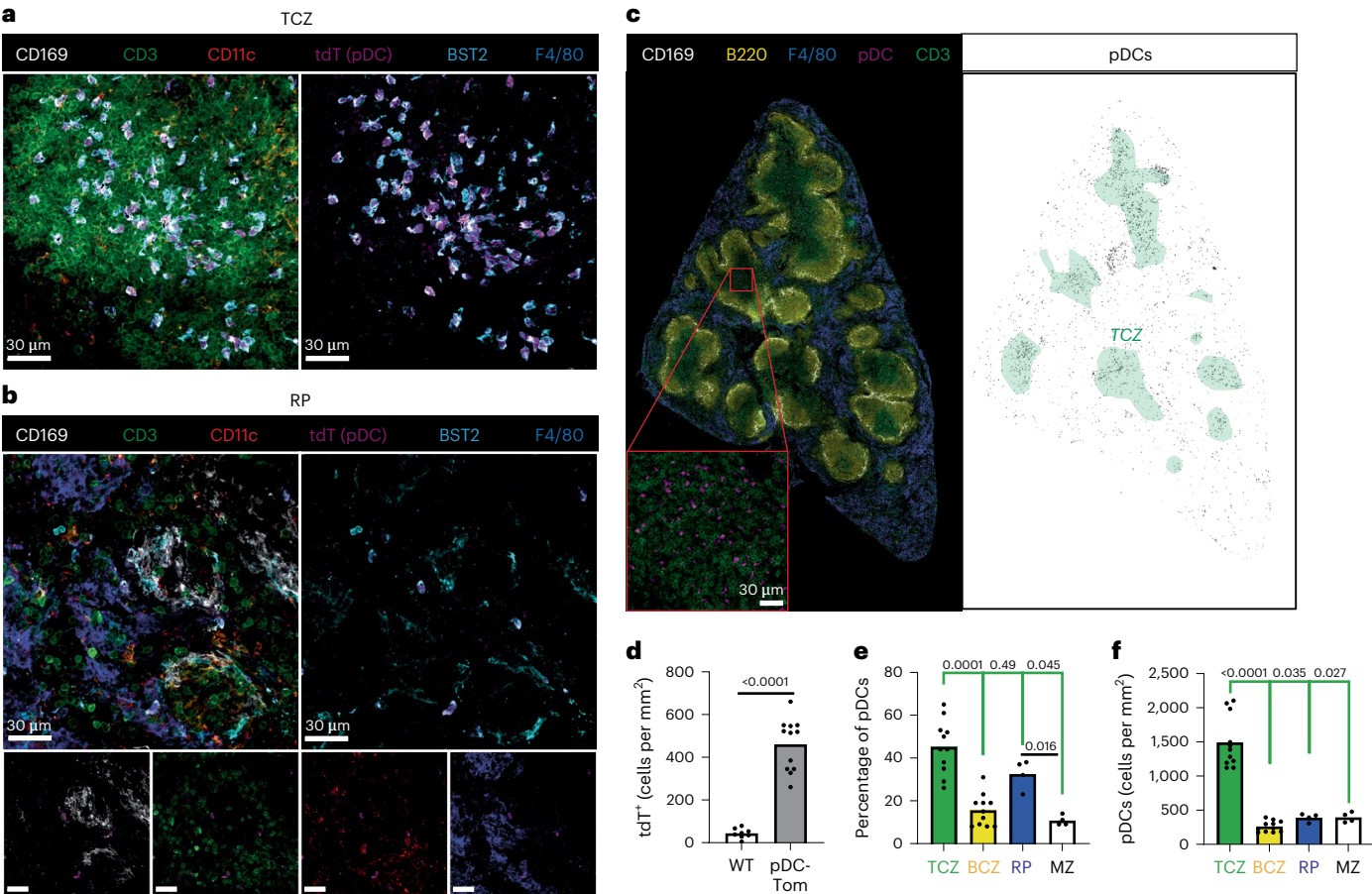

**Fig. 5 | The pDC-Tom mice allow studying pDC microanatomical location in the spleen. a–c**, Spleen cryosections, 20 μm, from steady-state pDC-Tom mice stained with anti-tdT (magenta), anti-CD169 (white), anti-CD3 (green) and anti-F4/80 (blue) antibodies, combined with anti-CD11c (red) and anti-BST-2 (cyan) antibodies (**a**, TCZ and **b**, RP) or an anti-B220 (yellow) antibody (**c**). Representative images of four sections from two animals are shown, for the TCZ (**a**), the RP (**b**) and the whole section (**c**), with the left panel corresponding to the raw signal and the right panel to the pDC mask used for quantification (black) relative to the location of the TCZ (green area, based on anti-CD3 staining). **d**, The number of tdT[+] cells mm[−2] of the whole spleen sections was quantified in pDC-Tom mice, in comparison to wild-type animals as background controls. The data shown are from eight whole sections from two C57BL/6 mice and 12 whole sections from three pDC-Tom mice. **e,f**, Microanatomical distribution of pDCs across the different areas of the spleen, namely the TCZ (CD3-rich region), BCZ (B220-rich region), RP (F4/80-rich area) and MZ (defined as the space between the CD169 staining and the F4/80 staining). **e**, Fraction of the pDC population present in each microanatomical area (calculated as the ratio between the pDC counts in the area and the total pDC counts in the whole section). **f**, The pDC density (cells mm[−2]) in each microanatomical area. The data shown are from 11 sections from three mice for TCZ and BCZ, and four sections from three mice for MZ and RP. The height of the colored boxes shows the mean value for each microanatomical area. One-way analysis of variance (ANOVA) was used for the statistical analysis, [*]$P < 0.05$, [**]$P < 0.01$, [***]$P < 0.001$, [****]$P < 0.0001$.

(Fig. 5b). Most TCZ or RP tdT[+] splenocytes expressed BST2 (Fig. 5a,b). Conversely, whereas most TCZ BST2[+] cells were tdT[+] (Fig. 5a), this was not the case in the RP (Fig. 5b), consistent with BST2 expression on plasma cells, macrophage subsets, pDC-like cells and tDCs[4,6,34]. We quantified the density of pDCs (tdT[+] cells) in the whole spleen (Fig. 5c,d) and in its microanatomical compartments: the RP, the MZ, the TCZ and the B cell zone (BCZ) (Fig. 5c,e,f). The pDC density in the whole spleen was around 400 cells mm[−2] (Fig. 5d). Steady-state splenic pDCs were primarily located in the TCZ and RP (Fig. 5e). The pDC density was much higher in the TCZ (~1,500 cells mm[−2]) than in other splenic compartments (≤400 cells mm[−2]) (Fig. 5c,f). In lymph nodes, pDCs were frequent in paracortical T cell (CD3[+]) areas (Extended Data Fig. 8a). In the small intestine, pDCs had been observed in both Peyer's patches[35] and the lamina propria (LP), primarily based on BST2 signal[36]. However, interpretation may have been confounded by BST2 expression on macrophage and DC subsets in the gut[37,38]. Hence, we re-examined pDC microanatomical location in the small intestine. The tdT[+] cells were detectable in the LP, generally close to EpCAM[+] epithelial cells

(Extended Data Fig. 8b). Very few tdT[+] cells were observed in Peyer's patches (Extended Data Fig. 8c). The pDCs were not detectable in the LP of the large intestine (Extended Data Fig. 8d). Thus, at steady state, in the gut, pDCs are detectable quasi-exclusively in the small intestine LP.

**SCRIPT mice allow tracking IFN-I[+] and IFN-I[−] pDCs in situ**

During MCMV infection, the *Ifnb1*[EYFP] reporter mouse model[39] does not only allow detection, at 36 h p.i., of the pDCs actively producing IFN-β[7,27], but, in addition, at 48 h, of the pDCs that have already produced and secreted IFN-β but maintained enhanced YFP expression for >12 h[27]. Hence, the *Ifnb1*[EYFP] mice are reliable and well suited to fate-map the pDCs that have previously produced IFN-I, enabling determination of their activation trajectory in vivo during infection[27]. The pDCs clustered in the MZ near to infected cells at 36 h p.i., at the time of peak IFN-I production, then migrated to the TCZ after termination of their IFN-I production between 40 h and 48 h p.i., while acquiring transcriptomic, phenotypic and functional features of mature cDCs[27]. However, the lack

of adequate markers prevented us from assessing the spatiotemporal repositioning of the pDCs that were not producing IFN-I. To answer this question, we generated *Siglech*[iCre];*Ifnb1*[EYFP];*Pacsin1*[LSL-tdT] (SCRIPT) mice (Fig. 6a) for unambiguous identification of both the pDCs that are producing (or have produced) IFN-I (tdT[+]YFP[+]) and those that do not (tdT[+]YFP[−]) (Fig. 6a,b). By flow cytometry, both tdT[+]YFP[−] and tdT[+]YFP[+] cells were detectable at 36 h and 48 h p.i., expressing Ly6D and BST2 consistent with their pDC identity (Fig. 6b). We detected only very rare tdT[−]YFP[+]Ly6D[−]BST2[low] cells, consistent with pDCs being the main source of IFN-I during infection[7,20,27,40]. We examined the localization of both tdT[+]YFP[−] and tdT[+]YFP[+] pDCs in spleens from MCMV-infected mice (Fig. 6c–f). At 36 h p.i., both pDC populations formed clusters in the MZ (Fig. 6c,d), leading to an increase in the proportion of pDCs at this microanatomical site (Fig. 6e). Although pDCs were still detectable in the TCZ of infected animals, their local density was reduced compared with uninfected mice (Fig. 6c,e).

### Only IFN-I[fate_mapped] pDCs entered the TCZ during infection
At 48 h p.i., we detected large aggregates of tdT[+]YFP[−] cells in the MZ, whereas tdT[+]YFP[+] cells were mostly detected in the TCZ (Fig. 6c,d). Hence, we observed an opposite trend in the spatiotemporal repositioning of the pDCs that either produced or did not produce IFN-I within the spleen during infection: whereas most of the pDCs fate-mapped for IFN-I production (YFP[+]) were located in the MZ at 36 h and then in the TCZ at 48 h, the reverse distribution was observed for the pDCs that did not produce IFN-I (YFP[−]) (Fig. 6f). This analysis showed that, at the peak of IFN production, the high recruitment of pDCs to the MZ occurs independent of their ability to produce IFN-I, whereas, at later time points, only the pDCs that have produced IFN-I are licensed to migrate to the TCZ; the other pDCs are retained in the MZ. Next, we sorted DC types from the spleens of infected mice and examined their morphology, also comparing YFP[+] versus YFP[−] pDCs at 60 h p.i. (Fig. 6g,h and Extended Data Fig. 9). A fraction of YFP[+] pDCs acquired an irregular morphology with pseudopods or dendrites (Fig. 6g), harboring significantly lower circularity indices compared both with their YFP[−] counterparts and with steady-state pDCs (Fig. 6h). Hence, the pDCs that had produced IFN-I during infection selectively acquired a dendritic morphology, consistent with their known transcriptomic and functional convergence toward cDCs[27].

### The pDCs clustered around infected cells but few produced IFN-I
Due to the lack of specific markers to track pDCs in situ before their peak IFN-I production, determination of the early kinetics of their microanatomical redistribution in the spleen during infection has not been possible previously. The pDC-Tom mice allowed us to address this issue (Fig. 7). Recruitment of pDCs and their clustering in the MZ were detectable as early as 12 h p.i. (Fig. 7a,b), with a clear increase in the proportion of MZ pDCs approaching the plateau values observed between 18 h and 48 h (Figs. 6e and 7c). Cells replicating MCMV (expressing the

viral immediate early gene 1, *IE1*) were already detectable at 12 h p.i., mainly in the MZ (Fig. 7d–f). The pDC clusters were already localized in close proximity to MCMV-infected cells at 12 h (Fig. 7d), consistent with the proximity between IFN-I-producing pDCs and infected cells observed at later time points[20]. Thus, the recruitment of pDCs to the vicinity of infected cells in the MZ occurred early, already at 12 h p.i., 24 h before their peak IFN-I production.

In the MZ of the spleen of 40-h infected mice, three-dimensional (3D) reconstructions from confocal microscopy images showed that IFN-I-producing pDCs established tight interactions with virus-infected cells (Supplementary Videos 1–3), consistent with in vivo establishment of interferogenic synapses as previously observed only in vitro[41,42].

### The pDC-like cells and CD11c[high] tDCs diverge on activation
We analyzed our FB5P-seq data to compare the responses of the five DC types to MCMV infection in vivo. Four Seurat clusters were identified for pDCs, corresponding to distinct activation states (Fig. 8a and Supplementary Table): quiescent pDCs, intermediate pDCs harboring induction of IFN-stimulated genes (ISGs) but lacking expression of cytokines genes, activated pDCs expressing moderate levels of cytokine genes and IFN-I-producing pDCs expressing high levels of *Ifnl2*, and all the genes encoding IFN-I. CD11c[high] tDCs, cDC1s and cDC2s each split into three clusters, corresponding to quiescent, intermediate and activated states. The pDC-like cells split into two clusters only: quiescent and activated.

The cell types and activation states identified by scRNA-seq were shared across individual mice (Supplementary Fig. 1a), showing similar proportions between mice at the same time p.i. (Supplementary Fig. 1b).

Expression of tdT remained highly selective in pDCs, irrespective of their activation states (Fig. 8b), despite SiglecH downregulation in IFN-I-producing pDCs[7] (Extended Data Fig. 10a). Activation increased autofluorescence but did not induce tdT in other DC types (Fig. 8b). The pDCs remained Ly6D[high]CCR9[high] across activation states. BST2 expression increased on all activated DC types (Fig. 8b), as reported previously[7,43,44]. Activation decreased CX3CR1 expression in CD11c[high] tDCs (Extended Data Fig. 10). Together with their increased BST2 expression (Fig. 8b), this contributed to shifting a fraction of the CD11c[high] tDC population from infected mice into the phenotypic gate used for sorting pDC-like cells, which was corrected by deductive cell-type reassignment on scRNA-seq computational analysis (Extended Data Fig. 7f and Supplementary Table). The cells assigned to CD11c[high] tDCs expressed higher levels of CD11c and GFP than pDC-like cells (Extended Data Fig. 10a). Although the expression of XCR1 on cDC1s and CD11b on cDC2s decreased with activation, these surface markers remained clearly detectable, allowing phenotypic identification of these DC types (Extended Data Fig. 10a), contrasting with near extinction of *Xcr1* expression in activated cDC1s (Extended Data Fig. 10b).

As observed in previous reports based on flow cytometry or bulk RNA-seq data[6], quiescent CD11c[high] tDCs and pDC-like cells were close

**Fig. 6 | Diverging intrasplenic migration patterns and morphological changes between the pDCs producing and those not producing IFN-I during MCMV infection. a**, Strategy for generating SCRIPT mice. **b**, Characterization by flow cytometry of *Ifnb1*-expressing splenocytes from SCRIPT mice at 36 h and 48 h after MCMV infection. Within live nonautofluorescent (AF[−]) cells, pDCs were identified as producing IFN-I (YFP[+]tdT[+], green boxes and contour plot) or not producing (YFP[−]tdT[+], red); other IFN-I producing cells were identified as tdT[−]YFP[+] (violet); their expression of Ly6D and BST2 was examined. **c–f**, Spleen cryosections, 20 μm, from SCRIPT mice infected or not infected by MCMV stained with antibodies against indicated markers. **c**, The masks used for quantification shown on the right of the photographs, with pDCs in black, the TCZ in green and the MZ in blue. **d**, Representative images of a pDC cluster in the MZ at 36 h and of YFP[+] pDCs in the TCZ at 48 h. **e**, Microanatomical distribution of splenic pDCs during MCMV infection. The data shown are from six animals for NI

mice, nine for 36 h, seven for 40 h, eight for 44 h and nine for 48 h, with one whole spleen section analyzed per mouse. **f**, Fraction of YFP[+] versus YFP[−] pDCs in the MZ or TCZ. The data are from the same mice as in **e**. The height of the boxes shows the mean value. A two-sided Wilcoxon's *t*-test was used for the statistical analysis: *P < 0.05, **P < 0.01. **g,h**, Quantitative assessment of the cellular morphology of YFP[+] versus YFP[−] pDCs from 60-h MCMV-infected SCRIPT mice, compared with pDCs and cDC2s from uninfected mice. **g**, One representative confocal microscopy image of each DC type. **h**, Distribution of the circularity indices for individual cells across DC types. The height of the boxes shows the mean value. The data shown are from 2 independent experiments each with 42–53 cells analyzed for each DC type from one mouse, as shown below the graph. The Kruskal–Wallis test was used for the statistical analysis: *P < 0.05, **P < 0.01, ***P < 0.001, ****P < 0.0001.

together in the Uniform Manifold Approximation and Projection (UMAP) space, between pDCs and cDC2s (Fig. 8a). On activation, they moved in opposite directions, with activated pDC-like cells close to

intermediate pDCs, but intermediate or activated CD11c^high tDCs and cDC2s close together (Fig. 8a). Hence, pDC-like cells and CD11c^high tDCs underwent divergent activation in vivo during infection. Indeed,

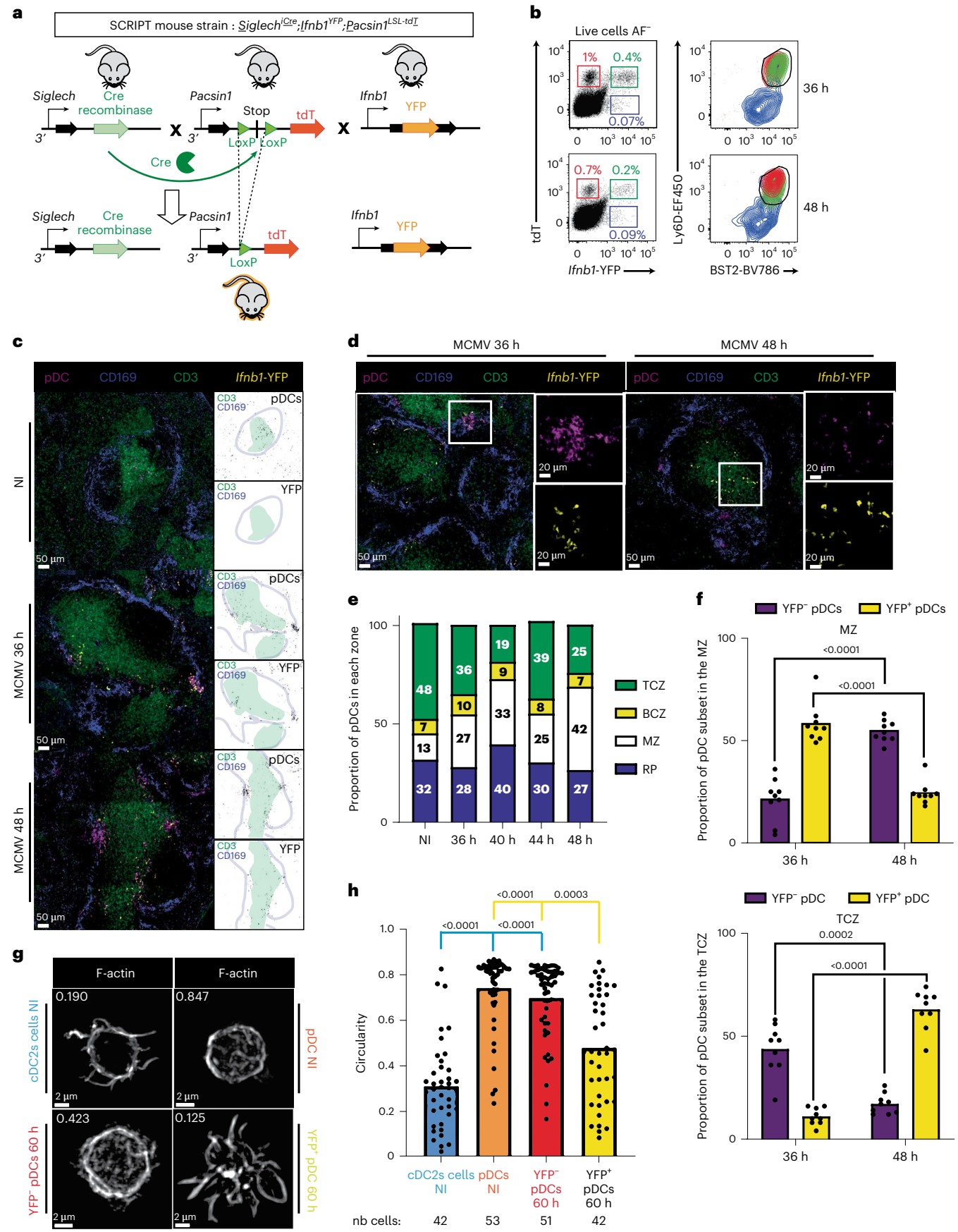

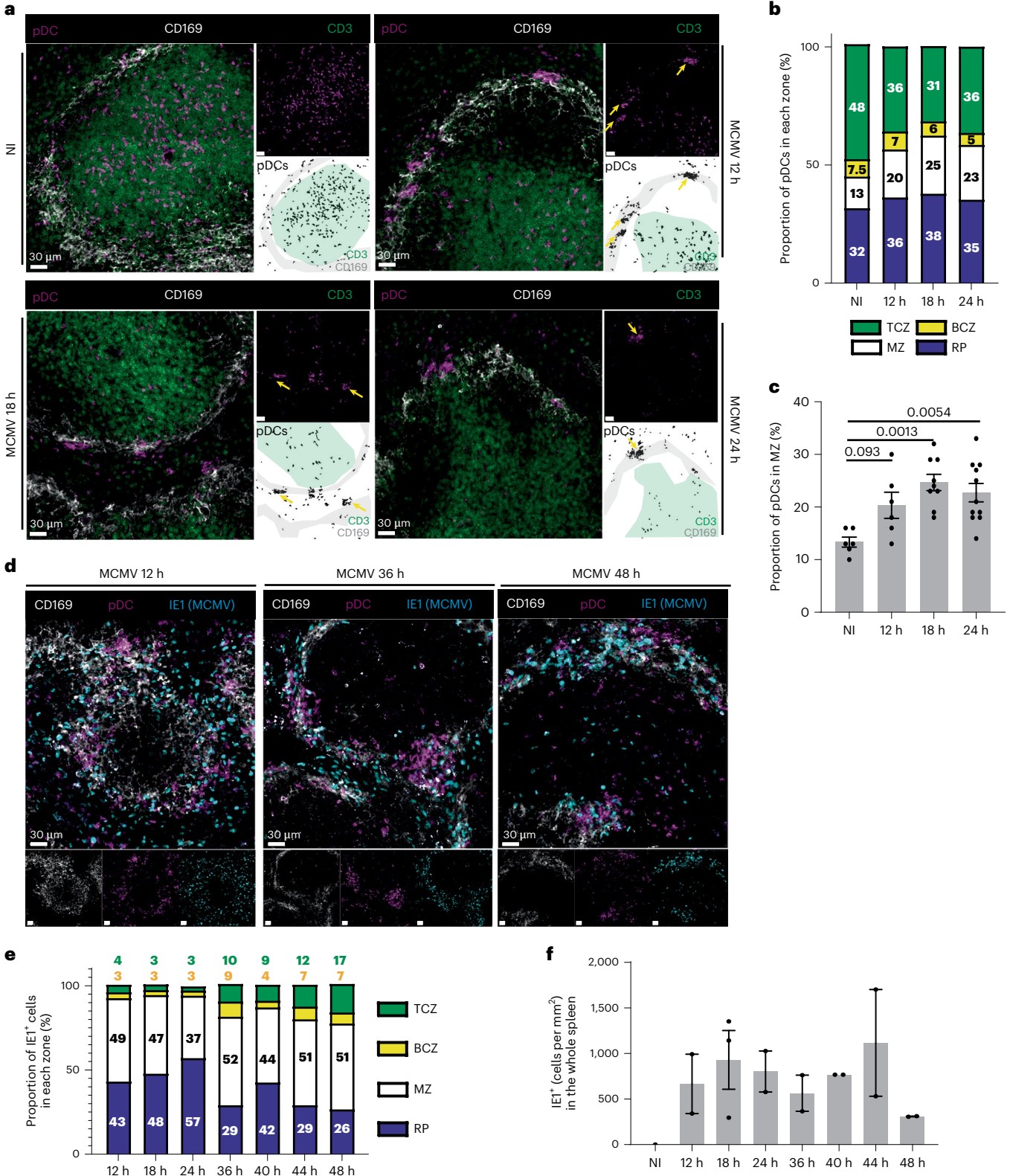

**Fig. 7 | Early during MCMV infection pDCs are recruited at the MZ where they contact infected cells. a–f**, Spleen cryosections, 20 μm, from pDC-Tom mice stained with anti-tdT (magenta), anti-CD169 (white) and anti-CD3 (green) antibodies (**a–c**) or with anti-tdT (magenta), anti-CD169 (white) and anti-IE1 (cyan) antibodies (**d–f**). **a**, Representative images for NI or 12-h, 18-h and 24-h MCMV infection conditions. **b**, Microanatomical distribution of pDCs across the different areas of the spleen, during the course of MCMV infection. The data shown are from six animals for uninfected mice, six animals for 12 h, nine for 18 h and 11 for 24 h, with one whole spleen section analyzed per mouse.

**c**, Quantification of the proportion of pDCs in the MZ. The data are shown as mean ± s.e.m. One-way ANOVA was used for the statistical analysis: $^*P < 0.05$, $^{**}P < 0.01$, $^{***}P < 0.001$, $^{****}P < 0.0001$. **d**, Representative images for 12-h, 36-h and 48-h MCMV infection conditions. **e**, Microanatomical distribution of IE1[+] cells across the different areas of the spleen, during the course of MCMV infection. The data shown are from two animals for each time point analyzed, with one whole spleen section analyzed per mouse. **f**, Number of IE1[+] cells mm[−2] quantified in the whole spleen section. The data are shown as mean ± s.e.m.

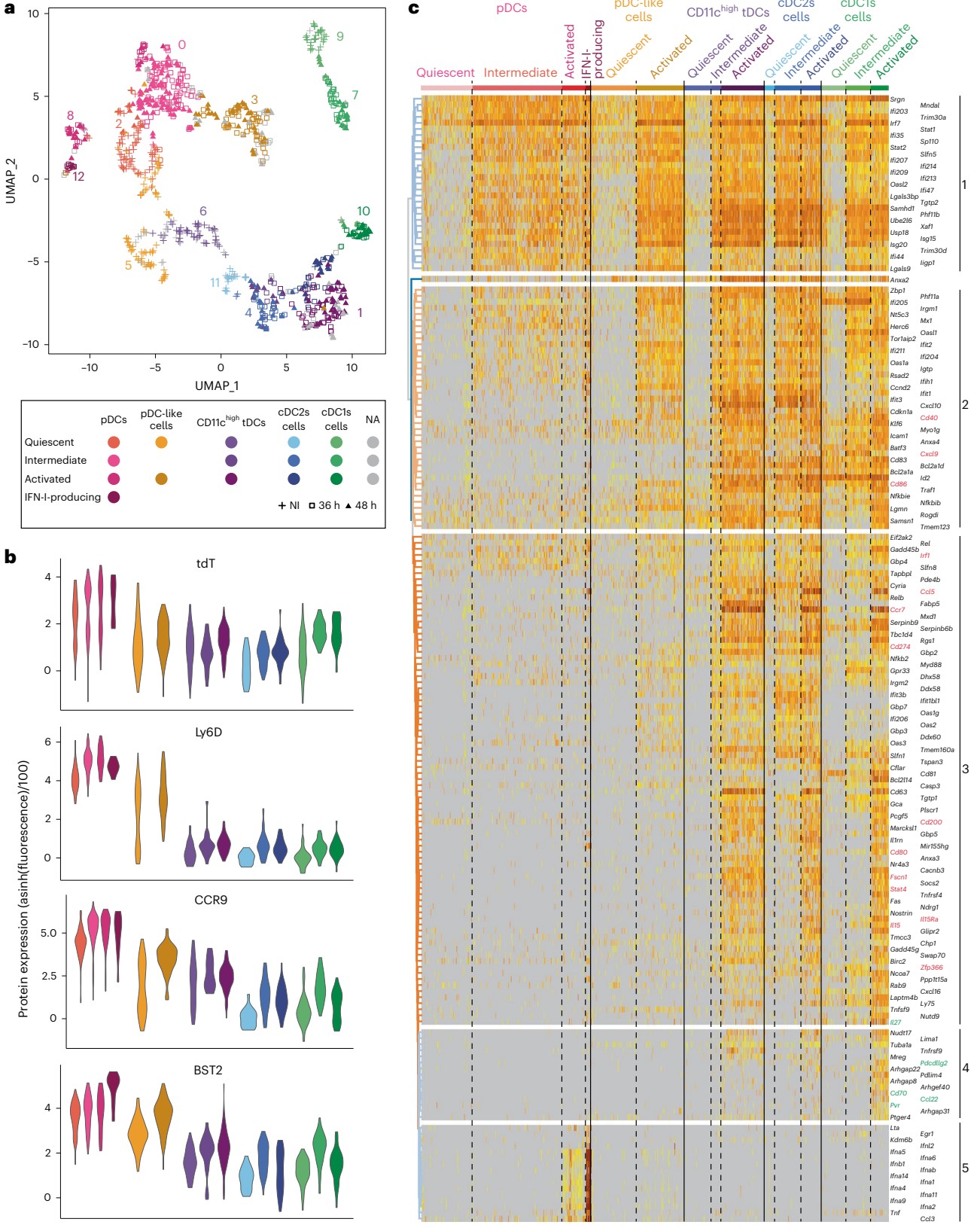

**Fig. 8 | ScRNA-seq confirms the unique capacity of pDCs for high IFN-I/III expression during infection and shows divergent activation patterns for pDC-like cells and tDCs. a**, Projection of assigned DC type and activation states (color code) on to the UMAP space based on FB5P-seq gene expression for DC types isolated from the spleens of eight ZeST mice (three NI; three MCMV infected for 36 h and two infected for 48 h; see the key below the figure; Extended Data Fig. 7a). DC-type assignment is the same as in Fig. 4a. Seurat clusters are indicated on the UMAP (Extended Data Fig. 7b). Activation states were assigned based on mining of the marker genes of Seurat clusters (see Supplementary Table). **b**, Violin plots showing the expression of selected phenotypic markers across DC types and activation states. **c**, Heatmap showing mRNA expression levels of selected genes (rows) across all 951 individual cells (columns), with hierarchical clustering of genes using Euclidean distance, and ordering of individual cells (column) according to their assignment into cell types and activation states using the same color code (top) as in **a**. The color scale for gene expression levels is the same as in Fig. 4c.

whereas only 84 genes were differentially expressed between CD11c^high tDCs and pDC-like cells at the steady state, this number increased to 527 for activated cells (Supplementary Table). Whereas all DC types induced ISGs (Fig. 8c, cluster 1), only CD11c^high tDCs, cDC2s and cDC1s induced high levels of genes linked to DC maturation/migration and interactions with T cells, encompassing *Fscn1*, *Ccr7*, *Il15*, *Il15ra*, *Cd80*, *Cd200* and *Cd274* (Fig. 8c, cluster 3). Another set of ISGs and genes associated with DC interactions with T cells gradually increased in expression from intermediate to activated to IFN-I-producing pDCs (Fig. 8c, cluster 2). These genes were induced to similar levels in activated pDC-like cells, but to even higher levels in activated CD11c^high tDCs, cDC2s and cDC1s. Only a few genes were differentially expressed between CD11c^high tDCs and cDC2s: 16 at steady state, 1 at the intermediate activation state and 11 for activated cells (Supplementary Table). The number of genes differentially expressed between pDCs and pDC-like cells remained stable over activation: 113 at steady state versus 138 in activated cells (Supplementary Table). Hence, on activation, pDC-like cells behaved more like pDCs, although not producing IFN-I (Fig. 8c, cluster 5), whereas CD11c^high tDCs converged further toward cDC2s.

Activated pDC-like cells maintained a higher *Ly6c2* expression (Extended Data Fig. 10c) and a lower CD11c MFI (Extended Data Fig. 10a) than CD11c^high tDCs, confirming the reliability of these markers even in inflammation. Genes encoding MHC-II molecules, such as *H2-DMb2* and *H2-Eb1*, were higher in CD11c^high tDCs than in pDC-like cells, reaching the same levels as in cDC2s (Extended Data Fig. 10c). *Tmem176a* and *Tmem176b*, encoding cell-surface markers, were higher in CD11c^high tDCs than in pDC-like cells, pDCs, cDC1s and to a lesser extent cDC2s, across activation conditions (Extended Data Fig. 10c,d). *Cd8a* expression remained lower in cDC2s across activation conditions (Extended Data Fig. 10c). *Ms4a4c* was higher in activated pDC-like cells, intermediate and activated CD11c^high tDCs and cDC2s than in pDCs and cDC1s (Extended Data Fig. 10c). *Apod* was expressed to higher levels in pDC-like cells, although it was induced in all other DC types except cDC2s on activation (Extended Data Fig. 10c,d).

Finally, beyond confirming the unique ability of pDCs to produce IFN-I/III during MCMV infection[7,20,40], our analysis unraveled selective induction in cDC1s of genes encoding costimulation molecules or cytokines involved in the crosstalk with natural killer (NK) or T cells, encompassing *Cd70*, *Pdcd1lg2*, *Pvr* and *Ccl22* (Fig. 8c, cluster 4), *Il27* (Fig. 8c, cluster 3), *Il18* and *Il12b* (Extended Data Fig. 10e). This is consistent with a division of labor between DC types, with cDC1s promoting NK and CD8 T cell antiviral responses[45,46].

## Discussion

We generated the first reporter mouse model, to our knowledge, specifically and efficiently tagging pDCs, across organs and activation conditions: the pDC-Tom (*Siglech^iCre*;*Pacsin1^LSL-tdT*) mice.

We did not provide data to solve the current debate on pDC ontogeny[4,5,47], because tdT expression in our model starts only from the late CD11c^+ pre-pDC differentiation state that is common to the 'lymphoid' and 'myeloid' paths.

Using pDC-Tom mice, we could study the choreography of the relocation of all pDCs in the spleen during MCMV infection. All pDCs were attracted to the microanatomical sites of viral replication early p.i. Thus, the failure of most pDCs to produce IFN-I is unlikely to result from restricted access to infected cells. Alternatively, pDC tight clustering raises the hypothesis that the first to produce IFN-I may have repressed this function in their neighbors, through a quorum-sensing mechanism, to prevent excessive inflammation and consequent immunopathology. Future studies using pDC-reporter mice and pDC-specific genetic manipulations should help to better understand this regulation.

We generated the first side-by-side transcriptomic comparison, to our knowledge, of pDCs, pDC-like cells and CD11c^high tDCs, through scRNA-seq. We uncovered a divergent activation between CD11c^high tDCs and pDC-like cells. Only CD11c^high tDCs underwent a maturation closely resembling that of cDCs.

CD11c^high tDCs and activated cDC2s selectively expressed high levels of *H2-DMb2* and *Tmem176a/b*, new candidate markers for these DC types. *Tmem176a/b* colocalizes with human leukocyte antigen (HLA)-DM in the late endolysosomal system, promotes antigen presentation to naive T cells and contributes to tolerance/immunosuppression[48–50]. This raises the question of the division of labor between CD11c^high tDCs and cDC2s, including for CD4 T cell tolerance. Our scRNA-seq dataset will be a valuable resource to mine the gene expression profiles of pDC-like cells and CD11c^high tDCs, compared with pDCs, cDC1s and cDC2s, to infer and experimentally test hypotheses on their functional specialization and molecular regulation.

## Online content

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

## Methods

### Mice

All animal experiments were performed in accordance with national and international laws for laboratory animal welfare and experimentation (EEC Council Directive 2010/63/EU, September 2010). Protocols were approved by the Marseille Ethical Committee for Animal Experimentation (registered by the Comité National de Réflexion Ethique sur l'Expérimentation Animale under no. 14; APAFIS no. 1212-2015072117438525 v.5 and APAFIS no. 21626-2019072606014177 v.4). C57BL/6 mice were purchased from Janvier Labs. All other mouse strains were bred at the Centre d'ImmunoPhénomique (CIPHE) or the Centre d'Immunologie de Marseille-Luminy (CIML), under specific pathogen free-conditions and in accordance with animal care and use regulations. Mice were housed under a 12 h dark:12 h light cycle, with a temperature range of 20–22 °C and a humidity range of 40–70%. *Siglech^iCre* mice (*B6-Siglech^tm1(iCre)Ciphe*)[20] and *Pacsin1^LoxP-STOP-LoxP-tdTomato(LSL-tdT)* (*B6-Pacsin1^tm1(tdT)Ciphe*) mice were generated by CIPHE. *Siglech^iCre;Pacsin1^LSL-tdT* mice (pDC-Tom) were generated by crossing *Siglech^iCre* mice with *Pacsin1^LSL-tdT* mice, and then maintained and used in a double homozygous state. SCRIPT mice were generated by crossing pDC-Tom mice with *Ifnb1^Eyfp* mice (*B6.129-Ifnb1^tm1Lky*)[39], and then maintained and used in a triple homozygous state. ZeST mice were generated by crossing *Zbtb46^GFP* mice (*B6.129S6(C)-Zbtb46^tm1.1Kmm/J*)[26] with pDC-Tom mice and were used in a heterozygous state. All animals used were sex and age matched (used between 8 and 16 weeks of age).

### Virus and viral infection

Virus stocks were prepared from salivary gland extracts of 3-week-old, MCMV-infected BALB/c mice. All mice used in the experiments were infected intraperitoneally with $10^5$ plaque-forming units of Smith MCMV and sacrificed at the indicated time points.

### Cell preparation for flow cytometry analysis or cell sorting

Spleens or lymph nodes were harvested and submitted to enzymatic digestion for 25 min at 37 °C with Collagenase IV (Worthington biochemical) and DNase I (Roche Diagnostics). Organs were then mechanically digested and passed over 100-µm cell strainers (Corning). Bone marrow cells were flushed from mouse femurs. Red blood cells (RBCs) were then lysed by using RBC lysis buffer (Life Technologies) for spleen and bone marrow cell preparation. Livers were harvested, minced and submitted to enzymatic digestion, as for the spleen. Liver pieces were then crushed and the cell suspension obtained was washed 2× with phosphate-buffered saline (PBS) 1×, before performing a 80:40 Percoll gradient. Small intestines were harvested, opened longitudinally and then cut into 1-cm pieces. Pieces were washed extensively with PBS 1×, then incubated 3× at 37 °C on shaking (200 r.p.m.) with PBS 1× containing 2% fetal calf serum (FCS) and 5 mM EDTA. At the end of each incubation, supernatants were collected and centrifuged. Pelleted cells, mainly intraepithelial lymphocytes, from the three incubations were pooled together and submitted to a 67:44 Percoll gradient. Cells isolated from the middle ring of Percoll gradients were washed once with PBS 1×, then used for flow cytometry.

### Flow cytometry analysis

Extracellular staining was performed in PBS 1× supplemented with 2 mM EDTA (Sigma-Aldrich) and 1% FCS. All extracellular staining was performed for at least 30 min at 4 °C. Dead cell staining (LIVE/DEAD Fixable Aqua Dead Cell Stain or Blue Dead Cell Stain, Life Technologies) was performed in PBS 1× according to the manufacturer's recommendations. Samples were acquired with a FACS Fortessa X20 (BD Biosciences) or sorted with FACS ARIA III (BD Biosciences) using BD Diva v.9.0. All data were analyzed with FlowJo v.10.8.1 software. For the unsupervised analysis, t-SNE plots were generated with the dedicated plugin in FlowJo software. Dimensional reduction was performed on singlets/nonautofluorescent/live dead⁻/lineage⁻/CD19⁻/CD11c⁺ and/or

SiglecH⁺ cells. Dimension reduction was calculated with the following markers: B220, BST2, CD11b, CD11c, CCR9, CX3CR1, Ly6D, SiglecH, tdT, XCR1 and *Zbtb46*-GFP. We used the HyperFinder plugin of FlowJo for the generation of unsupervised gating strategy. The calculation was performed on singlets/nonautofluorescent/live dead⁻ cells. We defined the pDC population for the calculation as being lineage⁻/CD19⁻/CD11b⁻/XCR1⁻/CD11c^low/BST2^high/tdT⁺. The software calculated the best and fastest gating strategy with the following markers: B220, BST2, CD11b, CD11c, CD19, CCR9, CX3CR1, lineage (CD3/Ly6G/NK1.1), Ly6D, SiglecH and XCR1.

### Spectral flow cytometry analysis

All antibodies were purchased from Becton Dickinson Biosciences, BioLegend or eBioscience. Dead cells were discriminated in all experiments using Live/dead (ZombieNIR) fixable dead stain (Life Technologies). All staining was carried out with Fc-block (2.4G2 and 9E9) in staining buffer (PBS, 2 mM EDTA, 0.5% bovine serum albumin (BSA) and 20% Brilliant Violet stain buffer (BD Biosciences)). Cell suspensions were stained first for 15 min at 37 °C by monoclonal antibodies targeting chemokine receptors and certain antigens (CCR9, CD26, CD64, CX3CR1, SiglecH and XCR1) and for 25 min more by monoclonal antibodies targeting other antigens (B220, Bst2, CD45, CD3, CD8a, CD11b, CD11c, CD19, CD88, F4/80, IgD, IgM, Ly6C, Ly6D, Ly6G, NK1.1 and MHC-II). Cells were washed 3× in FACS buffer (PBS, 2 mM EDTA and 0.5% BSA) and resuspended in PBS. In all flow cytometric plots, doublets, aggregates, dead cells and autofluorescence were excluded. Acquisition was performed on an Aurora five lasers (Cytek Biosciences) using SpectroFlo v.3.0.1 software (Cytek Biosciences); the quality control (QC) indicates a similarity and complexity index of 11.7. Data were analyzed using OMIQ (app.OMIQ.ai). First, the data were run through flowCut[51] to check for aberrant signal patterns or events. Second, the data were cleaned by manual gating to remove doublets, debris and dead cells. Third, the autofluorescence of the tissue was subtracted. UMAP[52] was run to reduce the dimensions to a two-dimensional space and thus group phenotypically similar events into 'islands' to illustrate differences both between and inside each population. PARC (phenotyping by accelerated refined community)[53] was subsequently used to cluster the events based on UMAP parameters.

### ScRNA-seq data generation

For the generation of the scRNA-seq data, we followed the FB5P method previously published[28]. Briefly, single cells were FACS sorted into ice-cold 96-well PCR plates (Thermo Fisher Scientific) containing 2 µl of lysis mix per well. Immediately after cell sorting, each plate was covered with an adhesive film (Thermo Fisher Scientific), briefly spun down in a benchtop plate centrifuge and frozen on dry ice. The reverse transcription (RT) reaction was performed with SuperScript II (Thermo Fisher Scientific) in the presence of RNaseOUT (Thermo Fisher Scientific), dithiothreitol (Thermo Fisher Scientific), betaine (Sigma-Aldrich), MgCl₂ (Sigma-Aldrich) and well-specific template-switching oligonucleotide. For complementary DNA amplification, KAPA HiFi HotStart ReadyMix (Roche Diagnostics) was used with adapted primers. For library preparation, amplified cDNA from each well of a 96-well plate was pooled, purified with two rounds of 0.6× solid-phase reversible immobilization beads (AmpureXP, Beckman or CleanNGS, Proteigene) and finally eluted in nuclease-free water. After tagmentation and neutralization, tagmented cDNA was amplified with Nextera PCR Mastermix containing Nextera i5 primer (Illumina) and customized i7 primer mix. Libraries generated from multiple 96-well plates of single cells and carrying distinct i7 barcodes were pooled for sequencing on an Illumina NextSeq2000 platform, with 100 cycles of P2 flow cells, targeting $5 × 10^5$ reads per cell in paired-end, single-index mode with the following cycles: Read1 (Read1_SP, 67 cycles), Read i7 (i7_SP, 8 cycles) and Read2 (Read2_SP, 16 cycles). Two to three individual mice were used as a source for the single cells for each time point, with three

independent sorts performed with two or three animals each time (sorts for mice nos. 56 and 58 on 3 November 2020, for nos. 52, 53 and 61 on 17 December 2020 and for nos. 81, 84 and 86 on 11 February 2021); sorting plates were frozen until all samples had been collected and all libraries were generated and sequenced simultaneously to avoid eventual batch effects.

## Bioinformatics analyses of scRNA-seq data

A mark-up file of how the scRNA-seq analysis was performed is provided (Supplementary Fig. 2). FB5P-seq data were aligned and mapped to a reference genome using STAR (v.2.5.3a) and HTSeqCount (v.0.9.1) and processed to generate a single-cell, unique molecular identifier (UMI) counts matrix as described[28]. The counts matrix was loaded to R (v.4.0.3) and Seurat (v.3.2.0)[54] was used for downstream analyses as described[27]. Gene expression is shown as log(normalized values) and protein expression as inverse hyperbolic arcsine (asinh) of fluorescence intensity-scaled values (asinh(fluorescence)/100). For dimensionality reduction, we performed UMAP, using the RunUMAP function. The differentially expressed genes (DEGs) were determined using the FindMarkers function.

A convergent transcriptional reprogramming occurs in all DC types during their maturation, which can lead to their clustering primarily according to their activation states rather than to their cell types[31]. In particular, many of the genes that are specific for a given DC type at steady state are strongly downregulated on activation[30–32], such that they cannot be used individually to identify DC types[33]. These phenomena recurrently caused issues for identifying DC types in the analyses of certain scRNA-seq data, which led to an inadequate inflation of DC subset nomenclature, with new DC subset names coined for clusters that actually corresponded to activation states of already identified DC types. This problem is exemplified by the current use of the name 'DC3' for mature DC clusters in scRNA-seq studies, whereas 'DC3' is also being used for a human DC type that is ontogenetically and functionally distinct from human pDCs, cDC1s, cDC2s and monocyte-derived DCs, as discussed in a recent commentary[55]. Similar to the authors of this commentary, we emphasize, in scRNA-seq data analysis, the necessity for clearly distinguishing distinct DC types from different activation states of the same DC type. To achieve this aim, proper strategies are required for robust assignment of a cell-type identity to each individual cell, irrespective of its activation state, in a manner enabling resolution of the heterogeneity of mixed-cell clusters. This can be achieved by integrating phenotypic and transcriptomic data (as enabled with index sorting or CITE-seq[33]) or by performing a gene set enrichment analysis at the level of individual cells, by using the single-cell CMap algorithm[56] with specific composite up/down gene modules carefully defined from an independent dataset. We used both strategies in the present study as a technical guide to help readers in their future analyses of scRNA-seq data.

A first Seurat analysis was performed only on cells from uninfected mice (345 cells after QC, 2 of which were removed due to lack of index sorting data). Clusters were identified based on either gene expression using Seurat (k-nearest neighbor = 5; resolution = 0.2) or phenotypic marker expression using Rphenograph[57] (v.0.99.1) (number of nearest neighbors k = 20), taking into account the protein expression for GFP, B220, Ly6D, XCR1, CX3CR1, BST2, SiglecH, CCR9, CD11b and CD11c (but not for tdTomato). A CMap analysis[56] was performed on this dataset using cell-type-specific signatures (tDCs, cDC2s, cDC1s and pDCs) established on reanalysis with the BubbleGUM GeneSign module[58] of a published[29], independent, bulk RNA-seq dataset (Gene Expression Omnibus (GEO) accession no. GSE76132), and a relative pDC_vs_pDClike signature retrieved from a previously published study[4]. The GSE76132 dataset encompassed four cell types, tDCs, cDC2s, cDC1s and pDCs, analyzed through bulk RNA-seq. We retrieved and reanalyzed these data to generate signatures for each DC type compared with the three others. Hence, we ended up with four composite signatures, one for

each DC type, with its 'UP' and 'DN' genes when compared with the three other DC types. The study by Rodrigues et al.[4] encompassed only two cell types, pDCs and pDC-like cells, analyzed through scRNA-seq. Hence, from this dataset, we retrieved one composite signature encompassing the genes expressed higher in pDCs above pDC-like cells ('UP') and reciprocally the genes less expressed in pDCs than in pDC-like cells ('DN'). The CMap algorithm has been made available on Github (https://github.com/SIgN-Bioinformatics/sgCMAP_R_Scripts), as well as an example and recommendations on how to use it (https://github.com/DalodLab/MDlab_cDC1_differentiation/blob/main/scRNAseq_pipeline.md#cell-annotation-using-cmap).

Integration of Seurat and Rphenograph cluster information together with the CMap scores allowed identification of, and focus on, bona fide steady-state pDCs, pDC-like cells, CD11c^high tDCs, cDC1s and cDC2s (205 total cells), from which we computed relative signatures based on all pairwise comparisons between cell types, using the FindMarkers function (default parameters), for consecutive single-cell CMap analyses.

A second Seurat analysis was then performed on cells from both uninfected and MCMV-infected mice (1,132 cells after QC). Clusters of contaminating cell types (macrophages, NK cells and a small cluster of proliferative cells of mixed types) were identified by their top markers using the FindMarkers function (test.use = 'bimod') and removed (181 cells). A third Seurat analysis was performed on the remaining cells (951 cells).

Clusters were identified based on gene expression using Seurat (k-nearest neighbor = 9; resolution = 0.7) or based on phenotypic marker expression using Rphenograph (number of nearest neighbors k = 50). Integration of Seurat and Rphenograph clusters allowed identification of unambiguous cell types and activation states (851 total cells), which was corroborated on performing a single-cell CMap analysis using the relative signatures identified in the previous step. DEGs between cell types at equivalent activation states or between activation states for a given cell type were then extracted, using the FindMarkers function (default parameters, threshold for adjusted P < 0.05) (see Supplementary Table).

The heatmaps were plotted using the Gene-E program.

## Immunohistofluorescence, microscopy and image analysis

Organs were fixed with Antigen Fix (Diapath) for 2 h for the small intestine, colon and lymph nodes or 4 h for the spleen at 4 °C, and then washed several times in phosphate buffer (PB; 0.025 M NaH$_2$PO$_4$ and 0.1 M Na$_2$HPO$_4$). Organs were then immersed in a solution of 30% sucrose overnight at 4 °C. Organs were then embedded in OCT (optimal cutting temperature; Sakura), snap frozen and stored at −80 °C. Cryosections, 20 μm, were performed using a microtome (Leica 3050s Cryostat) at temperatures between −20 °C and −22 °C. For immunostaining, sections were blocked with PB, 0.1% Triton X-100 and 2% BSA for 30 min at room temperature and then stained overnight at 4 °C with primary antibodies (Supplementary Table) diluted in PB, 0.1% Triton X-100 and 2% BSA. After several washings with PB, sections were then stained in PB, 0.1% Triton X-100 and 2% BSA with secondary antibodies (Supplementary Table) for 2 h at 4 °C. To analyze YFP signal, after the incubation with secondary antibodies, sections were washed and incubated with PB, 0.1% Triton X-100, 2% BSA and 5% rabbit serum for 30 min at room temperature. Sections were then stained with anti-GFP antibodies directly labeled with Alexa 488 for 2 h at 4 °C. Finally, sections were washed with PB and mounted with a coverslip and Prolong Antifade Gold mounting medium (Life Technologies). Whole sections were acquired by spectral confocal microscope (Zeiss LSM 880) with ×20 or ×40 objectives. Pictures were then analyzed using ImageJ v.1.52p software, including through the development of specific macros (https://github.com/Imagimm-CIML/Micro-anatomical-location-of-splenic-IFNpos-vs-IFNneg-pDC-). The TCZ was defined as a CD3-rich region within the WP and the BCZ as

a B220-rich area within the WP. The RP was defined as an F4/80-rich region and the MZ as the space between the CD169 staining and the F4/80 staining. For the calculation of pDC counts mm$^{-2}$, the function 'analyze particles' was used and a threshold for tdT intensity and the size (>8 µm$^2$) was applied. For the quantification of the repartition of pDCs on infection, the intensity of tdT was calculated in the different zones after a threshold for the intensity had been applied. A ratio between the intensity in each zone and the total intensity of the whole section was then calculated.

### Cell immunofluorescence and cell morphology analysis

The different cell subsets were FACS sorted in a cold 5-ml FACS tube containing 1 ml of RPMI (Roswell Park Memorial Institute) medium supplemented with 10% FCS, 1% L-glutamine (Gibco), 100 U ml$^{-1}$ of penicillin–streptomycin, 1% nonessential amino acids, 1% sodium pyruvate and 0.05 mM 2-mercaptoethanol. After washing in PBS, cells were resuspended in RPMI and 1% FCS. Then, 1,000–50,000 cells were allowed to adhere for 1 h at 37 °C to coverslips coated with 5 µg cm$^{-2}$ of poly(D-lysine) (Sigma-Aldrich). Cells were then washed in PBS, fixed in 4% paraformaldehyde for 10 min and permeabilized/blocked in PBS, 0.2% Triton X-100, 2% FCS, 2% rat serum, 2% goat serum and 2% donkey serum. F-actin staining was performed at 4 °C overnight in PBS, 0.1% Triton X-100 and 2% FCS with phalloidin AF405plus (Life Technologies). Samples were washed in PBS and mounted with Prolong Antifade Gold mounting medium. Images were acquired by spectral confocal microscope (Zeiss LSM 880) with a ×63 objective and analyzed with ImageJ v.1.52p software. For the cell morphology, a binary image was created for each individual cell based on the F-actin staining. The circularity index ($(4\pi \times area)/(perimeter \times 2)$) was then calculated with the adequate function of the software.

### RT-qPCR

For each indicated cell type, 100 cells were FACS sorted into ice-cold, 96-well PCR plates (Thermo Fisher Scientific) containing 10 µl of TCL buffer (QIAGEN) supplemented with 1% 2-mercaptoethanol (Invitrogen). Cell lysates were sealed, vortexed and spun down at 300$g$ and 4 °C for 1 min, immediately placed on dry ice and transferred for storage at −80 °C. The SS2 protocol was performed on cell lysate as previously described[27]. Briefly, plates containing cell lysates were thawed on ice, followed by RNA purification and first-strand synthesis. Complementary DNA was amplified by PCR for 22 cycles. Serial dilutions of the cDNA obtained were used to perform quantitative (q)PCR with the ONeGreenFAST kit (Ozyme) and run on a 7500 Real Time PCR System apparatus (Applied Biosystems). Relative gene expression was calculated using the $\Delta\Delta^{Ct}$ method with *Actb* as housekeeping gene for normalization. The primers used were as follows: *Actb* forward 5′-GGCTGTATTCCCCTCCATCG-3′; reverse 5′-CCAGTTGGTAACAATGCCATGT-3′; *Xcr1* forward 5′-CCTACGTGAAACTCTAGCACTGG-3′; reverse 5′-AAGGCTGTAGAGGACTCCATCTG-3′; *Siglech* forward 5′-GGAGGCAAAACATGGAATTTATG-3′; reverse 5′-CACATCACATTGGTAGGACGAC-3′; *Tmem176a* forward 5′-GCCGGATGCTCATTGCTAAG-3′; reverse 5′-ATGGCCTATGTAGAGGGTTCC-3′; *Tmem176b* forward 5′-CAGTCCGCTCACATCAGCAT-3′; reverse 5′-GCTGCCCATAGTGGATTCTGG-3′; *Apod* forward 5′-TCACCACAGCCAAAGGACAAA-3′; reverse 5′-CGTTCTCCATCAGCGAGTAGT-3′.

### Statistical analysis

No statistical methods were used to predetermine sample sizes but our sample sizes are similar to those reported in previous publications[20,27]. Data distribution was assumed to be normal but this was not formally tested. All quantifications were performed with awareness of experimental groups, meaning not in a blinded fashion. Animals were matched in age and gender between experimental groups, without randomization. No animals or data were excluded. Statistical parameters including the definitions and exact value of $n$ (number of biological replicates and total number of experiments) and the types of statistical tests are reported in the figures and corresponding legends. Statistical analyses were performed using Prism v.8.1.2 (GraphPad Software) or R v.4.0.3 statistical programming language. Statistical analysis was conducted on data with at least three biological replicates. Comparisons between groups were planned before statistical testing and target effect sizes were not predetermined. Error bars displayed on graphs represent the mean ± s.e.m. Statistical significance was defined as: $^*P < 0.05$, $^{**}P < 0.01$, $^{***}P < 0.001$ and $^{****}P < 0.0001$.

### Reporting summary

Further information on research design is available in the Nature Portfolio Reporting Summary linked to this article.

## Data availability

The scRNA-seq data have been deposited in the GEO repository under accession no. GSE196720. All other data generated or analyzed during the present study are included in this report (and its Supplementary Information files). Source data are provided with this paper.

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

## Acknowledgements

We thank CIPHE for the generation of *Siglech*$^{iCre}$ and *Pacsin1*$^{LSL-tdT}$ mice and their assistance in the breeding of mice, as well as the staff of the CIML mouse houses, flow cytometry, histology, genomics bioinformatics and imaging (ImagImm) core facilities. We thank S. Henri (CIML) for the generous gift of the *Zbtb46*$^{GFP}$ mice. We thank the two master trainees, A. Trinh and D. Anselmo, as well as M. Toxé, for technical assistance in some experiments. We thank C. Dong for help in the preprocessing of the FB5P-seq data. We thank the Centre de Calcul Intensif d'Aix-Marseille for granting access to its high-performance computing resources. We thank M. Bajénoff for discussions and advice. This research was funded by grants from the Fondation pour la Recherche Médicale (nos. DEQ20110421284 and DEQ20180339172, Equipe Labellisée, to M.D.) and from the French National Research Agency (ANR, no. ANR-21-CO12-0001-01, 'RIPCOV' to M.D. and no. ANR-21-CE15-0044-01, 'DECITIP' to E.T.). We also thank

the DCBIOL Labex (ANR-11-LABEX-0043, grant no. ANR-10-IDEX-0001-02 PSL*), the A*MIDEX project (grant no. ANR-11-IDEX-0001-02), funded by the French government's Investissements d'Avenir program managed by the ANR, and institutional support from CNRS (French National Centre for Scientific Research), INSERM (French National Institute of Health and Medical Research) and Aix-Marseille Université and Marseille Immunopole. This work was supported by the ANR through the Investments for the Future program (France-BioImaging, no. ANR-10-INBS-04).

## Author contributions

M.V. designed, performed and analyzed most of the experiments and wrote the manuscript. N.C. performed part of the microscopy experiments, genotyping of mutant mice, and generation and stabilization of SCRIPT mice in the triple homozygous state. T.-P.V.M. performed the bioinformatics analysis and contributed to writing the manuscript. D.P. and C.M. contributed to design, and performed and analyzed the spectral flow cytometry experiments. K.R. contributed to performing and analyzing the micro-bulk RT-qPCR experiments. R.R. gave advice on the spectral flow cytometry experiments. K.N. performed the crosses and genotyping for the generation and stabilization of the S-RFP and pDC-Tom mouse strains. G.B. contributed to the genotyping of mutant mice and performed the crosses and genotyping for the generation and stabilization of the ZeST mice. L.G. trained M.V. for the generation of the FB5P-seq data. P.M. coordinated the generation and first-level analysis of the FB5P-seq data. E.T. directed and contributed to funding the study, designed experiments, performed and analyzed some experiments and wrote the manuscript. M.D. directed and funded the study, designed experiments, contributed to the bioinformatics analyses and wrote the manuscript. All authors contributed to the revision of the manuscript.

## Competing interests

The authors declare no competing interests.

## Additional information

**Extended data** is available for this paper at https://doi.org/10.1038/s41590-023-01454-9.

**Correspondence and requests for materials** should be addressed to Michael Valente, Elena Tomasello or Marc Dalod.

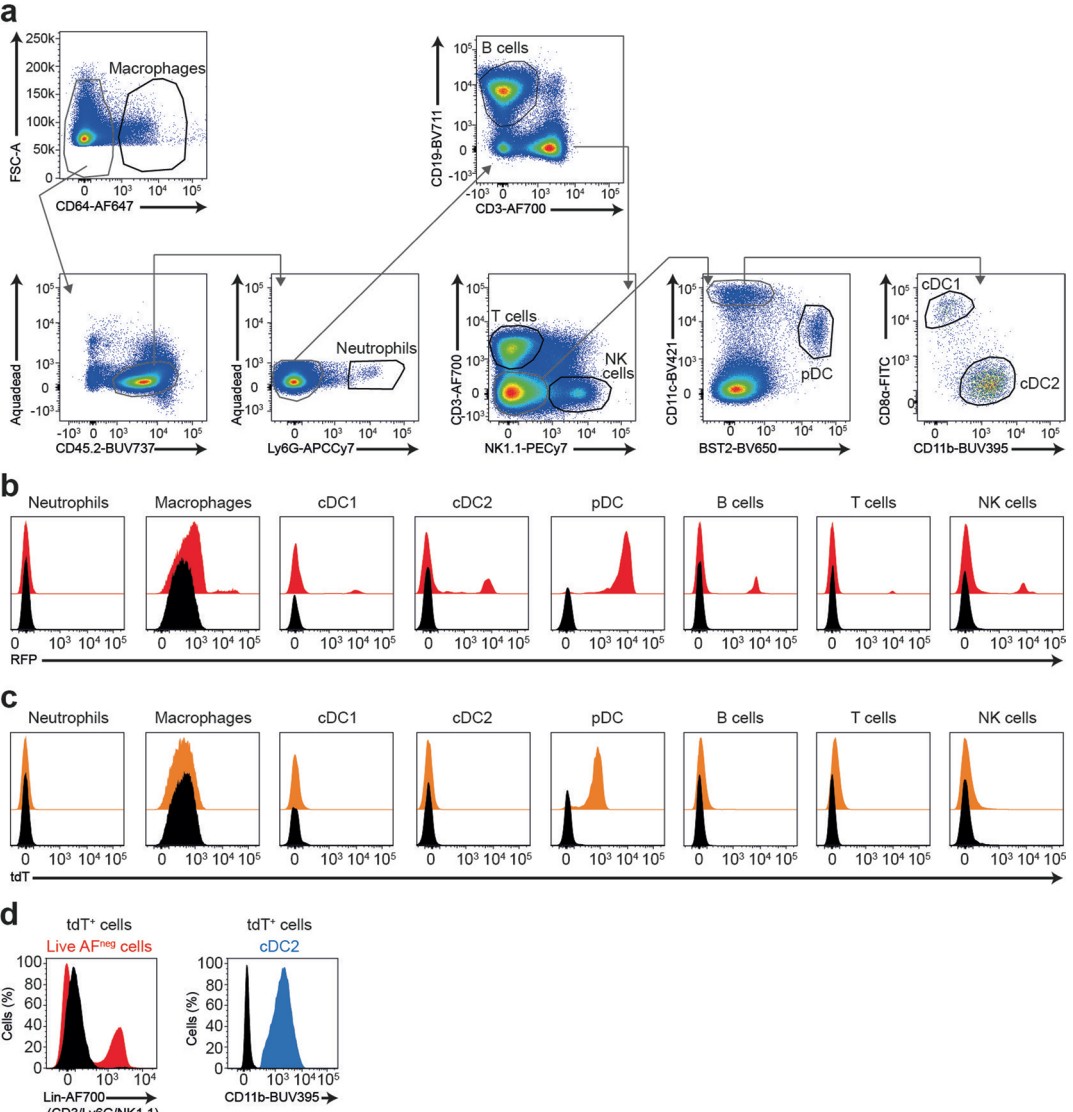

**Extended Data Fig. 1 | Expression of RFP and tdT in different immune lineages in S-RFP and pDC-Tom mice. a**, Gating strategy used for the identification of the different immune cell types studied. **b**, **c**, Splenocytes were isolated from S-RFP (**b**) or pDC-Tom (**c**) animals, stained with fluorescently labeled antibodies and analyzed by flow cytometry. The expression of RFP (**b**, red histograms) or tdT (**c**, orange histograms) was evaluated in different immune cell types as indicated.

**d**, The expression of Lineage cocktail antibodies and CD11b was assessed on tdT+ cells and compared to Live non-autofluorescent cells (Red) (Lineage, left panel) or cDC2 (Blue) (CD11b, right panel). C57BL/6 mice were used as negative controls (black histograms). The data shown are from one mouse representative of 8 animals for (**b**) and 6 animals for (**c**, **d**) from 2 independent experiments.

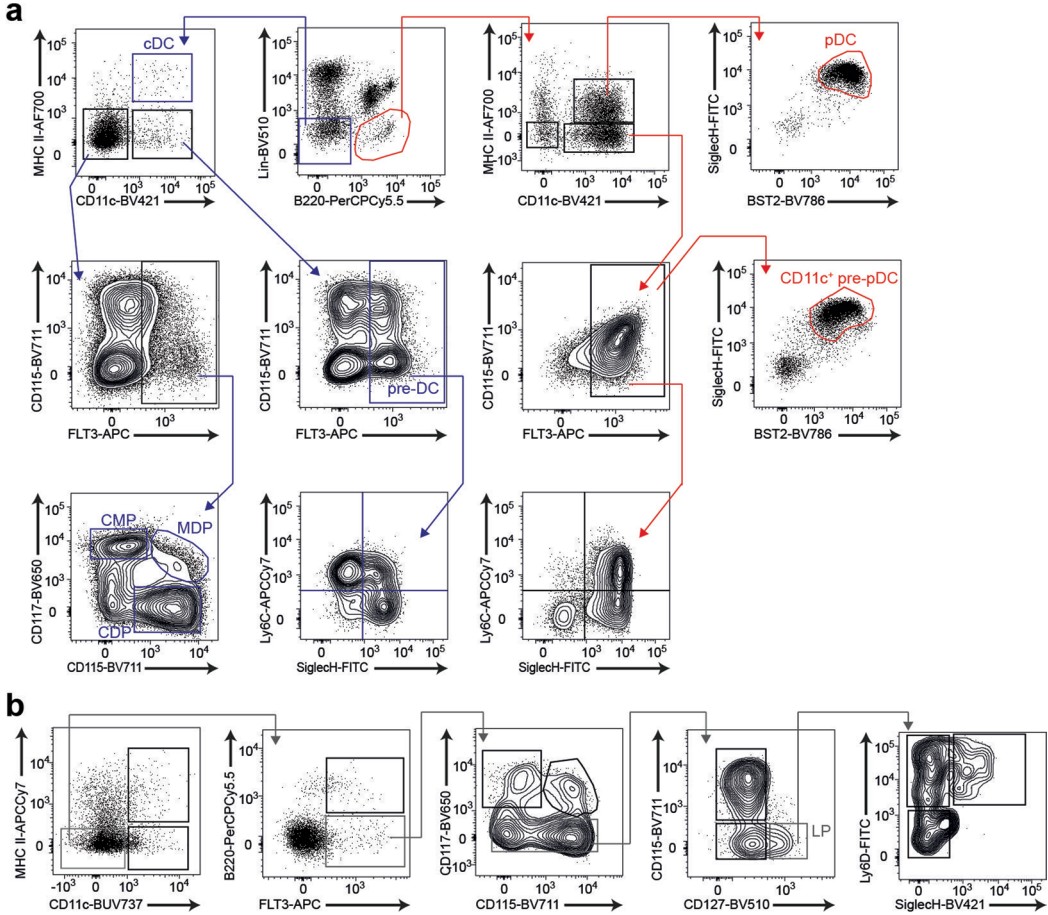

**Extended Data Fig. 2 | Gating strategy for bone marrow precursor analysis of pDC-Tom mice.** Bone marrow cells were isolated from pDC-Tom animals, stained with fluorescently labeled antibodies and analyzed by flow cytometry. The gating strategy used to identify myeloid progenitors (**a**) or lymphoid progenitors (**b**) is depicted. The data shown are from one mouse representative of 5 pDC-Tom animals from 2 independent experiments.

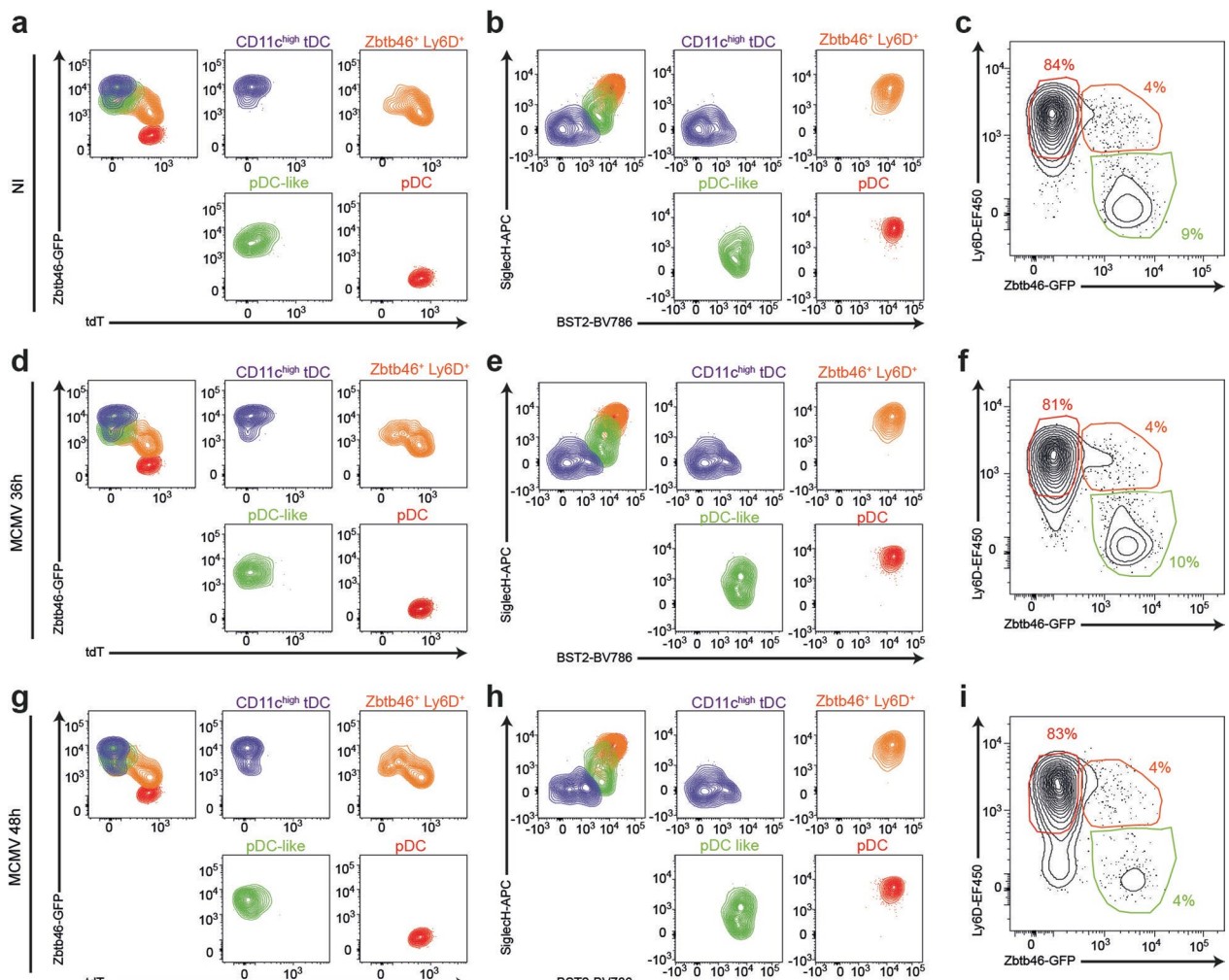

**Extended Data Fig. 3 | The phenotype and the proportions of Zbtb46⁺ Ly6D⁺ cells, CD11c^high tDC and pDC-like cells are relatively stable during MCMV infection.** Splenocytes from ZeST mice were stained with fluorescently labeled antibodies and analyzed by flow cytometry. Representative contour plots from uninfected animals (**a-c**), MCMV-infected mice at 36 h p.i (**d-f**) and from MCMV-infected mice at 48 p.i (**g-i**). The data shown are from one ZeST mouse representative of at least 10 uninfected animals (**a-c**), and for 7 MCMV-infected animals at 36 h p.i (**d-f**) or 8 MCMV-infected animals at 48 h p.i. (**g-i**).

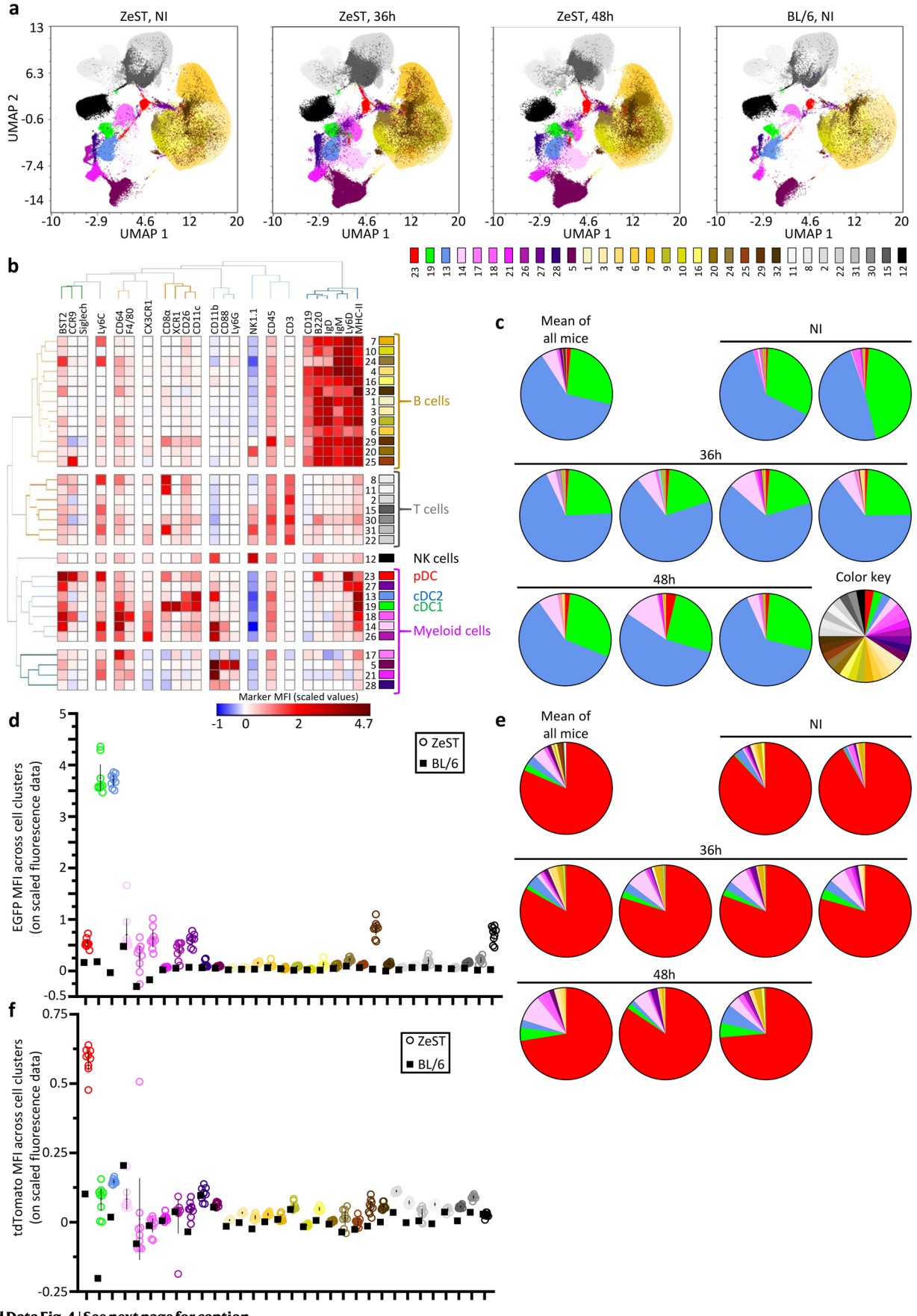

**Extended Data Fig. 4 | See next page for caption.**

**Extended Data Fig. 4 | Spectral flow cytometry-based unsupervised characterization of the expression pattern of tdT and GFP in splenocytes of ZeST mice. a**, Unsupervised dimensional reduction, and cell clustering, for the analysis by spectral flow cytometry of the expression of surface markers on CD45⁺ splenocytes from ZeST mice and one control C57BL/6 mouse, without considering tdT and GFP signals. The UMAP was calculated for all samples together, with downsampling to 200,000 CD45⁺ cells/sample, and is shown for each infection time point with individual mice pooled together (ZeST not infected, n = 2; 36 h, n = 4; 48 h, n = 3; control C57BL/6 mouse, not infected, n = 1). The cluster color code represents related cell types in similar colors (lower right, see also panel **b**). **b**, Cell cluster annotation for cell type identities, based on cell surface marker expression. The mean fluorescence intensity (MFI) of each cluster for each marker was calculated by averaging cluster MFI values across the 10 mice analyzed, and used for hierarchical clustering (heatmap). The corresponding

expression patterns were then used to assign cell cluster to indicated cell types. **c**, Percent of each cluster within GFP⁺ cells, shown first as mean percent of each cluster within GFP⁺ cells across all ZeST mice, and then for each mouse. The color code of the clusters is the same as in (**a, b**) and their ordering in each pie chart shown in the color key (lower right). **d**, GFP MFI (mean on scaled data, Y-axis) for each ZeST mouse (one dot per mouse) and each cluster (X-axis, with same color code and ordering as in (**a, c, d**). For each cluster, the mean ± SEM of the MFI across all ZeST mice is shown as the black lines. For comparison, the autofluorescent signal in the GFP channel in each cell cluster in the C57BL/6 mouse is shown as a black square. **e**, Percent of each cluster within tdT⁺ cells, designed as in (**c**). **f**, tdT MFI for each cluster and each ZeST mouse, designed as in (**d**). The data shown are from one experiment representative of two independent ones.

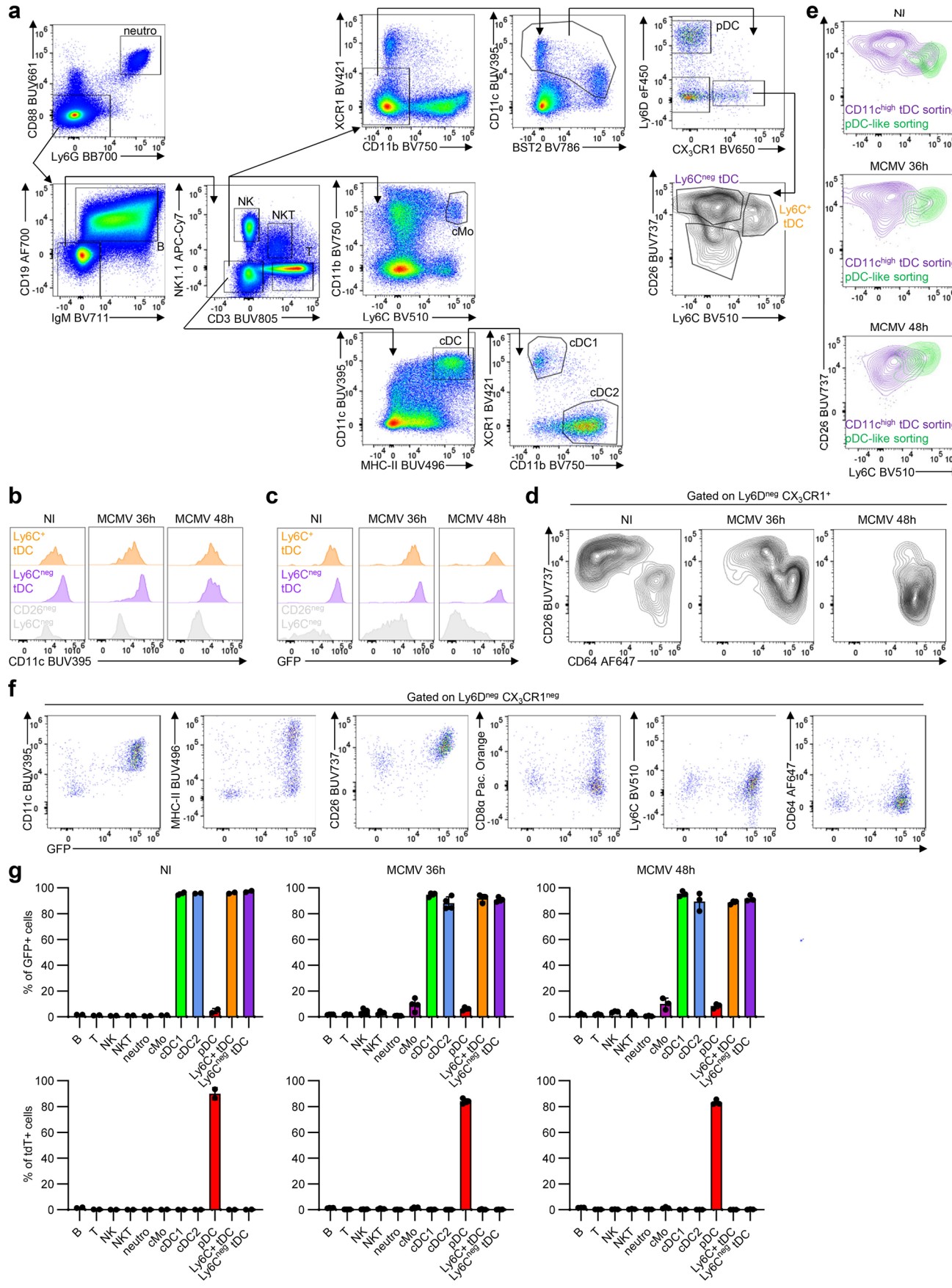

**Extended Data Fig. 5 | See next page for caption.**

**Extended Data Fig. 5 | Spectral flow cytometry-based supervised analysis of the expression of tdTomato and GFP in splenocytes of ZeST mice. a**, Gating strategy followed to identify the indicated lymphoid vs myeloid cell populations. Splenocytes were isolated from uninfected or MCMV-infected ZeST mice. Cells were previously gated as CD45$^+$ after exclusion of dead cells and doublets. **b**, **c**, The expression of CD11c (**b**), GFP (**c**) and of CD26 vs CD64 (**d**) was analyzed on Ly6C$^+$ tDC cells (orange), Ly6C$^{neg}$ tDC (purple) and CD26$^{neg}$ Ly6C$^{neg}$ cells gated as shown in (**a**). **e**, CD11c$^{high}$ tDC (purple) and pDC-like cells (green) were gated from the spectral flow cytometry dataset according to the strategy used with the initial regular flow cytometry dataset (as shown in Fig. 3b), and then analyzed for CD26 vs Ly6C expression, for comparison with the gating strategy used to identify Ly6C$^{neg}$ tDC and Ly6C$^+$ tDC (panel **a**). **f**, Analysis of the expression of GFP vs the indicated cell markers upon gating on Ly6D$^{neg}$ CX$_3$CR1$^{neg}$ cells. **g**, Percentages (mean ± SD) of GFP$^+$ cells (top) or tdT$^+$ cells (bottom) in indicated cell populations gated as shown in (**a**). For (**a**) and (**f**), the analysis shown is representative one ZeST mouse out of 9 tested. For (**b**-**e**) n = 2 for NI, n = 4 for MCMV 36 h and n = 3 for MCMV 48 h.

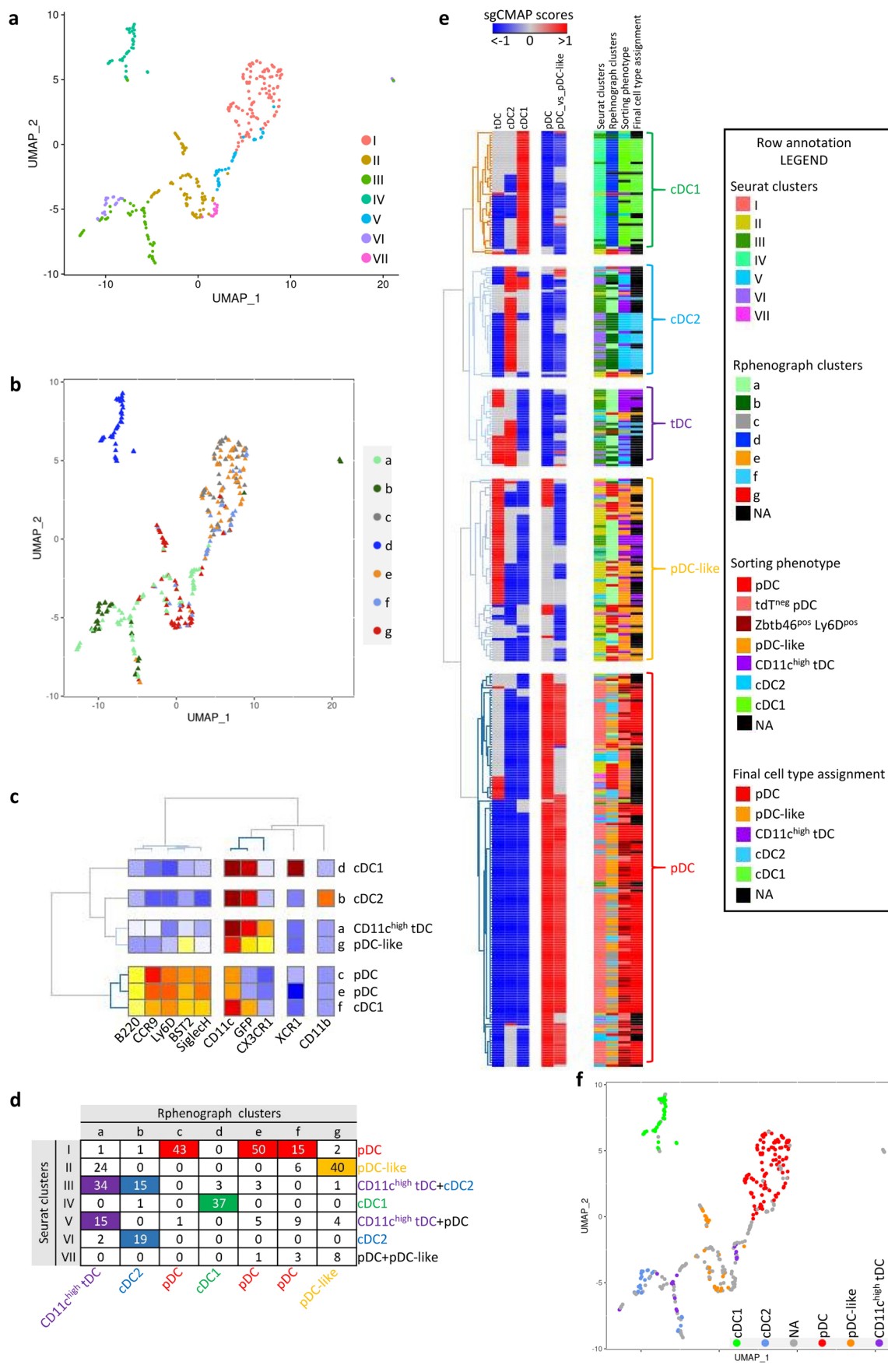

**Extended Data Fig. 6 | See next page for caption.**

**Extended Data Fig. 6 | Analysis of the FB5P-seq data for cells from uninfected mice to generate DC type-specific signatures for helping annotation of the total FB5P-seq dataset. a**, UMAP dimensionality reduction of the gene expression data for DC types isolated from the spleen of 3 uninfected ZeST mice. Cells were index sorted into the 5 DC types studied (see Fig. 3b), and used for single cell RNA sequencing. After quality controls, 343 cells were kept for the analysis. The color code indicates belonging of the cells to the 7 Seurat clusters obtained. **b**, Projection onto the UMAP space of the phenotype of cells based on their belonging to the Rphenograph clusters (color code) obtained upon re-analysis of the fluorescent signals for 10 of the phenotypic markers acquired during index sorting as listed in panel **c**. **c**, Annotation of the Rphenograph clusters for DC type identity based on mean fluorescent intensities per marker and cluster as shown on the heatmap. **d**, Number of cells belonging to the intersections between Seurat (rows) and Rphenograph (column) clusters. Seurat clusters were annotated for cell types based on analysis of their marker genes (see Supplementary Table). **e**, Heatmap showing sgCMAP scores of individual cells (rows) for DC type-specific signatures (columns) generated from published independent RNA-seq datasets. Hierarchical clustering was performed, using the Pearson's minus one metric for signatures (columns), and the Euclidian distance for individual cells (rows), with annotation of individual cells (rows) for i) belonging to Seurat clusters, ii) belonging to Rphenograph clusters, iii) sorting phenotype, and iv) final cell type assignment. 205 cells were assigned a cell type identity based on consistency between their belonging to Seurat and Rphenograph clusters (panel **d**) and their sgCMAP scores. 138 cells were left non-annotated (NA). **f**, Projection onto the UMAP space of the final cell type assignment.

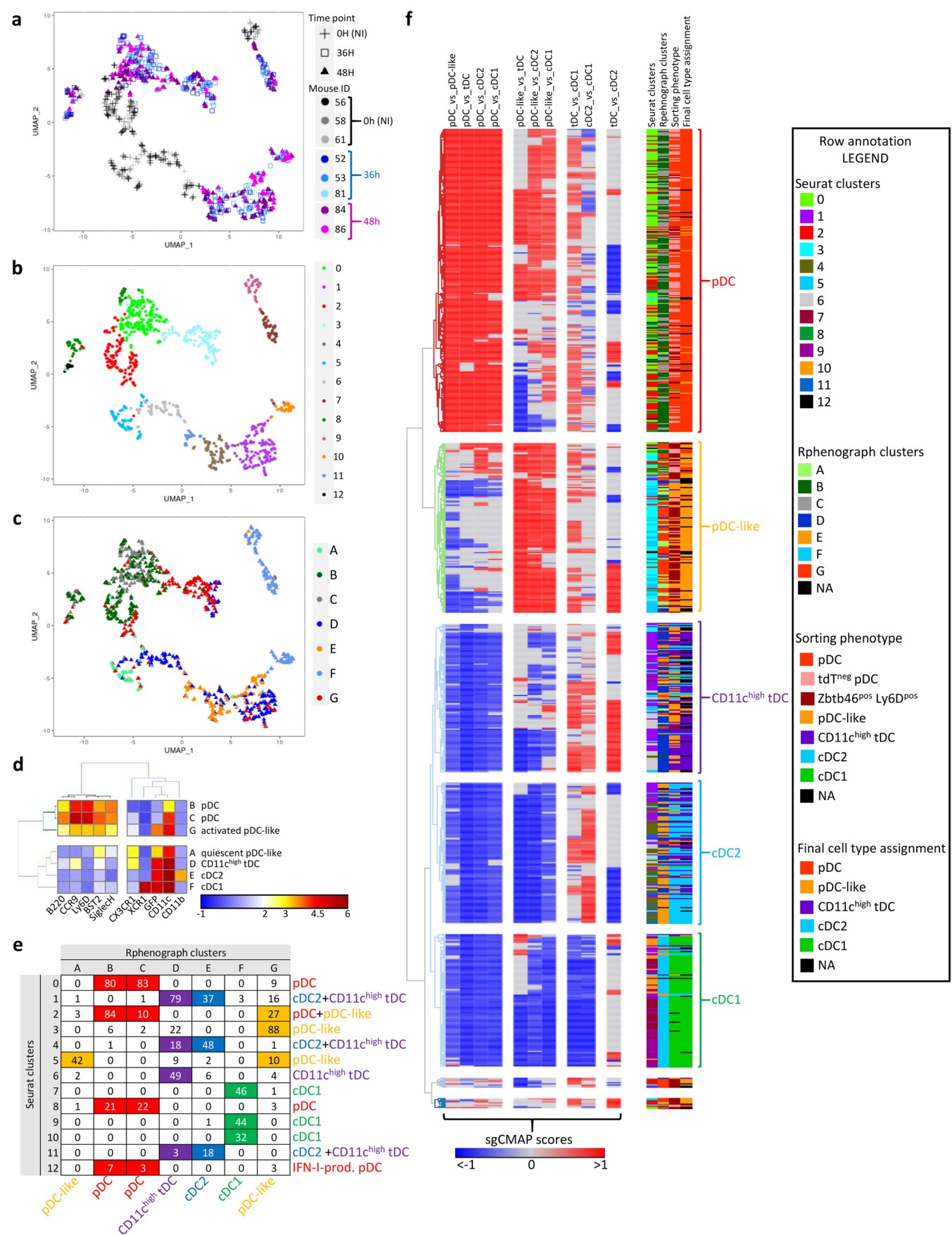

**Extended Data Fig. 7 | See next page for caption.**

**Extended Data Fig. 7 | Analysis of the FB5P-seq data for all cells from uninfected and infected mice. a**, UMAP dimensionality reduction of the gene expression data for DC types isolated from the spleens of eight ZeST mice (3 uninfected, 3 MCMV-infected for 36 h and 2 infected for 48 h, see graphical legend). Cells were index sorted into the 5 DC types studied (see Fig. 3b), and used for single cell RNA sequencing. After quality controls, 951 cells were kept for the analysis. **b**, Projection onto the UMAP space of the 13 Seurat clusters obtained (see graphical legend). **c**, Projection onto the UMAP space of the phenotype of cells based on their belonging to the Rphenograph clusters (color code) obtained upon re-analysis of the fluorescent signals for 10 of the phenotypic markers acquired during index sorting as listed in panel **d**. **d**, Annotation of the Rphenograph clusters for DC type identity based on mean fluorescent intensities per marker and cluster as shown on the heatmap. **e**, Number of cells belonging to the intersections between Seurat (rows) and Rphenograp (column) clusters. Seurat clusters were annotated for cell types based on analysis of their marker genes (see Supplementary Table 2). **f**, Heatmap showing sgCMAP scores of individual cells (rows) for the DC type-specific sgCMAP signatures (columns) generated from the analysis of the data focused on cells from uninfected mice (see Extended Data Fig. 6). Hierarchical clustering was performed, using the Pearson's minus one metric for signatures (columns), and the Euclidian distance for individual cells (rows), with annotation of individual cells (rows) for i) belonging to Seurat clusters, ii) belonging to Rphenograph clusters, iii) sorting phenotype, and iv) final cell type assignment. 851 cells were assigned a cell type identity based on consistency between their belonging to Seurat and Rphenograph clusters (panel **e**), which was well corroborated by the sgCMAP scores. 100 cells were left non-annotated (NA).

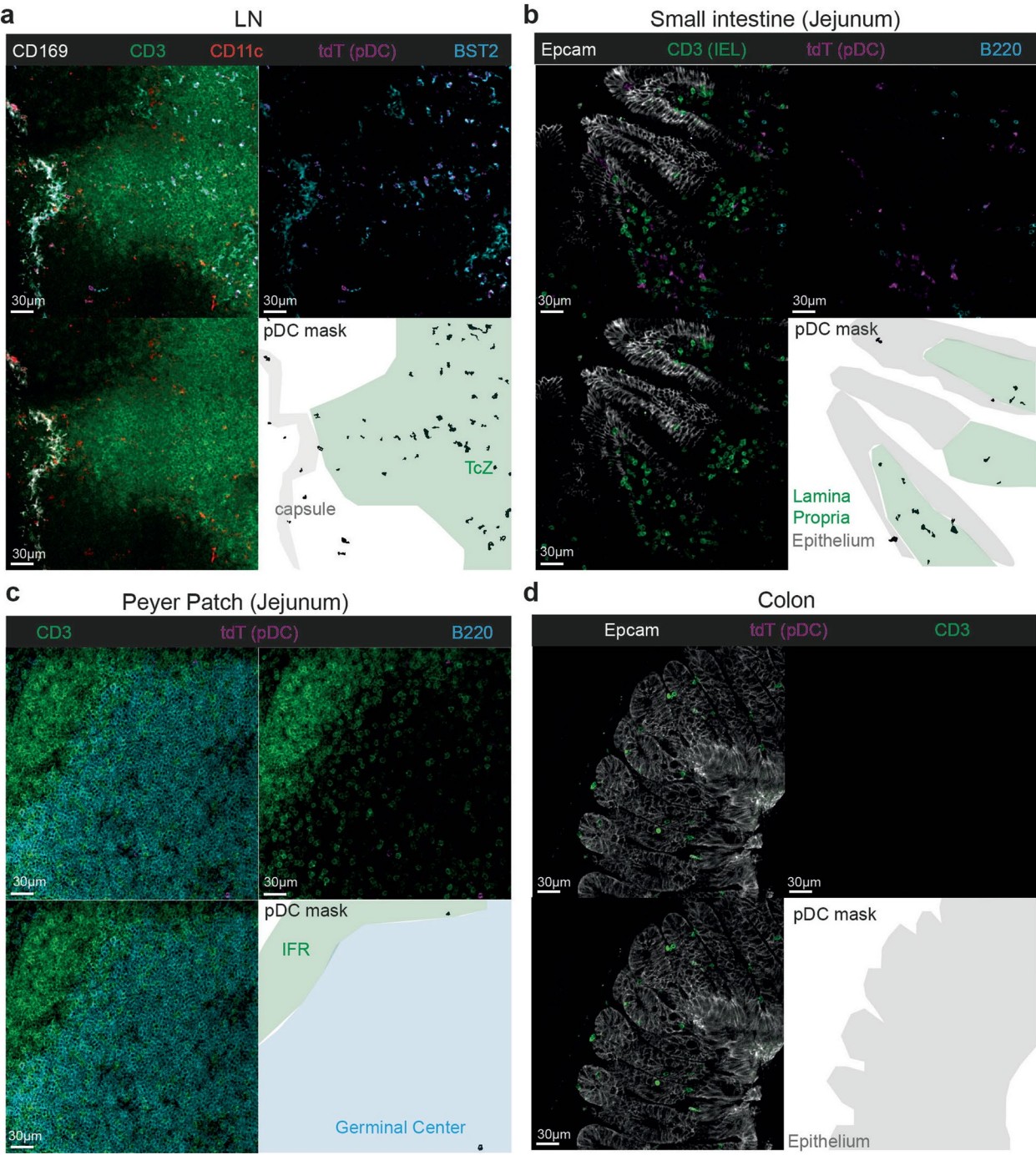

**Extended Data Fig. 8 | pDC-Tom mice allow to determine pDC micro-anatomical location in the different organs.** 20μm organ cryosections from uninfected pDC-Tom mice were stained with anti-tdT (magenta), anti-CD169 (white), anti-CD3 (green), anti-CD11c (red) and anti-BST2 (cyan) antibodies (**a**) or with anti-tdT (magenta), anti-Epcam (white), anti-CD3 (green) and anti-B220 (cyan) antibodies (**b-d**). Representative images are shown for LN of 3 mice (**a**), small intestine of 4 mice (**b**, **c**) and colon of 3 mice (**d**).

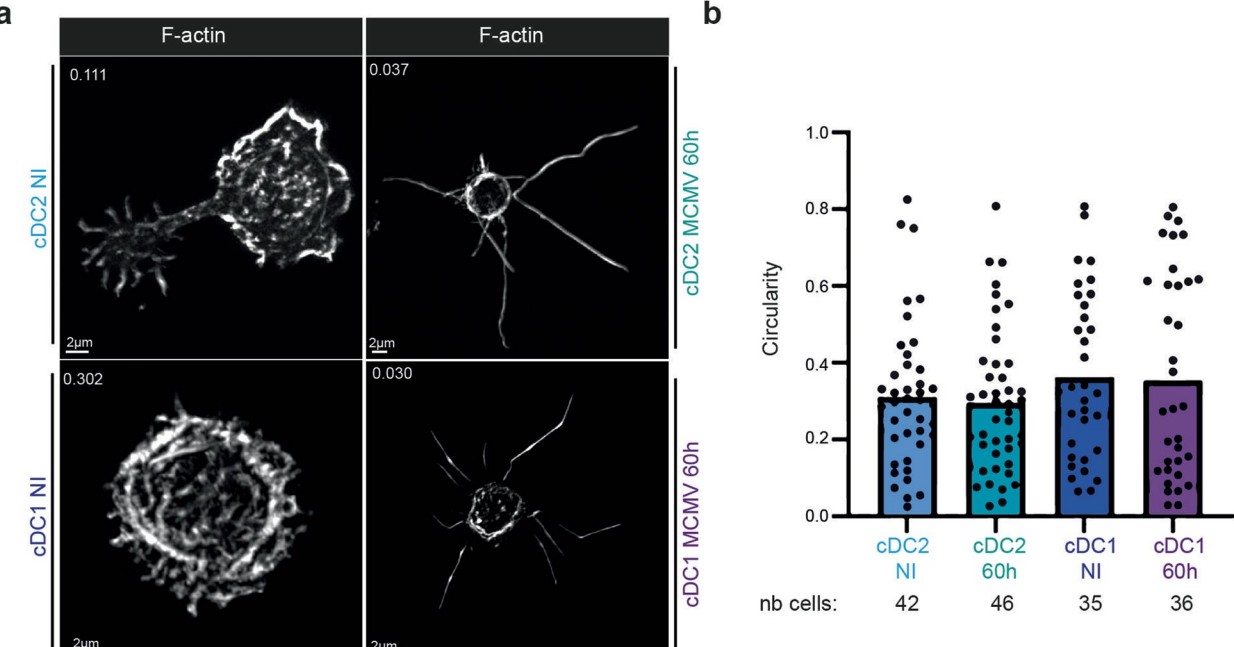

**Extended Data Fig. 9 | Circularity indices for cDC1 and cDC2 upon MCMV infection. a**, **b**, Quantitative and unbiased assessment of the cellular morphology of cDC1 and cDC2 isolated from 60 h MCMV-infected SCRIPT mice, as compared to cDC1 and cDC2 from uninfected mice. **a**, One representative confocal microscopy image of each DC type is shown. **b**, The distribution of the circularity indices for individual cells across DC types is shown as dots, with the overlaid color bars showing the mean circularity indices of each DC type. The data shown are from 2 independent experiments, each performed with one mouse, with 35 to 46 individual cells analyzed for each DC type as indicated below the graph. The data are shown as mean ± SEM.

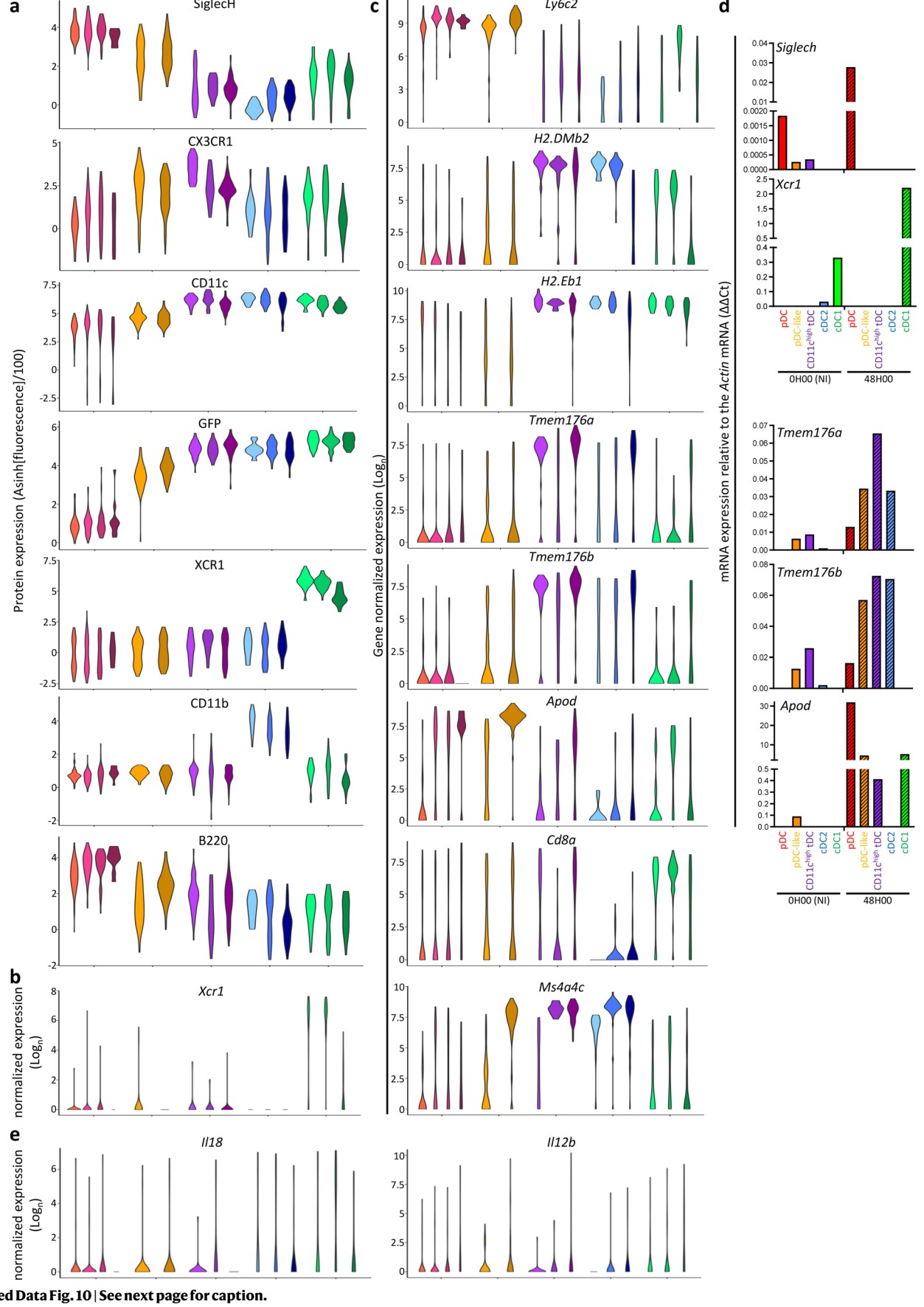

**Extended Data Fig. 10 | See next page for caption.**

**Extended Data Fig. 10 | Expression of selected phenotypic markers and genes across DC types and activation states. a**, Violin plots showing the expression of phenotypic markers across DC types and activation states. **b**, Violin plot showing expression of the *Xcr1* gene across DC types and activation states. **c**, Violin plots showing mRNA expression levels of selected genes across DC types and activation states. **d**, Mini-bulk qRT-PCR analysis of the expression level of control (*Siglech* and *Xcr1*) and new candidate marker genes (*Tmem176a*, *Tmem176b* and *Apod*) across DC types and activation states. **e**, Violin plots showing mRNA expression levels of *Il18* and *Il12b* across DC types and activation states.

|---|---|

# Reporting Summary

## Statistics

For all statistical analyses, confirm that the following items are present in the figure legend, table legend, main text, or Methods section.

| n/a | Confirmed | |
|---|---|---|
| ☐ | ☒ | The exact sample size (*n*) for each experimental group/condition, given as a discrete number and unit of measurement |
| ☐ | ☒ | A statement on whether measurements were taken from distinct samples or whether the same sample was measured repeatedly |
| ☐ | ☒ | The statistical test(s) used AND whether they are one- or two-sided *Only common tests should be described solely by name; describe more complex techniques in the Methods section.* |
| ☒ | ☐ | A description of all covariates tested |
| ☒ | ☐ | A description of any assumptions or corrections, such as tests of normality and adjustment for multiple comparisons |
| ☐ | ☒ | A full description of the statistical parameters including central tendency (e.g. means) or other basic estimates (e.g. regression coefficient) AND variation (e.g. standard deviation) or associated estimates of uncertainty (e.g. confidence intervals) |
| ☐ | ☒ | For null hypothesis testing, the test statistic (e.g. *F*, *t*, *r*) with confidence intervals, effect sizes, degrees of freedom and *P* value noted *Give P values as exact values whenever suitable.* |
| ☒ | ☐ | For Bayesian analysis, information on the choice of priors and Markov chain Monte Carlo settings |
| ☒ | ☐ | For hierarchical and complex designs, identification of the appropriate level for tests and full reporting of outcomes |
| ☐ | ☒ | Estimates of effect sizes (e.g. Cohen's *d*, Pearson's *r*), indicating how they were calculated |

*Our web collection on statistics for biologists contains articles on many of the points above.*

## Software and code

Policy information about availability of computer code

| Data collection | The RNA Seq data (FB5P technology) were generated using an Illumina NextSeq2000 platform, with 100-cycles P2 flow cells. |
|---|---|
| Data analysis | Conventional flow cytometry data were acquired using BD Diva v9.0. Spectral flow cytometry data were acquired with SpectroFlo 3.0.1. Flow Cytometry data were analyzed using FlowJo v10.8.1 (Treestar) and OMIQ (app.OMIQ.ai). GraphPad Prism (8.1.2) was used for graphical and statistical analyses. Images were processed and analyzed using ImageJ (1.52p). FB5P sequencing data were aligned and mapped to reference genome using STAR (v2.5.3a) and HTSeqCount (v0.9.1) and processed to generate a single-cell UMI counts matrix (Attaf et al. Front Immunol 2020 11;216). The counts matrix was loaded to R (v.4.0.3), and Seurat (v3.2.0) was used for downstream analyses. |

For manuscripts utilizing custom algorithms or software that are central to the research but not yet described in published literature, software must be made available to editors and reviewers. We strongly encourage code deposition in a community repository (e.g. GitHub). See the Nature Portfolio guidelines for submitting code & software for further information.

## Data

Policy information about availability of data

All manuscripts must include a data availability statement. This statement should provide the following information, where applicable:

- Accession codes, unique identifiers, or web links for publicly available datasets
- A description of any restrictions on data availability
- For clinical datasets or third party data, please ensure that the statement adheres to our policy

The datasets generated during and/or analysed during the current study are available in the GEO repository, under accession numbers GSE76132 (PMID: 26903243) and GSE196720 (this study). All other data generated or analysed during this study are included in this published article (and its supplementary information files).

# Field-specific reporting

Please select the one below that is the best fit for your research. If you are not sure, read the appropriate sections before making your selection.

☒ Life sciences ☐ Behavioural & social sciences ☐ Ecological, evolutionary & environmental sciences

For a reference copy of the document with all sections, see nature.com/documents/nr-reporting-summary-flat.pdf

# Life sciences study design

All studies must disclose on these points even when the disclosure is negative.

| | |
|---|---|
| Sample size | No statistical methods were used to pre-determine sample sizes but our sample sizes are similar to those reported in previous publications ( Tomasello et al. EMBO J 2018, PMID: 30131424; Abbas et al. Nat Immunol 2020, PMID: 32690951). |
| Data exclusions | No animals or data points were excluded from the analyses |
| Replication | All experiments except scRNAseq data were reproduced at least twice. The number of experiments have been specified in each legend of figure. scRNAseq data were generated from 3 uninfected mice, 3 MCMV-infected mice for 36h and 2 infected mice for 48h, with 3 independent sorts performed with 2 or 3 animals each time (sorts for mice #56, 58 on 2020/11/03, for #52, 53 and 61 on 2020/12/17, for #81, 84 and 86 on 2021/02/11); sorting plates were frozen until all samples had been collected, and all libraries generated and sequenced simultaneously to avoid eventual batch effects. All attempts at replication were successful. |
| Randomization | No randomization was performed in this study. It was not necessary since mice were matched in age and gender between experimental groups and comparisons were made across cell types in the same mice or between infection time points for the same mouse strain. Hence, there were no confounding covariates in our analyses. |
| Blinding | No blinding was performed in this study. As we performed quantitative measurements with the use of computational analyses, the use of blinding was not required in the present study. |

# Reporting for specific materials, systems and methods

We require information from authors about some types of materials, experimental systems and methods used in many studies. Here, indicate whether each material, system or method listed is relevant to your study. If you are not sure if a list item applies to your research, read the appropriate section before selecting a response.

### Materials & experimental systems

| n/a | Involved in the study |
|---|---|
| ☐ | ☒ Antibodies |
| ☒ | ☐ Eukaryotic cell lines |
| ☒ | ☐ Palaeontology and archaeology |
| ☐ | ☒ Animals and other organisms |
| ☒ | ☐ Human research participants |
| ☒ | ☐ Clinical data |
| ☒ | ☐ Dual use research of concern |

### Methods

| n/a | Involved in the study |
|---|---|
| ☒ | ☐ ChIP-seq |
| ☐ | ☒ Flow cytometry |
| ☒ | ☐ MRI-based neuroimaging |

## Antibodies

| | |
|---|---|
| Antibodies used | anti-B220 Alexa Fluor 594 clone RA3-6B2 biolegend cat# 103254 ; RRID:AB_2563229 1/400<br>anti-B220 Alexa Fluor 647 clone RA3-6B2 biolegend cat#103229; RR1D:AB_492875 1/400<br>anti-B220 APC-Fire810 clone RA3-6B2 biolegend cat# 103278; RRID: AB_2860603 1/400<br>anti-B220 PE-cy7 clone RA3-6B2 biolegend cat# 103222 ; RRID:AB_313005 1/400 |

anti-B220 PerCP-Cy5.5 clone RA3-6B2 biolegend cat# 103236 ; RRID:AB_893354 1/200
anti-APC biotin clone APC003 biolegend cat# 408004 ; RRID:AB_345360 1/200
anti-BST2 BV650 clone 927 biolegend cat# 127019 ; RRID:AB_2562477 1/200
anti-BST2 BV786 clone 927 BD biosciences cat# 747603 ; RRID:AB_2744171 1/200
anti-BST2 biotin clone eBio927 Thermofisher cat# 13-3172-82 ; RRID:AB_763415 1/200
anti-BST2 purified clone 120G8 Dendritics cat#DDX0390; RRID:AB_DDX0390 1/100
anti-CCR9 PE-Cy7 clone CW-1.2 biolegend cat# 128712 ; RRID:AB_10933082 1/400
anti-CD115 BV421 clone AFS98 biolegend cat# 135513 ; RRID:AB_2562667 1/200
anti-CD115 BV711 clone AFS98 biolegend cat# 135515 ; RRID:AB_2562679 1/200
anti-CD117 BV650 clone 2B8 BD biosciences cat# 563399 ; RRID:AB_2738183 1/800
anti-CD127 BV510 clone SB/199 BD biosciences cat#563353 ; RRID:AB_2738153 1/100
anti-CD11b BUV395 clone M1/70 BD biosciences cat# 565976 ; RRID:AB_27382 1/400
anti-CD11b BV750 clone M1/70 biolegend cat# 101267;RRID:AB_2810328 1/600
anti-CD11c BUV395 clone N418 BD biosciences cat#744180;RRID:AB_2742045 1/200
anti-CD11c BUV737 clone N418 BD biosciences cat# 749039 ; RRID:AB_2873433 1/400
anti-CD11c BV421 clone HL3 BD biosciences cat# 562782 ; RRID:AB_2737789 1/400
anti-CD11c BV785 clone N418 biolegend cat# 117336 ; RRID:AB_2565268 1/200
anti-CD11c purified clone N418 biolegend cat# 117302  ; RRID:AB_313771 1/100
anti-CD135 APC clone A2F10.1 BD biosciences cat# 560718 ; RRID:AB_1727425 1/200
anti-CD169 biotin clone MOMA-1 abcam cat# ab51814 1/500
anti-CD169 Alexa Fluor 647 clone 3D6.112 biolegend cat# 142408 ; RRID:AB_2563621 1/500
anti-CD19 clone 1D3 Alexa Fluor 700 BD biosciences cat# 557958 ; RRID:AB_396958 1/200
anti-CD19 clone 1D3 BV510 BD biosciences cat# 562956; RRID:AB_2737915 1/300
anti-CD26 BUV737 clone H194-112 BD biosciences cat# 741729; RRID: AB_2871099 1/200
anti-CD3 purified  clone 145-2C11 BD biosciences cat# 550275 ; RRID:AB_393572 1/300
anti-CD3 Alexa Fluor 700 clone eBio500A2 eBioscience  cat# 56-0033-82 ; RRID:AB_837094 1/200
anti-CD3 Alexa Fluor 488 clone 17A2 biolegend cat# 100210; RRID:AB_389301 1/300
anti-CD3 BUV805 clone 17A2 BD biosciences cat#741982; RRID; AB_2871285 1/200
anti-CD3 BV510 clone 17A2 BD biosciences cat#740147 ; RRID:AB_2739902 1/300
anti-CD3 EF450 clone 17A2 Thermofisher cat# 48-0032-82 ; RRID:AB_1272193 1/300
anti-CD45 PerCP clone 30F11 biolegend cat# 103130; RRID:AB_893343 1/200
anti-CD45.2 BUV395 clone 104 BD biosciences cat# 564616 ; RRID:AB_2738867 1/300
anti-CD45.2 BUV737 clone 104 BD biosciences cat# 612778 ; RRID:AB_2870107 1/300
anti-CD64 BV711 Alexa Fluor 647 clone X54-5/7.1 biolegend cat# 139322; RRID:AB_2566560 1/200
anti-CD8a APC-cy7 clone 53-6.7 biolegend cat# 100714 ; RRID:AB_312753 1/200
anti-CD8a Pacific Orange clone 5H10  eBioscience cat# MCD0830; RRID:AB_10376311 1/200
anti-CD88 BUV661 clone 20/70 BD biosciences cat#750080; RRID:AB_2874295 1/400
anti-CX3CR1 BV650 clone SA011F11 biolegend cat#149033; RRID:AB_2565999 1/200
anti-CX3CR1 BV711 clone SA011F11  biolegend cat# 149031 ; RRID:AB_2565939 1/100
anti-EpCAM EF450 clone G8.8 Thermofisher cat# 48-5791-82 ; RRID:AB_10717090 1/200
anti-F4/80 BB700 clone T45-2342 BD biosciences cat#746070; RRID:AB_2743450 1/400
anti-F4/80 biotin clone BM8 biolegend cat# 123106 ; RRID:AB_893501 1/200
anti-GFP Alexa Fluor 488 polyclonal Thermofisher cat# A-21311 ; RRID:AB_221477 1/200
anti-I-A/I-E BUV496 clone M5/114.15.2 BD biosciences cat#750281; RRID:AB_2874472 1/400
anti-IE1 purified Capri clone IE1.01 cat# HR-MCMV-12 1/1000
anti-IgD Spark NIR 685 clone 11-26c.2a biolegend cat# 405750; RRID:AB_2888693 1/1000
anti-IgM BV711 clone RMM-1 biolegend cat# 406539; RRID:AB_2814386 1/200
anti-Ly6C APC-Cy7 clone AL-21 BD Biosciences cat#560596 ; RRID:AB_1727555 1/400
anti-Ly6C BV510 clone HK1.4 biolegend cat# 128033;RRID:AB_2562351 1/800
anti-Ly6D EF450 clone 49-H4 thermofisher cat# 48-5974-80 ; RRID:AB_2574089 1/400
anti-Ly6D FITC clone 49-H4 biolegend cat# 138606 ; RRID:AB_11203888 1/400
anti-Ly6G BB700 clone 1A8 BD Biosciences cat#566453; RRID:AB_2739730 1/400
anti-Ly6G BV510 clone 1A8 BD Biosciences cat#740157 ; RRID: AB_2739910 1/300
anti-Ly6G clone 1A8Alexa Fluor 700 biolegend cat# 561236 ; RRID:AB_10611860 1/200
anti-mcherry purified polyclonal Rockland cat# 600-401-P16 1/500
anti-NK1.1 Alexa Fluor 700 clone PK136 BD biosciences cat# 553162 ; RRID:AB_394674 1/200
anti-NK1.1 APC-Cy7 clone PK136 biolegend cat# 156505; RRID:AB_2876525 1/200
anti-NK1.1 BV510 clone PK136 biolegend cat# 108738 ; RRID:AB_2562217 1/300
anti-RFP purified polyclonal Rockland cat# 600-401-379 1/500
anti-SiglecH BV421 clone 440c BD biosciences cat# 566581 ; RRID:AB_2739747 1/400
anti-SiglecH FITC clone 551 biolegend cat# 129604  ; RRID:AB_1227761 1/200
anti SiglecH PerCP-cy5.5 clone 551 biolegend cat# 129614 ; RRID:AB_10643995 1/200
anti SiglecH PerCP-eFluor710 clone eBio440c eBiosciences cat#46-0333-82; RRID:AB_1834443 1/100
anti-SiglecH APC clone 551 biolegend cat#129612 ; RRID:AB_10641134 1/400
anti-XCR1 BV421 clone ZET biolegend cat#148216; RRID:AB_2565230 1/400
anti-XCR1 BV650 clone ZET biolegend cat#148220  ; RRID:AB_2566410 1/800
streptavidin APC BD biosciences cat# 349024 1/400
streptavidin EF450 Thermofisher cat# 129614 1/400
streptavidin Alexa Fluor 633 Thermofisher cat# S-21375 1/400
Donkey anti-rabbit IgG (H+L) Alexa Fluor 488  poyclonal Jackson Immunoresearch cat# 711-545-152 1/200
Donkey anti-rabbit IgG (H+L) Alexa Fluor 594 poyclonal Jackson Immunoresearch cat# 127-585-152 1/500
Donkey anti-rabbit IgG (H+L) Alexa Fluor 647 poyclonal Thermofisher cat# A-31573 1/500
Goat anti-Hamster IgG (H+L) Cyanine 3 poyclonal Jackson Immunoresearch cat# 127-165-160 1/500
Goat anti-Hamster IgG (H+L) Rhodamine Red X poyclonal Jackson Immunoresearch cat# 127-295-159 1/500
Goat anti-Hamster IgG (H+L) Alexa Fluor 594 poyclonal Thermofisher cat# A-21113 1/500
Goat anti-Mouse IgG2a Alexa Fluor 633 polyclonal Thermofisher cat# A-21136 1/2500

| | Goat anti-Rat Alexa Fluor 546 polyclonal Thermofisher cat# A-11081 1/350 |
|---|---|
| Validation | All antibodies were validated by the manufacturer, for the species and the specific application for which they were used in the study (flow cytometry, immunohistofluorescence). Each antibody has been previously titrated for optimal performance in the used assay. |

# Animals and other organisms

Policy information about studies involving animals; ARRIVE guidelines recommended for reporting animal research

| Laboratory animals | C57BL/6J M/F 6-16 weeks<br>Siglech-iCre;Rosa26-LSL-RFP M/F 8-16 weeks<br>pDC-Tom (Siglech-iCre;Pacsin1-LSL-tdT) M/F 8-16 weeks<br>SCRIPT (Siglech-iCre;Pacsin1-LSL-tdT;Ifnb1-eYFP) M/F8-16 weeks<br>ZeST (Siglech-iCre;Pacsin1-LSL-tdT;Zbtb46-eGFP) M/F 8-16 weeks<br>Mice were housed under 12h dark/12h light cycle, with a temperature range of 20-22°C and a humidity range of 40-70%. |
|---|---|
| Wild animals | This study did not use wild animals |
| Field-collected samples | This study did not involve field-collected samples |
| Ethics oversight | All animal experiments were performed in accordance with national and international laws for laboratory animal welfare and experimentation (EEC Council Directive 2010/63/EU, September 2010). Protocols were approved by the Marseille Ethical Committee for Animal Experimentation (registered by the Comité National de Réflexion Ethique sur l'Expérimentation Animale under no. 14; APAFIS#1212-2015072117438525 v5 and APAFIS#21626-2019072606014177 v4). |

Note that full information on the approval of the study protocol must also be provided in the manuscript.

# Flow Cytometry

## Plots

Confirm that:

☒ The axis labels state the marker and fluorochrome used (e.g. CD4-FITC).

☒ The axis scales are clearly visible. Include numbers along axes only for bottom left plot of group (a 'group' is an analysis of identical markers).

☒ All plots are contour plots with outliers or pseudocolor plots.

☒ A numerical value for number of cells or percentage (with statistics) is provided.

## Methodology

| Sample preparation | Samples originated from mouse organs.<br>For flow cytometry, spleens or lymph nodes were harvested and submitted to enzymatic digestion for 25 minutes at 37°C with Collagenase IV (Wortington biochemicals) and DNAse I (Roche Diagnostics). Organs were then mechanically digested and passed over 100μm cell strainers (Corning). Red blood cells were then lysed by using RBC lysis buffer (Life Technologies) for spleen and bone marrow cell preparation. Livers were harvested, minced and submitted to enzymatic digestion, as for the spleen. Liver pieces were then crushed and cell suspension obtained was washed 2 times with PBS 1x, before performing a 80/40 Percoll gradient. Cells isolated from the middle ring of the gradient were washed once with PBS 1x, then used for flow cytometry. Small intestines were harvested, opened longitudinally, then cut into 1mm pieces. Pieces were washed entensively with PBS 1x, then incubated 3 times at 37°C upon shaking (200 rpm) with PBS 1X containing 2% Fetal Calf Serum (FCS) and 5mM ethylenediamine tetraacetic acid (EDTA). At the end of each incubation, supernatants were collected and centrifuged. Pelleted cells, mainly IntraEpithelial Cell (IEL), from the three incubations were pooled together and submitted to a 67/44 Percoll gradient. Cells isolated from the middle ring of the gradient were washed once with PBS 1x, then used for flow cytometry.<br>For microscopy, organs were fixed with Antigen Fix (Diapath) for 2 hours for the small intestine, colon, lymph nodes or 4 hours for the spleen at 4°C, and then washed several times in PB (0,025 M NaH2PO4 and 0,1 M Na2HPO4). Organs were then immerged in a solution of 30% sucrose O/N at 4°C. Organs were then embedded in OCT (Sakura), snap frozen and stored at -80°C. |
|---|---|
| Instrument | BD Fortessa X-20, BD ARIA 3, Cytek Aurora |
| Software | BD DIVAv9.0, FlowJo v10.8.1, SpectroFlo 3.0.1, OMIQ |
| Cell population abundance | The abundance of the population was low since it was single cell sorting (scRNAseq) |
| Gating strategy | The gating strategies have been provided in the Extended Data Figures.<br>Figure 1= CD64-neg/CD45-pos/live, then for neutrophils: Ly6G-pos; for B cells: CD19-pos/Ly6G-neg/CD3-neg; for NK and T cells: Ly6G-neg and then CD3-neg/NK1.1-pos versus CD3-pos/NK1.1-neg, respectively; for pDC: Ly6G-neg/CD3-neg/NK1.1-neg/CD11c-low/BST2-pos, and for cDC: Ly6G-neg/CD3-neg/NK1.1-neg/CD11c-pos/Bst2-neg and CD8a-pos (for cDC1) vs CD11b-pos (for cDC2). |

Figure 2a= for CMP, CDP and MDP CD11c-neg/MHCII-neg/FLT3-pos followed by CD117-high/CD115-neg vs CD117-pos/ CD115-pos vs CD117-neg/CD115-pos, respectively; for preDC: CD11c-pos/MHCII-neg/FLT3-pos/CD115-neg-to-pos followed by segregation according to Ly6C vs SiglecH expression; for pDC: Lin-neg/B220-pos/CD11c-pos/MHC-II-pos/SiglecH-pos/Bst2-pos; for CD11c+ pre-pDC: Lin-neg/B220-pos/CD11c-pos/MHC-II-neg/FLT3-pos/CD115-neg-to-pos/SiglecH-pos/Bst2-pos.

Figure 2b= for LP: MHCII-neg/CD11c-neg/FLT3-pos/B220-neg/CD117-neg/CD115-neg/CD127-pos, followed by segregation according to Ly6D vs SiglecH expression.

Extended Figure 5a= CD88-neg/Ly6G-neg, then for B cells: CD19pos IgMpos; for NK, NKT and T cells: CD19-neg/IgM-neg followed by segregation according to CD3 vs NK1.1 expression; for pDC vs tDC: CD3-neg/NK1.1-neg followed by XCR1-neg/ CD11b-neg/(CD11c-pos or BST2-pos) and then Ly6D-pos for pDC versus Ly6D-neg/ CX3CR1-pos/CD26-pos for tDC further split according to Ly6C expression; for cMo: CD3-neg/NK1.1-neg/CD11b-pos/Ly6C-pos; for cDC: CD3-neg/NK1.1-neg/CD11c-pos/ MHC-II-pos followed by XCR1-pos (for cDC1) vs CD11b-pos (for cDC2).

☒ Tick this box to confirm that a figure exemplifying the gating strategy is provided in the Supplementary Information.

