## [Peer Review File · Nature Immunology]

Peer Review Information

Journal: Nature Immunology

Manuscript Title: Novel mouse models based on intersectional genetics to identify and characterize plasmacytoid dendritic cells

Corresponding author name(s): Dr. Marc Dalod, Michael Valente, Elena Tomasello

Editorial Notes:

Redactions – unpublished data	Parts of this Peer Review File have been redacted as indicated to maintain the confidentiality of unpublished data.
Redactions – confidential patient information	Parts of this Peer Review File have been redacted as indicated to maintain patient confidentiality.
Redactions – published data	Parts of this Peer Review File have been redacted as indicated to remove third-party material.
Redactions – reviewer opt-out	Parts of this Peer Review File have been redacted as indicated as we could not obtain permission to publish the reports of reviewer no. XX .
Reviewer comments in marked-up manuscript	In their review of the [first/second/third/...] version of this manuscript, reviewer no. XX added their comments to the manuscript file. These comments, excluding minor textual revisions, have been copied into this Peer Review File.

Reviewer Comments & Decisions:

Decision Letter, initial version:

16th Mar 2022

Dear Dr Dalod,

Your Resource manuscript entitled, "Novel mouse models based on intersectional genetics to identify and characterize plasmacytoid dendritic cells" has now been seen by 3 referees. We received fairly consistent reports from the referees, noting that although the new mouse model for specifically identifying pDCs is appreciated, all three referees noted that they were disappointed that not much new insight into pDC biology was provided using this genetic model, and referee #2 noted that the scRNA-seq analysis used a low number of cells and the findings were not further validated, diminishing the value of the study as a Resource. However, upon discussion with the other editors, we thought the study might still be of benefit to the immunology community as a Technical Report to introduce the new mouse model, pending revision to address the referees' comments below.

We hope you will find the referees' comments useful as you decide how to proceed. If you wish to submit a substantially revised manuscript, please bear in mind that we will be reluctant to approach the referees again in the absence of major revisions.

If you choose to revise your manuscript taking into account all reviewer and editor comments, please highlight all changes in the manuscript text file in Microsoft Word format.

* If you have not done so already please begin to revise your manuscript so that it conforms to our Resource format instructions at <http://www.nature.com/ni/authors/index.html>. Refer also to any guidelines provided in this letter.

The Reporting Summary can be found here:

<https://mts-ni.nature.com/cgi-bin/main.plex?el=A7E4Yb3A7HyA5J2A9ftdy40YW1qALA35TVVY8PNTgQZ>

Note: This URL links to your confidential home page and associated information about manuscripts you may have submitted, or that you are reviewing for us. If you wish to forward this email to co-

authors, please delete the link to your homepage.

If you wish to submit a suitably revised manuscript we would hope to receive it within 6 months. If you cannot send it within this time, please let us know. We will be happy to consider your revision so long as nothing similar has been accepted for publication at Nature Immunology or published elsewhere.

Nature Immunology is committed to improving transparency in authorship. As part of our efforts in this direction, we are now requesting that all authors identified as 'corresponding author' on published papers create and link their Open Researcher and Contributor Identifier (ORCID) with their account on the Manuscript Tracking System (MTS), prior to acceptance. ORCID helps the scientific community achieve unambiguous attribution of all scholarly contributions. You can create and link your ORCID from the home page of the MTS by clicking on 'Modify my Springer Nature account'. For more information please visit www.springernature.com/orcid.

Thank you for the opportunity to review your work.

Kind regards,

Laurie

Laurie A. Dempsey, Ph.D.
Senior Editor
Nature Immunology
l.dempsey@us.nature.com
ORCID: 0000-0002-3304-796X

Referee expertise:

Referee #1: dendritic cell development

Referee #2: dendritic cells

Referee #3: plasmacytoid DCs

Reviewers' Comments:

Reviewer #1:

Remarks to the Author:

In study by Valente et. al the authors generate/take advantage of a new mouse model SiglecH cre-Pacsin1 LSL-tdT to understand and dissect the dynamics of pDCs and include in their detailed analysis the recently described pDC-Like and tDCs. Given the unspecific expression of markers across these subsets and cDCs, the functional properties and the precise interrelation of pDCs with these other subsets remained so far elusive. Combining two pDC markers SiglecH and Pacsin1 with the report lines Zbtb46-gfp and IFN-I YFP the authors to fully characterize pDCs, pDC-like cells and tDCs at steady state and during MCMV infection.

This newly generated mouse model is certainly a significant advance as SiglecH-gfp mice were shown to label more cells than pDC. Also, the use of Tomato allows for intercross with several lines that express GFP or YFP improving the characterization of pDC like and tDCs. The flow panels are very clean and the data presented is solid and focused on the characterization of the subsets rather than engaging in the current debate of pDC development.

Specific comments

Fig. 1 pDCs are the large majority of Tomato positive cells. However, some Tomato+ cells are siglecH negative/low (Panel F) are those cells pDC like cells or do they identify other subsets?

All gates are nicely shown but if only a small fraction of cells is tomato + the provided histogram plots may prevent its identification (panel E) i.e. some BST2 negative Totamto+ cells are present in Lymph nodes. Therefore, please provide in the Supplementary data also DC-subset gating for cells pregated on Tomato+.

In Fig. 2 the authors propose a gating strategy that includes Ly6D and CD19. As shown in fig. 1G. The use of CD19 could be slightly misleading as a non-peer reviewed study suggested that some pDC express CD19 (<https://doi.org/10.1101/310680>), although it is clear that pDCs do not express CD19 (given the few citations of this work) I would recommend to replace CD19 with IgM to prevent future misunderstandings and questioning.

Fig 3 B

Please add the % values for the indicated gates.

(lower plots left) cells are pre-gated as tdT-Zbtb46-; but these pDCs negative for tomato or do they represent cells that have low expression? Are those circulating precursors for pDCs?

Figure 4

pDC, cDC1 and tDCs cells appear based on Seurat clustering to be composed of 2 or 3 clusters.

For cDC1 the expression of group 3 genes is mostly differential despite those being cDC1 specific, i.e. XCR1. Why did the author choose a phenotype-based clustering over a transcriptional one?

Are differences related to specific set of genes? Could you please include the table referring to Seurat clusters (0-12) indicating which marker define each subset/cluster and what are the genes used for the subset assignment (i.e.

Cluster 0, 2, 8 and 12 are pDCs; clusters 7; 9 and 10 are cDC1.

Which clusters are exclusive of infected mice?

Why were Seurat clusters reduced to 7 in Supp. Fig 4 please explain in the text and relate to which subset comprise multiple clusters.

As numbers are used across each panel to define different clusters using different types of analysis I would suggest to use unique numbering for each analysis used (a-z) (i, ii, ...) (I, II....) to avoid confusion in what type of analysis is used and defines the shown clusters across all figures.

Figure 5 and Extended 6

pDCs were previously shown to attach to HEV, which would correspond to the T cell areas.

Please include the missing reference (Diacovo et al. JEM 2005) and stain for HEV to confirm the localization. Diacovo et al showed by intravital microscopy that in the absence of inflammation pDCs remain within the HEV vascular bed and do not extravasate. It is therefore important to address this aspect in your study.

Is the subsequent localization within the MZ reflecting the re-entry within the lymphatics?

The fact that only pDC that produced IFN-I are present in the TCZ could reflect the active recruitment of pDC from peripheral blood occurs at HEV for a prolonged window of time during MCMV infection.

A possibility to address this topic would be to inject few minutes before harvest labeled antiCD45 to visualize pDC that are still contained within HEV from cells which would have extravasated and be in contact with T cells.

Alternatively, vessels can be stained with labeled Dextran. This would allow to discriminate between cells that were exposed to viral products and actively migrated as IFN- producing cells to the T cell zone (TCZ) and are retained within it, and cells that keep migrating to the TCZ from the periphery.

Also, if only IFN producing cells are recruited to the TCZ it implies that specific chemokine receptors allowing for extravasation are expressed. Since CX3CR1 can be excluded, is this a CCR5 dependent mechanism or which cytokine would define the IFN producing subsets?

Please use caution in defining migration patterns as this interpretation of the data does not take into account the high apoptotic rate of activated pDCs, which would influence the proportions of pDCs in the different locations.

Fig. 8

The labelling of the clusters and the table data is quite unclear. NI- was previously used for Not infected, therefore it is not clear if the clusters identified after infection are also included in the table. Please Provide the data on the infected mice and define clusters more precisely or in a separate sheet. I would expect to see the clusters defined as in Figure 8a

The tables Reizis_Tussiwand up and down have the headings

tDC_cDC2_cDC1_pDC_pDC vs pDClike. While the first 4 columns define subsets, the last one is a comparison could you please specify the analysis done and include pDC and pDC-like as individual columns. Please clarify the data.

In the discussion, the developmental aspect is not raised despite the data on progenitors seems very solid. It would be worth including some comments as there is currently a heated debate on the origin of pDCs from lymphoid versus

myeloid progenitors.

Please include the Study from Diacovo et al. (Colonna) where the localization and migration of pDCs to T cell areas is shown by intravital microscopy and proven to be L-selectin dependent.

Please include in the citation also the work of Kastenmuller et al (Germain) that showed in vivo imaging of pDCs.

Minor comments

Include In the colon Panel the CD3 marker (blue) and please check the panel on the right. The red is visible on the left panel but not on the right one.

The gene for L-Selecting (Sell) could be also highlighted as it is likely implicated in the observed migratory behavior.

The use of colors that are colorblind friendly should be preferred especially for the histology. I recommend avoiding the combination of green, red, and yellow despite the convenience as they cannot be easily discriminated.

Reviewer #2:

Remarks to the Author:

Valente et al. introduce a newly generated reporter mouse model that is truly specific for plasmacytoid DCs and allows distinction of pDCs from other phenotypically similar DC subpopulations by flow cytometry and microscopy. They use an elegant approach by combining Siglec H driven Cre recombinase expression with insertion of loxP-STOP-loxP-tdTomato into the Pacsin1 gene locus. Therefore, tdTomato is only expressed in cells where both Siglec H and Pacsin1 are transcriptionally active. After crossing with Zbtb46-eGFP knockin mice which express eGFP in cDC lineage cells, this reporter mouse model is used to delineate pDCs, cDCs, pDC-like cells and transitional DCs which are subsequently characterized by scRNA-seq. pDC-tdTomato reporter mice are used to determine localization of pDCs in lymphoid and non-lymphoid tissues in the steady state and after crossing with IFN-beta-YFP reporter mice to study the positioning of pDCs with and without IFN I production in the spleen after infection with murine cytomegalovirus. This was found to be time-dependent with a higher percentage of IFN-producing pDCs first in the marginal zone and later in the T cell zone. In this model they observed early recruitment of pDCs to the marginal zone where infected cells are found before maximal production of type I IFNs by pDCs. The authors show by scRNA-seq that pDC-like cells and transitional DCs in the spleen show a divergent response after MCMV infection with pDC-like cells behaving similar to pDCs and tDCs resembling cDC2.

The sequencing data generated using this mouse model is potentially useful to identify specific marker genes for tDCs and pDC-like cells that are stable after viral infection as well as to identify specific response genes that could be further studied to functionally characterize these cells.

This is a thorough study describing a useful pDC reporter mouse model, but doesn't provide a lot of new insights into the role of pDCs and other DC subpopulations during viral infection. The scRNA-seq data set is unique as these rare cell types resembling pDCs have not previously been studied separately from pDCs or cDCs after viral infection. However, the significance of this data set as a resource is limited due to the low number of cells that were studied and the restriction to predefined cell subpopulations which likely excludes intermediary cells or so far undefined cells. The authors suggest that their dataset can be used to derive marker genes specific for tDCs and pDC-like cells, such as Tmem176a, Tmem176b and Apod, but there is no validation of these genes to be really specific and useful for example for targeting tDCs specifically to investigate their function further. Similarly, an intersectional genetic strategy using for example Siglech-Cre;Pacsin1LSL-Dta mice to generate mice lacking pDCs specifically or pDC-specific conditional knockout mice is suggested, but not shown in the paper. Therefore, the data shown here remains highly descriptive and does not provide a lot of new functional insight into pDCs, pDC-like cells or tDCs.

Specific points:

- Major
1. Introduction: Siglec-H-eGFP, -Cre -DTR mice should be mentioned and referenced. BDCA2-DTR mice do exist which allow specific depletion of bona fide pDCs in mice – these are not mentioned at all in the paper. It would be good to describe them in the introduction and discuss their benefits and disadvantages in the discussion as well.
 2. Figure 1H: Siglec H expression is not shown, why? It should be shown even if it is not a good marker after infection.
 3. Results to Figure 1 conclusion, line 135: this phenotype was already known to identify bona fide pDCs (e.g. PMID 21508410, 22547585, 31213723, 29925996).
 4. Line 145: pre-pDC were defined by Dress et al. (31213723) as Lin-CD11c-MHCII-Flt3+c-KitintCD115-CD81+SiglecH+IL7Ra+ Ly6D+ cells. Therefore the gating would not be correct. Or the cells gated in extended figure

2A should be called CD11c+ pre-pDC.

5. Figure 3B, C (line 181-183):

6. What is the parent gate before CD11c vs Siglec H in Figure 3B? Show the full gating strategies in the supplemental figures and explain it in the legends.

7. The definitions of pDC-like cells and tDCs is not done the same way as in the publications that the authors refer to that first coined these names (Rodrigues et al. and Leylek et al.). This is very confusing and only clarified later (line 280-283). It would be really important to show the populations as previously defined and then show also the tdTomato signal. B220 is missing as an important marker and should be shown in Figure 3C. By limiting the tDC gate to Siglec H- Ly6D- cells only a subset of tDCs that has a phenotype closer to cDC2 is represented here (and also in the scRNA-seq data set). This should be made clear from the start by calling these cells CD11chigh (Siglec H-) tDCs. pDC-like cells as defined by Rodrigues et al are not gated Ly6D- and would therefore include the Zbtb46+ Ly6D+ cells.

8. CX3CR1- cells within the BST2-/lo SiglecH+ parent gate are ignored in the subsequent phenotype analysis (gaps in the tSNE plot?) and in the scRNA-seq analysis – this is a sizable population – which cells are these? Why were they not looked at? How is there expression of Zbtb46 and other markers? Ly6D could be express weakly on these cells – this depends on the sensitivity of the Ly6D staining.

9. In the tSNE plot in Fig. 3D it would be important to also make visible the cells that are not defined by the gating strategy shown in Fig. 3B and in addition to the backgating also show marker expression as overlays on the tSNE or UMAP plot for all cells that were included in the dimensionality reduction.

10. I think it should be mentioned here that the gated subsets here form a continuum of cells. This is also largely confirmed by the scRNA-seq data.

11. Line 218-219: the heterogeneity of the tDCs was already described by Leylek et al. PMID 31825848, so ZEST mice do not really contribute to this knowledge.

12. Lines 362-366, line 374-375 and line 505: This is not shown in the data that is shown in Figure 6F. The IFN- β -YFP reporter only detects cells that currently produce IFN- β and not cells that have already produced IFN β , because it is not a fate-map mouse model. So you cannot conclude that ex-IFN-producers have migrated from marginal zone to T cell zone. It should be avoided to overstate the results shown here. The data only show that the percentage of pDCs producing IFN is higher in the MZ and lower in the TCZ at the 36 hour time point and then it is lower in the MZ and higher in the TCZ 12 hours later. It is not clear that the IFN-producing cells have migrated to the TCZ or that only IFN-producing cells are licensed to migrate to the TCZ.

13. Line 434-435: "only few genes were found to be differentially expressed between tDC and cDC2" – this is not surprising as only the Cd11chi SigH- Ly6D- subset of tDCs was studied here, which is very close to cDC2.

14. Line 452-453 and 456: the genes mentioned here should be validated by qPCR and/or protein detection and checked how specific they really are to know how useful they really are for potential tDC targeting.

Minor:

1. Figure 2A: very confusing and hard to see– use more space to show this.

2. Extended Figure 2A: CDP should be gated Ly6D-negative to exclude any pDC progenitors (according to Dress et al. 31213723)

3. Extended Figure 2B: CD127 and Siglec H signals seem to be rather weak – this could be improved.

4. Line 262 and 264: discrepancy between CCR9 protein expression and Ccr9 mRNA expression in pDC-like cells – how can this be explained? Are these cells downregulating CCR9 and protein is still there when mRNA has already been downregulated?

5. Line 403: mention IFNL2 as type III IFN is also mentioned in the title of this chapter

Reviewer #3:

Remarks to the Author:

The authors present the generation and validation of an elegant pDC reporter mouse model that is able to specifically 'label' pDC and pDC-like cells in vivo through co-expression levels of Siglech and Pascin 1. The further development of the Zest mouse to refine the definition of pDC-like cells based on Zbt46 expression further strengthens this model.

The validation is robust with respect to the reporter mouse being able to sensitively label pDC and pDC-like cells. Indeed the authors have convinced this reviewer that this is a bona fide pDC reporter mouse, far surpassing current models available. However, the validation of the model, including the novel RNAseq data of pDC-like cells, is largely based on previous elegant work done by the authors in the MCMV infection model, looking at 36hrs of infection or later.

The differential localisation of pDC producing IFN or not during MCMV infection is novel data and of interest, although whether this is a stochastic event or not remains unanswered.

Suggestions for improvement:

Although this group and others cite peak MCMV-induced Type I IFN responses in serum at approx 36hrs after infection, the data of Fig 7 show beautifully that pDC activation/relocalisation to the MZ has occurred (maximally) by 12 hrs after infection. Using this mouse model to examine very early kinetics of infection, from 0-12 hrs, would elevate the novelty of this work and provide a deeper understanding of the role of pDC in these early stages, something that has been missing until now. RNAseq of pDC from these early stages to ascertain whether they are already capable of IFN production would be of interest to the field, as would the phenotype of the pDC like cells, similar to Fig 8.

Moreover, although the spleen has provided useful data in the development of the MCMV infection model, from the perspective of the role and function of pDC during infection it is important that the total numbers are tracked within the spleen (not just percentages as shown in Fig 3 and extended data). In addition, this model would allow the tracking of the efflux of pDC from the bone marrow, a missing 'link' from this work. Is pDC egress from bone marrow occurring at early time points? How does the reporter work on pDC in the bone marrow compartment, one of the most abundant sources of pDC and their immediate immature precursors?

Specific comments of data:

1. In Ext Fig 3 it is unclear how the percentages in Supp Table I relate back to the plots of extended Fig 3, particularly the far right dot plots.
2. Although line 197 states 'that proportions of the distinct DC populations were not 'significantly modified during MCMV infection'', in the data of Supp Table I the percentages of pDC in the 48h infected spleen range from 6-32%; 4-12% at 36h. As stated above, cell numbers are required here- is there a massive loss in some animals of cDC-numbers here for all cell types would help to clarify. Data from additional animals may be required if large variations between animals are observed.
3. Given the variation in cell populations noted in 2- it would seem inadequate to analyse cells from a single infected mouse (from each of 36 & 48h PI) in the data of Ext Fig 5 and Fig 8.

Author Rebuttal to Initial comments

Point-by-point response to the Editor's and reviewer's comments.

Answer to the Editors' comments.

We thank reviewers and editors for their careful evaluation of our manuscript. We are happy that our “new mouse model for specifically identifying pDCs is appreciated”. We value the editors’ decision to leave open the possibility to submit a revised version “as a Technical Report to introduce the new mouse model, pending revision to address the referees' comments” since it will be “of benefit to the immunology community”. We followed that excellent suggestion, by performing a focused series of additional experiments and correcting/editing specific sections of the paper to further strengthen the characterization of our novel mutant mouse strains. We have not included other lines of investigations to provide additional new insights into pDC biology, because that is beyond the scope of a Technical Report and will require several years of work.

1) To deepen our analysis of the expression pattern of the fluorescent reporters in the splenocytes from our mutant mouse models, at steady state and after MCMV-infection, we have performed novel experiments harnessing spectral flow cytometry (Extended Data Fig. 4 and 5). These novel data confirm the high specificity and strong penetrance of tdTomato expression in the pDC-Tom (*Siglech^{iCre};Pacsin1^{LSL-tdT}*) mice. These data also confirm that, in *Zbtb46^{GFP}* mice, most GFP⁺ cells are cDC2s and cDC1s, but that they also encompass tDCs, pDC-like cells and potential pre-cDCs, as well as a minor fraction of *bona fide* pDCs and a specific cluster of myeloid cells that might correspond to monocyte-derived DCs. A small fraction of lymphoid cell types also expresses GFP but with very low mean fluorescence intensity as compared to cDCs.

2) To verify the expression patterns of the *Tmem176a*, *Tmem176b* and *Apod* genes in DC types, at steady state and after MCMV-infection, we have performed mini-bulk qRT-PCR analyses.

3) Our initial statements that scRNA-seq has been achieved on cells isolated from one mouse only for each time point was a mistake, due to miscommunication between the first and last authors when finalizing the initial manuscript version. We apologize for this error. In fact, 2 to 3 mice have been used for each of the time point. This is now documented by projecting on the UMAP the belonging of cells to individual mice (Extended

Data Fig. 7a). A new supplemental Figure illustrates the quantitative contribution of each individual mouse to each cell cluster, and the proportion of cell clusters for each individual mouse. This shows that the cell types and states identified in our scRNA-seq experiment are robust, since they are shared across individual mice, with a similar distribution of cell types and states between mice studied at the same time point of MCMV infection (Supplementary Fig. 1). The corresponding figure legends and main text have been corrected.

4) To measure the absolute numbers of the different DC types during the infection, and in larger cohorts of mice, we have exploited our novel spectral flow cytometry experiments (new Supplementary Table 1).

5) To illustrate original *in vivo* observations enabled for the first time to the best of our knowledge thanks to our novel reporter mouse models, we have added novel confocal microscopy data as 3D reconstructions showing, in the marginal zone of the spleen of infected mice, the tight interactions that exist between pDCs producing IFN-I and virus-infected cells, as well as between pDCs producing or not IFN-I (movie S1-S3).

Our novel experiments/analyses have addressed the referees' concerns on the accuracy, robustness and reproducibility of our results, including for unbiased analysis of tdTomato expression in splenocytes, and for ensuring that our scRNA-seq data were not biased due to focus on too few animals that may behave peculiarly. We believe that our revised manuscript is well suited for publication as a Technical Report in *Nature Immunology*.

Changes in the revised manuscript are highlighted in the Microsoft Word file corresponding to the main text.

We have revised our manuscript in a manner conforming to the Technical Report format instructions. Although we edited our main text for concision, it is longer than recommended (6,524 words), due to the many editions required for answering the reviewers' comments.

Please find below a point-by-point reply to the reviewers' comments.

Reviewer #1:

We thank Reviewer #1 for stating that our “mouse model is certainly a significant advance”, “allows improving the characterization of pDC-like and tDCs”, “flow panels are very clean and the data solid”.

Specific comments

1) Fig. 1 pDCs are the large majority of Tomato positive cells. However, some Tomato⁺ cells are SiglecH^{neg/low} (Panel F). Are those cells pDC-like cells or do they identify other subsets?

The data from Figure 1F show that, in addition to B220, tdTomato⁺ cells express Ly6D and CCR9, a combination of markers that is rather specific to pDCs; other myeloid cells including the vast majority of pDC-like cells and tDCs do not express Ly6D and only much lower levels of CCR9. SiglecH is known to be down-regulated upon pDC activation (see new panel Fig. 1i). The SiglecH^{neg/low} cells from Fig. 1f may correspond to the small fraction of splenic DCs that have undergone homeostatic maturation under steady state conditions.

To deepen our analysis of the identity of tdTomato⁺ and GFP⁺ cells in the splenocytes from our ZeST mice, we have now performed novel experiments harnessing spectral flow cytometry with unsupervised cell clustering based on all cell surface markers, without considering the expression of tdTomato and GFP. These data show that pDC represent 81.5±6% of tdTomato⁺ splenocytes, depending on individual mice and on timepoints after MCMV infection (Extended Data Fig. 4e, main text lines 207-209). The other splenocytes falling into the tdTomato⁺ gate belong to a variety of myeloid or B cell clusters, which the contribution of each of those clusters being extremely low although increasing a little in infected animals. Moreover, the MFI for tdTomato is much higher in the pDCs than in the other cell clusters where it is not much above the autofluorescence signal observed in control C57BL/6 mice (Extended Data Fig. 4f, main text lines 213-215). It is likely that most of the non-pDC cells that fall into the tdTomato⁺ gate are the few cells that do not express tdTomato but have a higher autofluorescence than most other cells in each of the other phenotypic cell clusters.

2) All gates are nicely shown, but if only a small fraction of cells is Tomato⁺ the provided histogram plots may prevent its identification (panel E), i.e. some BST2^{neg} Tomato⁺ cells are present in Lymph nodes. Therefore, please provide in the Supplementary data also DC-subset gating for cells pre-gated on Tomato⁺. We have replaced the data from the lymph nodes in Fig. 1e by a dot plot from another, more representative, mouse, where ~93% of the tdTomato⁺ cells co-express CD11c and high levels of BST2. The cells from the tdTomato⁺ gate that are not expressing CD11c and BST2 appear to express either CD19 or F4/80 with Ly6C, suggesting that they are autofluorescent B cells or macrophages.

3) In Fig. 2 the authors propose a gating strategy that includes Ly6D and CD19. The use of CD19 could be slightly misleading as a non-peer reviewed study suggested that some pDC express CD19 (<https://doi.org/10.1101/310680>), although it is clear that pDCs do not express CD19 [...] I would recommend to replace CD19 with IgM to prevent future misunderstandings and questioning.

Between 2004 and 2009, the group of Mellor and Munn published a series of papers reporting the existence and characterization of CD19⁺ pDC (Baban et al., 2009; Baban et al., 2005; Manlapat et al., 2007; Mellor et al., 2005; Munn et al., 2004). In 2010, they published another study correcting this wrong interpretation, showing that these cells were not pDC but a subset of B cells (Johnson et al., 2010). I suspect that the same problem of mis-attribution of cell identity applies to the preprint study suggesting that some pDC express CD19 and that is cited by the reviewer. To further examine this issue, we included IgM in addition to CD19 in our novel spectral flow cytometry experiments. Unbiased clustering of splenocytes show that pDC express neither CD19 nor IgM, which are detected only on B cells. Hence, we do not think that replacing CD19 by IgM is required for the proposed gating strategy for pDC.

4) Fig 3 B. Please add the % values for the indicated gates.

We thank the reviewer for pointing out this issue that we have now corrected.

5) Fig 3 B (lower plots left) cells are pre-gated as tdT⁺Zbtb46⁻; but are these pDCs negative for Tomato, or do they represent cells that have low expression? Are those circulating precursors for pDCs?

Contrary to cDCs, pDCs have been shown to exit the BM as already terminally differentiated cells. We showed that, in the BM, tdTomato is already expressed at intermediate levels in the late CD11c⁺ pre-pDC precursor stage (Fig. 2c, Lin^{neg} B220⁺ CD11c⁺ MHC-II^{neg} SiglecH⁺ BST2⁺ cells). Moreover, BM pDC (Fig. 2b) express

the same level of tdTomato as spleen pDC (Extended Data Fig. 1c). Hence, it is unlikely that the cells expressing SiglecH and Ly6D in the spleen but negative or low for tdTomato are pDC precursors.

This is specifically to address this question that we enriched Tomato^{neg} cells bearing a pDC phenotype and characterized them by scRNA-seq. Our analysis shows that the vast majority of these cells cluster with Tomato⁺ pDCs in the Seurat and R-Phenograph analyses and are assigned as bona fide pDCs by the sgCMAP analysis (Extended Data Fig. 6, Extended Data Fig. 7 and Supplementary Table 2). Thus, these cells are a small fraction of bona fide pDC that are low or negative for tdTomato. We further confirmed this in our novel spectral flow cytometry experiments, where up to 18% of the cells identified as pDC by supervised gating are negative or low for tdTomato, depending on individual mice and infection conditions (Extended Data Fig. 5f).

6) Figure 4. pDC, cDC1 and tDCs appear based on Seurat clustering to be composed of 2 or 3 clusters. For cDC1 the expression of group 3 genes is mostly differential despite those being cDC1 specific, i.e. XCR1. Why did the author choose a phenotype-based clustering over a transcriptional one?

The clustering on Fig. 4 is not a phenotype-based clustering. This is a clustering based on cell type assignment, irrespective of cell states, combining both transcriptomic and phenotypic analyses, as explained in (Extended Data Fig. 7). The data shown include all the cells that passed quality controls, not only from uninfected animals but also from mice that were infected by MCMV for 36h or 48h.

Here, the goal was to assign a DC type to each individual cell irrespective of its activation state. The analysis considering the 2 or 3 clusters for each DC type, e.g. the cell activation states, are shown later in the paper, as (Fig. 8) and (Extended Data Fig. 10).

Assigning a DC type to each individual cell irrespective of its activation state is key, because the numbers of genes whose expression is affected by DC activation far exceeds the number of genes that define DC types. Moreover, many genes that are specific of a given DC type at steady state are strongly downregulated upon activation, such that they cannot be used anymore individually to identify DC types. This is the case for the *Xcr1* gene and many others of the group 3 genes, as we reported previously (Ardouin et al., 2016; Crozat et al., 2011; Manh et al., 2013). This issue can lead to transcriptomic clustering of DC primarily according to their activation states rather than to their DC types, as we first reported when performing bulk microarray studies (Manh *et al.*, 2013). Hence, in scRNA-seq analyses, it is relatively frequent that certain clusters are heterogeneous and encompass different DC types in a similar activation state. This is especially the case on the one hand for proliferating DC/DC-precursors, and on the other hand for terminally mature DC. This is well illustrated in the paper by the group of Miriam Merad on mregDC (Maier et al., 2020). This is also discussed in a recent commentary published in Nature Immunology, where, as we do ourselves, the authors emphasize the necessity to clearly distinguish in scRNA-seq data distinct DC types from different activation states of the same DC type, to avoid coining new names for activation states of already identified and named DC types, referring in particular to the current use in the literature of the name «DC3» for the mature DC cluster identified in many scRNA-seq data, whereas «DC3» is also being used for a human DC type that is ontogenetically and functionally distinct from human pDC, cDC1, cDC2 and MoDC (Ginhoux et al., 2022).

Of note, however, while the *Xcr1* gene is strongly downregulated in cDC1 upon maturation, this is not the case of the XCR1 protein, as we reported before (Crozat *et al.*, 2011). This is clearly visible on Fig. 4, where all individual cells identified as cDC1 express the XCR1 protein (Fig. 4b) whereas a fraction are low or negative for the *Xcr1* gene as rightfully pointed out by the reviewer (Fig. 4c). This is even clearer when splitting cDC1 according to their activation state as analyzed later in the paper, whereby expression of the *Xcr1* gene is strongly decreased in activated cDC1 whereas XCR1 protein expression is only slightly decreased (Extended Data Fig. 10a,b). Indeed, in the Maier et al. paper referred in the paragraph just above, the mregDC cluster encompassed both cDC1 and cDC2, and this heterogeneity could be resolved by the authors only by using the phenotypic data enabled by their use of the CITE-seq technology, whereby mregcDC1, although negative for the *Xcr1* gene, were positive for expression of the XCR1 and CD103 cell surface markers but negative for CD11b, and reciprocally for mregcDC2.

Here, we show that using sgCMAP, with specific gene modules carefully defined from an independent dataset, can allow properly assigning a DC type to most individual cells (Extended Data Fig. 7).

We have now clearly stated this issue in the main text (lines 288-292, and lines 768-788).

7) Figure 4. Are differences related to specific set of genes? Could you please include the table referring to seurat clusters (0-12) indicating which marker define each subset/cluster and what are the genes used for the subset assignment (i.e. Cluster 0, 2, 8 and 12 are pDCs; clusters 7; 9 and 10 are cDC1).

The analysis of all clusters combining heterogeneity of both DC types and their activation state is shown as (Fig. 8) and (Extended Data Fig. 10). We did give in (Supplementary Table 2) the list of marker genes for each of the cluster, in the 13 spreadsheets named «Clean_C0_markers_pDC» until «Clean_C12_markers_IFN-pDC» as described in the first spreadsheet listing the content of this Excel file. DC type assignment was not done using individual genes but an unbiased approach combining R-Phenograph clustering, Seurat Clustering and sgCMAP analysis, as explained in (Extended Data Fig. 7). Indeed however, Seurat clusters were tentatively assigned to cell types in the first phase of our analyses, based on their marker genes. In Supplemental Table 2, in each of the spreadsheets giving the marker genes of the clusters, we highlighted in green genes known to be selectively expressed in the DC type that was assigned to the cluster. We apologize if this was not explained clearly enough in the paper. The legends of Extended data Fig. 6 and 7 been revised to make this information easier to find: “Seurat clusters were annotated for cell types based on analysis of their marker genes (see Supplemental Table 2)”.

Sets of genes differentially expressed in combinations of DC types and activation states are identified and discussed later in the paper, in (Fig. 8), (Extended Data Fig. 10) and the accompanying main text.

8) Figure 4. Which clusters are exclusive of infected mice?

This information was given on (Fig. 8a), where each individual cell on the UMAP is shown with a color corresponding to its DC type belonging, and with a symbol corresponding to the time point after infection of the mouse from which the cell was isolated. Cells from uninfected mice (NI) are shown as a «+», from the 36h-infected mice as a square, and from the 48h-infected mice as a triangle (see lower right information in the graphical legend box at the bottom of panel Fig. 8a). We apologize if this was not explained clearly enough in the paper. In Fig. 4a, we added the Seurat cluster numbers in grey boxes on the UMAP for clarification. In Extended Data Fig. 7a, we used a combination of shapes and colors to allow the reader visualizing which individual cells are from which mouse and from which infection condition. We edited the legend of Fig. 4 to make this information easier to find. This information is also given in detail in (Supplementary Table 2), in the second spreadsheet named «complete_metadata». Since our combined sgCMAP and R-Phenograph analyses had shown that certain Seurat clusters were heterogeneous, encompassing distinct DC types in a similar activation state, we split these heterogeneous clusters in subclusters corresponding each to a specific combination of a unique DC type and activation state. This explains why, in Fig. 8, there are 15 combinations of DC types/activation states, established from 13 Seurat clusters. Specifically, the Seurat clusters n° 1 and 4 were each split in 2, according to individual cell assignment to cDC2 or tDC (see Extended Data Fig. 7e).

8) Supp. Fig 4. Why were seurat clusters reduced to 7 in Supp. Fig 4? Please explain in the text and relate to which subset comprise multiple clusters.

Extended Data Fig. 6 is focusing on the cells isolated from the uninfected mice only. Hence, this dataset contains only one activation state (quiescent). Thus, it is less heterogeneous than the dataset also encompassing cells isolated from 36h- and 48h-infected mice, where 2 to 4 activation states are observed for each DC type. Hence, 7 clusters are sufficient to resolve the heterogeneity of the dataset from the uninfected mice. We apologize if this was not explained clearly enough in the paper. We have edited the main text of the paper to make this information easier to find, line 239: “We first analyzed 343 steady state splenocytes isolated from non-infected (NI) mice”.

9) As numbers are used across each panel to define different clusters using different types of analysis I would suggest to use unique numbering for each analysis used (a-z) (i, ii, ...) (I, II....) to avoid confusion in what type of analysis is used and defines the shown clusters across all figures.

We thank the reviewer for this excellent suggestion, which we have implemented in the revised version of the manuscript, both on the Figures and in Supplementary Table 2.

10) Figure 5 and Extended 6. pDCs were previously shown to attach to HEV, which would correspond to the T cell areas. Please include the missing reference (Diacovo et al. JEM 2005) and stain for HEV to confirm the localization. Diacovo et al showed by intravital microscopy that in the absence of inflammation pDCs remain within the HEV vascular bed and do not extravasate. It is therefore important to address this aspect.

This comment does not apply to our study. Indeed, the data shown in Figure 5 and Extended Data Fig. 8 focus on pDC in the spleen, an organ that does not contain HEV (Chauveau et al., 2020), contrary to lymph nodes. Therefore, there is no point in staining for HEV in the spleen.

11) Is the subsequent localization within the MZ reflecting the re-entry within the lymphatics?

To the best of our knowledge, only one paper has studied lymphatics in the spleen (Shimizu et al., 2009). These lymphatics look like efferent ones, meaning that they are used only to get out of the spleen, not to enter/get back to the spleen. In addition, the position of these lymphatics inside the T cell zone and close to the central artery rather excludes a physical proximity with the MZ. In any case, nobody has ever shown that these lymphatics would allow cells to enter the spleen. All circulating cells have been reported to enter the spleen via the venous sinuses in, or peripheral to, the marginal zone (Chauveau *et al.*, 2020). Therefore, we believe that this comment does not apply to our study.

Clustering of pDC at the MZ starting around 12h after infection could result either from intra-splenic migration of the pDC that already resided in this organ in the T cell zone before infection, since the proportion of pDC is decreasing in the T cell zone but not in the red pulp at the time it increases in the marginal zone (Fig. 7b). Alternately, pDC freshly egressed from the blood in the marginal zone may be retained there starting 12h00 after infection. Attempting to answer this question is beyond the scope of a Technical Report, representing a major undertaking for future studies.

12) The fact that only pDC that produced IFN-I are present in the TCZ could reflect that active recruitment of pDC from peripheral blood occurs at HEV for a prolonged window of time during MCMV infection. A possibility to address this topic would be to inject few minutes before harvest labeled antiCD45 to visualize pDC that are still contained within HEV from cells which would have extravasated and be in contact with T cells. Alternatively, vessels can be stained with labeled Dextran. This would allow to discriminate between cells that were exposed to viral products and actively migrated as IFN- producing cells to the T cell zone (TCZ) and are retained within it, and cells that keep migrating to the TCZ from the periphery.

This hypothesis is not relevant to our study, since we are looking at pDC in the spleen where there are no HEVs. All circulating cells have been reported to enter the spleen via the venous sinuses in, or peripheral to, the marginal zone, and then migrate from the marginal zone to the other splenic zones (Chauveau *et al.*, 2020).

In addition, we previously showed that the pDC circulating in the blood do not produce IFN-I during MCMV infection (Zucchini et al., 2008).

Moreover, consecutive to systemic MCMV infection, pDC production of IFN-I occurs during a tight window (33h-40h) and is terminated by 48h (Abbas et al., 2020; Zucchini *et al.*, 2008). As explained in response to point (13) of Reviewer #2, at 48h after infection, EYFP⁺ pDC are not anymore producing IFN-I but are fate-mapped for this past-activity due to their maintenance of high expression levels of EYFP for >12 hours after they have secreted all of the IFN-I they produced.

Finally, this hypothesis is not consistent with a number of orthogonal data that we reported in our previous paper (Abbas *et al.*, 2020), and which demonstrated that pDC first produce IFN-I in the MZ around 36h after infection and later migrate to the TCZ after they have ceased producing IFN-I but still express EYFP. Please see excerpts of our previous paper included at the end of this document as Fig. R1 for reviewers.

13) Also, if only IFN producing cells are recruited to the TCZ it implies that specific chemokine receptors allowing for extravasation are expressed. Since CX3CR1 can be excluded, is this a CCR5 dependent mechanism or which cytokine would define the IFN producing subsets?

We reported in our previous paper that the pDC that have produced IFN-I specifically upregulated CCR7,

which likely contributed to confer them their ability to migrate to the TCZ (Abbas *et al.*, 2020). Please see excerpts of our previous paper included at the end of this document as Fig. R1 for reviewers.

14) Please use caution in defining migration patterns as this interpretation does not take into account the high apoptotic rate of activated pDCs, which would influence the proportions of pDCs in the different locations. We agree that there is a high apoptotic rate of activated pDCs, which must be considered when defining migration patterns. However, in our previous paper (Abbas *et al.*, 2020), we showed that the pDC that are fate-mapped by EYFP for past-IFN-I production (EYFP^{pos} IFN-I^{neg} cells) accumulate over time until at least 48 hours after infection, whereas the fraction of pDC expressing IFN-I (EYFP^{pos} IFN-I^{pos}) peaks around 36h after infection (see Fig. R1 for reviewers at the end of this document). This shows that the pDC that have produced IFN-I during MCMV infection harbor a prolonged survival as compared to other pDC. Moreover, the novel data shown in the present manuscript clearly show a striking different behavior of *Ifnb1*^{EYFP(pos)} versus *Ifnb1*^{EYFP(neg)} pDC, where many of the later get stuck at the MZ whereas the former migrate to the TCZ, which cannot be explained by pDC apoptosis.

15) Fig. 8. The labelling of the clusters and the table data is quite unclear. NI- was previously used for Not infected, therefore it is not clear if the clusters identified after infection are also included in the table. Please Provide the data on the infected mice and define clusters more precisely or in a separate sheet. I would expect to see the clusters defined as in Figure 8a.

In (Fig. 8a), each individual cell on the UMAP is shown with a color corresponding to its DC type belonging, and with a symbol corresponding to the time point after infection of the mouse from which the cell was isolated. Cells from uninfected mice (NI) are shown as a «+», from the 36h-infected mice as a square, and from the 48h-infected mice as a triangle (see lower right information in the graphical legend box at the bottom of panel Fig. 8a). We apologize if this was not explained clearly enough in the paper. We edited the legend of Fig. 8 to make this information easier to find. We added the Seurat cluster numbers in grey boxes on the UMAP for clarification. In Extended Data Fig. 7a, we used a combination of shapes and colors to allow the reader visualizing which individual cells are from which mouse and from which infection condition. This information is also given in detail in (Supplementary Table 2), in the second spreadsheet named «complete_metadata», where individual cell belonging to the Seurat cluster is indicated in the column entitled «clean_full_dataset_Seurat_clusters», and individual cell belonging to the combination of DC type and activation state shown on (Fig. 8a) is indicated in the column entitled «clean_full_dataset_final_cell_type_activation_state». The infection status of the mouse from which the cells were sorted is indicated in the column entitled «timepoint», where cells from the uninfected mice are those of «0H», and the cells from infected mice those of «36H» and «48H».

16) The tables Reizis_Tussiwand up and down have the headings tDC_cDC2_cDC1_pDC_pDC vs pDClike. While the first 4 columns define subsets, the last one is a comparison. Could you please specify the analysis done and include pDC and pDC-like as individual columns. Please clarify the data.

The sgCMAP signatures are composite signatures, encompassing two sub-signatures, one for the genes expressed to higher levels in the target cell type as compared to other cell types (“UP” fraction of the signature), and the other for the genes less expressed in the target cell type than in other cell types (“DN” fraction of the signature).

The Tussiwand study encompassed only 2 cell types, pDC and pDC-like cells, analyzed through scRNA-seq. Hence, from this dataset, we retrieved one composite signature encompassing the genes expressed higher in pDC above pDClike (“UP”) and reciprocally the genes less expressed in pDC than in pDClike (“DN”).

The Reizis study encompassed 4 cell types, tDC, cDC2, cDC1 and pDC, analyzed through bulk RNA-seq. We retrieved and re-analyzed this dataset to generate signatures for each DC type as compared to the 3 other ones. Hence, we ended up with 4 composite signatures, one for each DC type, with its “UP” and “DN” genes when compared to the 3 other DC types.

Hence, in both cases, the signatures are derived from comparisons between DC types, but for pDC-like, the only dataset that was available was restricted to a comparison with pDC.

One of the originalities of our paper is to provide for the first time a side-by-side comparison of the gene expression profiles of both pDC-like cells and CD11c^{high} tDC, in comparison to one another, and for the pDC-like cells in comparison also to cDC1 and cDC2 and not only to pDC.

To make clearer how we generated the sgCMAP signatures, we have extended our description in the materials and methods section (lines 798-806): “An sgCMAP analysis⁵⁷ was performed on this dataset using cell type-specific signatures (tDC, cDC2, cDC1, pDC) established upon re-analysis with BubbleGUM GeneSign module⁵⁹ of a published²⁸, independent, bulk RNA-seq dataset (GEO accession number GSE76132), and a relative pDC_vs_pDClike signature retrieved from a previously published study⁹. The GSE76132 dataset encompassed 4 cell types, tDC, cDC2, cDC1 and pDC, analyzed through bulk RNA-seq. We retrieved and re-analyzed this data to generate signatures for each DC type as compared to the 3 other ones. Hence, we ended up with 4 composite signatures, one for each DC type, with its “UP” and “DN” genes when compared to the 3 other DC types. The study by Rodrigues et al⁹ encompassed only 2 cell types, pDC and pDC-like cells, analyzed through scRNA-seq. Hence, from this dataset, we retrieved one composite signature encompassing the genes expressed higher in pDC above pDClike (“UP”) and reciprocally the genes less expressed in pDC than in pDClike (“DN”).”

17) In the discussion, the developmental aspect is not raised despite the data on progenitors seems very solid. It would be worth including some comments as there is currently a heated debate on the origin of pDCs from lymphoid versus myeloid progenitors.

In his introductory summary of our study, this reviewer stated that “the data presented is solid and focused on the characterization of the subsets rather than engaging in the current debate of pDC development”. We appreciate this very positive statement. We chose not to engage here in the current debate of pDC development, because our data on progenitors does not provide any definitive result supporting preferential differentiation of pDC in vivo from a “lymphoid” or “myeloid” path. Hence, we have now only stated in the discussion (lines 501-504) that “Regarding the current debate on pDC ontogeny^{9, 10, 48}, our study does not provide any definitive data supporting preferential differentiation of pDC in vivo from either a “lymphoid” or a “myeloid” path, because our reporter model leads to Tomato expression only starting from the late CD11c⁺ pre-pDC differentiation state that is common to the two paths”.

18) Please include the Study from Diacovo et al. (Colonna) where the localization and migration of pDCs to T cell areas is shown by intravital microscopy and proven to be L-selectin dependent.

This study pertains to pDC entry into lymph nodes through HEVs. We believe that it is not relevant to our study since it focuses on the spleen where there are no HEVs. All circulating cells have been reported to enter the spleen passively via the venous sinuses in, or peripheral to, the marginal zone (Chauveau *et al.*, 2020).

19) Please include in the citation also the work of Kastenmuller et al (Germain) that showed in vivo imaging of pDCs.

We have added this citation in the revised discussion (lines 487-491): “For the same specificity issue, and because pDC downregulate Siglech upon activation⁴, Siglech-GFP mice are not suitable to identify pDC in vivo⁷, even though it has been attempted by using high GFP intensity combined with plasmacytoid morphology to discriminate pDC from other GFP⁺ cells and examine their behavior in lymph nodes from virus-infected mice⁴⁶”.

Minor comments

a) Extended Data Fig. 6. Include In the colon Panel the CD3 marker (blue) and please check the panel on the right. The red is visible on the left panel but not on the right one.

The corresponding figure has been replaced by a new one, with the color code changed to make it colorblind friendly, and inclusion of the CD3 marker. This figure shows that pDC are absent or extremely rare in the normal colon.

b) The gene for L-Selecting (Sell) could be also highlighted as it is likely implicated in the migratory behavior. As explained above for point (18), we believe that it is not relevant to our study.

c) The use of colors that are colorblind friendly should be preferred especially for the histology.

We have changed the color code to make it colorblind friendly, for the histology figures (Fig. 5, Fig. 6,

Extended data 8, Fig. 7).

Reviewer #2:

We thank Reviewer #2 for acknowledging that our study is “thorough”, demonstrating that our “newly generated reporter mouse model is truly specific for plasmacytoid DCs and allows distinction of pDCs from other phenotypically similar DC subpopulations by flow cytometry and microscopy”. We are also happy that this reviewer finds that “the approach used to engineer” our mutant mouse model is “elegant”, and our scRNAseq dataset “unique” and “potentially useful to identify specific marker genes for tDCs and pDC-like cells that are stable after viral infection as well as to identify specific response genes that could be further studied to functionally characterize these cells”.

General remarks.

A) The significance of [the scRNA-seq] data set as a resource is limited due to the low number of cells that were studied and the restriction to predefined cell subpopulations which likely excludes intermediary cells or so far undefined cells.

We do agree that it would have been ideal to be able to sort the cells in a more unbiased manner, i.e., to sort thousands of tdTomato⁺ cells or of GFP⁺ cells, in an attempt to characterize all cells expressing these fluorescent reporters, including the minor fractions that could not be captured with the strategy we used.

Flow cytometry sorting never reaches 100% purity. Contaminations of target cell types by other populations always occur, even if at low level, as shown in all a posteriori analyses of scRNA-sequencing experiments. Sorting all TdTomato⁺ cells without some level of cross-contamination by tdTomato^{neg} cells is make especially difficult by the relatively weak expression of tdTomato. A posteriori reanalysis of tdTomato expression cannot rely on the expression of the *Pacsin1* or tdTomato genes, because they generate low copy number transcripts, leading to increased frequency of drop-out, false negative, cells. Thus, for the purpose of a posteriori reanalysis of the phenotype of the cells used in the scRNA-seq experiment, it was necessary to preserve the information about their individual expression levels of tdTomato and GFP. This could not be performed by commercial high throughput technologies like 10X Genomics, because even though its CITEseq version allows recording the level of cell surface expression for marker genes for which antibodies exist, it does not yet allow recording fluorescence intensity for intracellular molecules including fluorescent reporters.

Thus, our purpose could only be achieved by index sorting, i.e. sorting each cell individually in wells of 96-well plates while recording their fluorescence intensity for tdTomato and GFP, followed by library generation individually for each cell to keep the link between its phenotype and transcriptome. We are not equipped to perform these experiments with a robot. Everything needed to be done manually. The cost per cell for this technology is ~5-fold higher than for the 10X Genomics technology. In the context of these technological and financial constraints, having generated data for 1,344 individual cells, representing fourteen 96-well plates, cannot be considered as a limited number of cells. After removal of cells that did not pass quality controls or were contaminants, we could analyze 951 cells, which remains a high cell number for the technology used.

In any case, our analysis did achieve our goal: to be used as a complementary, orthogonal, approach to our unbiased flow cytometry analysis, in order to (i) confirm that the vast majority of tdTomato⁺ cells are bona fide pDCs, (ii) confirm that a small fraction of bona fide pDC do not express tdTomato, and (iii) perform for the first time to the best of our knowledge a deep side-by-side characterization of the molecular make-up and transcriptional response to infection of pDC, pDC-like cells and CD11c^{high} tDC. As underlined by the reviewer, our dataset is unique and interesting. It is useful and sufficient for introducing to the community our novel mutant mouse strains, as a Technical Report rather than a Resource Paper as advised by the editors.

B) The authors suggest that their dataset can be used to derive marker genes specific for tDCs and pDC-like cells, such as *Tmem176a*, *Tmem176b* and *Apod*, but there is no validation of these genes to be really specific and useful for example for targeting tDCs specifically to investigate their function further.

We did not find commercial antibodies specific for mouse *Tmem176a* and *Tmem176b*. Hence, to confirm our results, we performed qRT-PCR on mini-bulk sorted DC types, at steady state and at 48h after MCMV infection. These data are included as supplemental material in the revised manuscript (Extended Data Fig.

10d). The results are consistent with the expression pattern observed in scRNA-seq. They confirm that *Tmem176a* and *Tmem176b* are expressed selectively in CD11c^{high} tDCs at steady state, but are induced in pDC-like cells and cDC2s upon activation to levels close to those observed in the CD11c^{high} tDCs isolated

from the same infected animals. Further assessing whether expression of *Tmem176a* or *Tmem176b* could help identifying and targeting tDCs, or contribute to better understand the heterogeneity of cDC2s, is not achievable in the frame of the current study, since it will require the development of antibodies or reporter mice. Our qRT-PCR results confirm preferential steady state expression of *Apod* in pDC-like cells, with strong induction at 48h after infection in pDC, pDC-like cells and cDC1s, lesser in tDCs and inexistent in cDC2s.

C) Similarly, an intersectional genetic strategy using for example *Siglech-Cre;Pacsin1LSL-Dta* mice to generate mice lacking pDCs specifically or pDC-specific conditional knockout mice is suggested, but not shown in the paper. Therefore, the data shown here remains highly descriptive and does not provide a lot of new functional insight into pDCs, pDC-like cells or tDCs.

We do agree that our data do not provide a lot of functional insights into pDCs, pDC-like cells or tDCs. Yet, our work has generated and validated novel mouse models, and generated unique data, that significantly advance knowledge on the identities and molecular make-up of pDC-like cells and $CD11c^{high}$ tDCs, which will be useful for further studies aiming at characterizing their functions and molecular regulation. Our work did provide novel insights into the dynamics of pDC recruitment and activation in vivo during MCMV infection. Hence, in line with the advice provided by the editors, we think that publishing our study as a Technical Report will be of strong interest and benefit to the immunology community.

To further illustrate original in vivo observations enabled for the first time to the best of our knowledge thanks to our novel reporter mice, we have added novel confocal microscopy data as 3D reconstructions showing, in the marginal zone of the spleen of MCMV-infected mice, the tight interactions that exist between pDCs producing IFN-I and virus-infected cells, as well as between pDCs producing or not IFN-I (movie S1-S3). Due to length constrains, we removed the sentence on *Siglech-Cre;Pacsin1LSL-Dta* mice.

Major points.

1) Introduction: *Siglec-H-eGFP, -Cre -DTR* mice should be mentioned. *BDCA2-DTR* mice do exist which allow specific depletion of bona fide pDCs in mice – these are not mentioned at all in the paper. It would be good to described them in the introduction and discuss their benefits and disadvantages in the discussion. We followed this advice by adding brief paragraphs in the introduction (lines 54–66) and discussion (lines 484–491). Since the main text of our paper is already longer than recommended by the Journal, we could not elaborate more on this aspect. We thus also referred readers to our recent review where different mutant mouse models used to target pDCs have been thoroughly discussed (Dalod and Scheu, 2022).

2) Fig. 1h: *SiglecH* expression is not shown, why? It should be shown even if not a good marker after infection. Fig. 1h is a gating strategy that has been identified in an unbiased way by the HyperFinder computational algorithm, taking advantage of the specificity of our novel pDC-Tomato mouse model to allow unequivocal pDC identification as $Tomato^{+}$ cells, both at steady state and during MCMV infection, to then identify the best cell surface marker combination to identify the same cells without considering their $tdTomato$ expression. Hence, we are showing the output of the algorithm, as an objective, unbiased, guide for improving future gating strategies for rigorous identification of mouse pDC irrespective of their activation state. The lack of inclusion of *SiglecH* in this automatically generated gating strategy means that, for proper phenotypic identification of pDC from infected mice, this cell surface marker is not necessary and less informative than the other markers included. This is indeed likely because *SiglecH* is not a good marker after infection, since it is downregulated, most strongly on IFN-I-producing pDCs (Zucchini *et al.*, 2008). We added *SiglecH* expression as Fig. 1i to document it and thus follow the reviewer recommendation. *Ly6D*, *CCR9* and *BST2* are not downregulated on activated pDC, therefore constituting more robust/universal pDC markers.

3). Results to Figure 1 conclusion, line 135: this phenotype was already known to identify bona fide pDCs (e.g. PMID 21508410, 22547585, 31213723, 29925996).

We propose to unambiguously identify pDC in homeostatic and inflammatory conditions as $Ly6D^{high} BST2^{high} CD19^{neg} B220^{+} CD11b^{neg} CD11c^{+}$. This gating strategy is not exactly the same as those cited by the reviewer.

The study PMID 21508410 (Schlitzer et al., 2011) was focused on steady state pDC. pDC were defined as CCR9⁺ BST2⁺ CD11c⁺ cells, which would lead to contamination by other cell types in certain tissues or

under inflammatory conditions. For pDC identification, Ly6D positivity was not used, and neither were negativity for CD19 and CD11b. CD11b negativity was assessed a posteriori on gated/sorted cells.

In the PMID 22547585 study (Schlitzer et al., 2012), pDC were defined as $CCR9^{+/-}$ Siglec-H^{high} BST2^{high} CD11c^{int} cell. For pDC identification, Ly6D positivity was not used, and neither were negativity for CD19 and CD11b. Under inflammatory conditions or in certain tissues, this gating strategy would lead to losing certain activated pDC and possibly to contamination by pDC-like cells, macrophages or mast cells.

In the PMID 31213723 study (Dress et al., 2019), pDC were defined as Lin^{-} CD11b⁻ CD11c^{int} SiglecH⁺ B220⁺ BST2⁺ cells, which is indeed close, but not identical, to the gating strategy we propose. This gating strategy could lead to contamination by pDC-like cells, which is overcome in our proposed gating strategy by selecting Ly6D^{high} BST2^{high} cells. In addition, under inflammatory conditions or in certain tissues, the gating strategy from PMID 31213723 would lead to losing certain activated pDC, due to their downregulation of SiglecH. The gating strategy that we propose overcomes this issue by not requiring SiglecH for identifying pDC.

In the PMID 29925996 study (Rodrigues et al., 2018), spleen pDC were defined as $CD3^{-}$ CD19⁻ BST2⁺ SiglecH⁺ Zbtb46-GFP⁻ cells. Without use of the Zbtb46-GFP reporter, this gating strategy would lead to contamination by pDC-like cells and macrophages. pDC contamination by pDC-like cells was indeed one of the main discoveries in this study. Here, we propose a gating strategy using only antibodies, for researchers who do not have access to the *Zbtb46*^{GFP} reporter mice. Contamination by pDC-like cell is avoided by selecting Ly6D^{high} BST2^{high} cells. Under inflammatory conditions or in certain tissues, the gating strategy from PMID 29925996 would lead to losing certain activated pDC, due to their downregulation of SiglecH. The gating strategy that we propose overcome this issue by not requiring SiglecH for identifying pDC.

The vast majority of researchers working on mouse immune cells and including pDC in their analyses still use gating strategies that are not specific enough. Many papers still define pDC as $BST2^{+}$ B220⁺ CD11c⁺, which leads to contamination by other cell types, not only by pDC-like cells and tDC but also by monocytes, macrophages, NK cells, B cells or mast cells. Hence, we believe that it is important to try to reach a consensus on a more robust phenotypic definition of pDC. We thought that a good way to achieve this aim is harnessing our novel reporter mice for unbiased phenotypic definition of Tomato⁺ cells with a computational algorithm.

We have edited the main text of the paper to further emphasize that the use of Ly6D as a marker crucial for pDC identification stems from the PMID 29925996 and 31213723 studies discussed above (lines 116-117).

4) Line 145: pre-pDC were defined by Dress et al. (31213723) as $Lin-CD11c-MHCII-Flt3+c-KitintCD115-CD81+SiglecH+IL7R\alpha+ Ly6D+$ cells. Therefore the gating would not be correct. Or the cells gated in extended figure 2A should be called CD11c⁺ pre-pDC.

We thank the reviewer for pointing this out. We have accordingly renamed these cells “CD11c⁺ pre-pDC”.

5) Figure 3B, C (line 181-183). What is the parent gate before CD11c vs Siglec H in Figure 3B? Show the full gating strategies in the supplemental figures and explain it in the legends.

In the figure legend, we stated: “Autofluorescent, Lineage⁺ and dead cells were excluded at the beginning of the strategy”. We apologize if this was not clear enough. We have now edited this text to “The first dot plot showing CD11c vs SiglecH expression was gated on singlets, live (LiveDead^{neg}), non-autofluorescent, Lineage(CD19, CD3, Ly6G, NK1.1)^{neg} cells”.

6) The definitions of pDC-like cells and tDCs is not done the same way as in the publications that first coined these names (Rodrigues et al. and Leylek et al.). This is very confusing and only clarified later (line 280-283). It would be really important to show the populations as previously defined and then show also the tdTomato signal. B220 is missing as an important marker and should be shown in Figure 3C.

We had to adapt our gating strategy to make sure that it would be suitable for the scRNA-seq analysis, meaning that it would unambiguously classify each cell in only one of the sorting gates, to avoid generating too many

ambiguous events or conflicting instructions on where a given cell belonged. Hence, we had to define exclusive gates for the pDC-like cells and for the tDC.

Rodrigues et al. defined pDC-like cells as $\text{Lin}^- \text{BST2}^+ \text{SiglecH}^+ \text{Zbtb46-GFP}^+$ cells (Rodrigues *et al.*, 2018). Our gating strategy is similar, except that we observed a clear distinction between $\text{Ly6D}^{+/high}$ and $\text{Ly6D}^{-/low}$ cells in this population and thus sorted these two cell populations to be able to compare them. We better explained that in our edited main text (lines 158-162).

Leylek et al. characterized tDCs as $\text{Lin}^- \text{CD11b}^- \text{XCR1}^- \text{SiglecH}^{low-to-high} \text{CD11c}^{low-to-high} \text{CX3CR1}^+$ cells (Leylek et al., 2019). They further showed that the tDC population they described was heterogeneous, encompassing a $\text{CD11c}^{low} \text{Ly6C}^{high}$ fraction more similar to pDCs and a $\text{CD11c}^{high} \text{Ly6C}^{low}$ fraction more similar to cDC2s. Importantly, the $\text{CD11c}^{low} \text{Ly6C}^{high}$ tDC expressed higher levels of SiglecH and of the Tcf4 transcription factor, whereas the $\text{CD11c}^{high} \text{Ly6C}^{low}$ tDC expressed high levels of the Zbtb46 transcription factor. Comparing the phenotypes and expression of key genes examined in both the $\text{CD11c}^{low} \text{Ly6C}^{high}$ tDC fraction (Leylek *et al.*, 2019) and the pDC-like cells (Rodrigues *et al.*, 2018) strongly suggested that corresponded to the same population. Thus, to avoid overlap between the definitions of the gates for pDC-like cells and for tDC, within singlet, live, non-autofluorescent, Lineage⁻ cells expressing CD11c or SiglecH⁻ and Tomato or Zbtb46-GFP, we focused on the CD11c^{high} tDC fraction, based on a phenotypic definition similar to that used by Leylek et al: $\text{XCR1}^- \text{CD11b}^- \text{SiglecH}^{-/low} \text{BST2}^{-/low} \text{CX3CR1}^+$. We have edited the main text of the manuscript to clarify this point (lines 162-172).

We have now generated novel data with spectral flow cytometry to be able to use each marker in an individual fluorescence channel, and to use more markers to identify the different DC types. This enabled us to gate Ly6C^+ tDCs versus CD11c^{high} tDCs in a similar manner as done by Leylek et al. This analysis confirmed that tdTomato is exclusively expressed by bona fide pDC in our novel mutant mouse model, and not by any of the tDC subset (Extended Data Fig. 4 and 5). Moreover, this analysis further supported the conclusion that the populations defined as pDC-like cells (Rodrigues *et al.*, 2018) or as $\text{CD11c}^{low} \text{Ly6C}^+$ tDC (Leylek *et al.*, 2019) largely overlap (Extended Data Fig. 5e).

7) By limiting the tDC gate to $\text{SiglecH}^- \text{Ly6D}^-$ cells only a subset of tDCs that has a phenotype closer to cDC2 is represented here (and also in the scRNA-seq data set). This should be made clear from the start by calling these cells CD11c^{high} (SiglecH^-) tDCs.

As explained above, for compatibility with the scRNA-seq experiment, we had to design a sorting strategy unambiguously classifying each cell in only one of the sorting gates, which indeed led us to limit the tDC gate to $\text{SiglecH}^- \text{Ly6D}^-$ cells to avoid phenotypic overlap with the gate for pDC-like cells that also correspond to the $\text{SiglecH}^+ (\text{Ly6C}^+)$ tDC fraction. We have edited the main text of the paper to explain this point more clearly (lines 158-172). We also followed the great advice of this reviewer to call from the start these cells CD11c^{high} tDCs, since we did show that they express higher levels of CD11c than pDC and pDC-like cells (Fig. 3c).

8) pDC-like cells [were not initially defined as Ly6D^-] and would therefore include the $\text{Zbtb46}^+ \text{Ly6D}^+$ cells. Rodrigues et al. defined pDC-like cells as $\text{Lin}^- \text{BST2}^+ \text{SiglecH}^+ \text{Zbtb46-GFP}^+$. Our gating strategy is similar, except that we observed two $\text{Ly6D}^{+/high}$ versus $\text{Ly6D}^{-/low}$ subpopulations, which we sorted separately to be able to compare them. The analysis of our scRNA-seq data showed that most of these Ly6D^+ cells clustered together with their Ly6D^- counterparts and that they shared enrichment for the pDC-like sgCMAP signature, consistent both populations being indeed pDC-like cells (main text line 270).

9) CX3CR1^- cells within the $\text{BST2}^{-/lo} \text{SiglecH}^-$ parent gate are ignored in the subsequent phenotype analysis (gaps in the tSNE plot?) and in the scRNA-seq analysis – this is a sizable population – which cells are these? Why were they not looked at? How is their expression of Zbtb46 and other markers? Ly6D could be expressed weakly on these cells – this depends on the sensitivity of the Ly6D staining.

These cells were not looked at because they did not fall in any of the phenotypic definitions of the DC types on which our study was focused (DC, pDC-like cells, CD11c^{high} tDC, cDC1 and cDC2). We used the FB5P-seq method for our scRNA-seq analysis because it enables recording the index sorting of each cell to link phenotype to transcriptome, which cannot be achieved with the 10X Genomics method for fluorescent reporter expression (only with cell surface markers with the CITE-seq method). Even though FB5P-seq is less labor

intensive than SmartSeq2, it remains difficult to treat more than a few hundred cells per experiment with this method that still requires individual cell sorting in 96-well plates with reverse transcription and bar-coding in each well. Hence, we were limited in the number of cell populations that we were able to sort and characterize.

We have now addressed this issue in our novel experiments harnessing spectral flow cytometry. We show that the vast majority of the cells expressing GFP in the spleen of mice bearing the *Zbtb46^{GFP}* reporter allele are cDC1 and cDC2. Other cells constitute only very minor fractions (Extended Data Fig. 4 and 5). Lin⁻, non-cDC1, non-cDC2, cells expressing CD11c or BST2 but not CX3CR1 were further characterized in these experiments (Extended Data Fig. 5g). A high proportion of these cells express CD11c and CD26, but not CD64, suggesting that these cells belong to the DC lineage. They encompass an MHC-II^{neg/low} fraction and an MHC-II^{high} fraction, suggesting a mixture of pre-DC and differentiated DC (main text lines 225-228).

10) In the tSNE plot in Fig. 3D it would be important to also make visible the cells that are not defined by the gating strategy shown in Fig. 3B and in addition to the backgating also show marker expression as overlays on the tSNE or UMAP plot for all cells that were included in the dimensionality reduction.

We have revised Fig. 3d accordingly.

11) I think it should be mentioned that the gated subsets here form a continuum of cells. This is also largely confirmed by the scRNA-seq data.

We agree. We edited the main text to mention this (lines 187-190): “The other gated DC types formed a continuum of cells between pDC and cDC2, with the small populations of tdTneg pDC and *Zbtb46⁺* Ly6D⁺ cells close to the pDC cluster, whereas the pDC-like cells and CD11c^{high} tDC settled closer to cDC2, consistent with previous similar observations¹¹”.

12) Line 218-219: the heterogeneity of the tDCs was already described by Leylek et al. PMID 31825848, so ZEST mice do not really contribute to this knowledge.

For better clarity, to explain our own gating strategy, we have now more precisely stated how tDC and their heterogeneity have been already described by Leylek et al (lines 162-165).

13) Lines 362-366, line 374-375 and line 505: This is not shown in the data that is shown in Figure 6F. The IFN- β -YFP reporter only detects cells that currently produce IFN- β and not cells that have already produced IFN- β , because it is not a fate-map mouse model. So you cannot conclude that ex-IFN-producers have migrated from marginal zone to T cell zone.

The *Ifnb1^{EYFP}* mouse model does not only detect the pDC that currently produce IFN- β but also the pDC that have already produced and secreted it. Hence, contrary to what the reviewer thinks, this reporter mouse model is reliable and well suited to fate-map the pDC that have produced IFN-I. We have demonstrated that in our previously published paper (Abbas *et al.*, 2020), by using a series of complementary, orthogonal, approaches, including flow cytometry analysis of IFN-I and EYFP expression in pDC isolated from the *Ifnb1^{EYFP}* reporter mice, and scRNAseq, which converged on the same conclusion that the pDC that have produced IFN-I survive after cessation of this activity and retain EYFP expression which allows to follow their fate. This is further explained below, with (Fig. R1 for reviewers) extracted from (Abbas *et al.*, 2020) at the end of this document.

In our previously published paper (Abbas *et al.*, 2020), we examined co-expression of EYFP and IFN-I in individual pDC, using intracellular staining with antibodies specific for these cytokines. This allowed us to unequivocally demonstrate the existence of three activation states in the process of IFN-I production by pDC: cells where only IFN-I are detectable (IFN-I⁺ EYFP⁻), cells positive for both signals (IFN-I⁺ EYFP⁺) and cells positive for YFP only (IFN-I⁻ EYFP⁺). IFN-I⁺ EYFP⁻ cells are at an “early state” of IFN-I expression, when they do not yet express sufficient levels of the fluorescent reporter protein to allow its detection. We showed via scRNA-seq that these cells start to express the *Eyfp* gene and all of the IFN-I genes, with a very high correlation between their expression. The IFN-I⁺ YFP⁺ cells are at the “peak state” of IFN-I production, expressing very high levels of both the EYFP and IFN-I proteins, and the *Eyfp* and IFN-I genes. The IFN-I EYFP^{+ / high} cells are at a “late state” where they have secreted all of their IFN-I, and stopped producing it (being negative for the *Eyfp* and IFN-I genes, and for the IFN-I proteins), but are fate mapped for this past function thanks to their maintenance of EYFP protein expression. Indeed, contrary to the IFN-I proteins, the EYFP protein is not secreted. Moreover, the EYFP protein has a long half-life, of >24 hours after termination of transcription of its gene (Hentschel *et al.*, 2013; Tombolini *et al.*, 1997), which is in part due to its high

resistance to proteases (Chiang et al., 2001). Hence, EYFP expression persists at high levels for >12 hours in the cells after they have ceased producing IFN-I and have secreted it. To formally prove this point, we sorted

YFP⁻ and YFP⁺ pDC from 36h infected CD45.2 mice and cultured them for 8 hours in vitro in recipient CD45.1 bone marrow FLT3-L cultures to assess the stability of YFP expression in ex vivo isolated pDC in our experimental settings. These data showed that none of the cells that were sorted as YFP⁺ reverted to being YFP⁻ (Abbas *et al.*, 2020).

In our previous study, we also examined kinetically the proportions of the three states described above (“early”, “peak” and “late” states) during the course of MCMV infection. The first two activation states were transient and tightly correlated in time, whereas the third state appeared later and accumulated over time, consistent with their proposed temporal succession.

Finally, in our previous study, we computationally inferred the pDC activation trajectory upon bio-informatics analysis of our scRNAseq data, by using two very different methods: Monocle and RNA velocity. Most importantly, RNA velocity does not take into account expression of the *Eyfp* and IFN-I genes, since they are intron-less, whereas RNA velocity exclusively uses information from intron-bearing genes to compute the next activation state of each single cell based on the integration of their ratio of unspliced to spliced RNA for hundreds of genes (La Manno *et al.*, 2018). Hence, the RNA Velocity method was based on using completely different information than the other methods. Yet, all methods converged onto the same activation trajectory including a temporal succession of the three activation states mentioned above: IFN-I⁺ EYFP⁻ → IFN-I⁺ EYFP⁺ → IFN-I⁻ EYFP⁺ (Abbas *et al.*, 2020).

We have made this point clearer in our revised manuscript (lines 335-340).

14) It should be avoided to overstate the results shown here. The data only show that the percentage of pDCs producing IFN is higher in the MZ and lower in the TCZ at the 36 hours time point and then it is lower in the MZ and higher in the TCZ 12 hours later. It is not clear that the IFN-producing cells have migrated to the TCZ or that only IFN-producing cells are licensed to migrate to the TCZ.

As explained above, we have previously demonstrated that the *Ifnb1*^{EYFP} reporter mouse model is reliable and well suited to fate-map the pDCs that have produced IFN-I (Abbas *et al.*, 2020). In the present study, we can clearly distinguish the pDC that are making, or have made, IFN-I, since they are double positive for EYFP and tdTomato, from the pDC not engaged into IFN-I production since they only express tdTomato. In several previous studies including (Abbas *et al.* 2020), we showed that pDC produce and secrete IFN-I between 33h and 40h after MCMV infection, and that this activity has ceased by 48h. Hence, the EYFP⁺ observed at 48h on spleen sections are pDC that are not producing IFN-I anymore but have produced it before at ~36h. Our data show that the proportion of EYFP⁺ cells in pDC decreases in the MZ and increases in the T cell zone between 36h and 48h, whereas the proportion of EYFP⁻ cells in pDC increases in the marginal zone and decreases in the T cell zone. Moreover, our data show that the density of EYFP⁺ cells decreases in marginal zone pDC clusters at 48h after infection, whereas their density in the T cell zone increases. Altogether, these data support the conclusions that the IFN-producing pDC have migrated to the TCZ between 36h and 48h after MCMV infection, and that only IFN-producing pDC are licensed for this migration to the TCZ.

13) Line 434-435: “only few genes were found to be differentially expressed between tDC and cDC2” – this is not surprising as only [CD11c^{hi} tDCs were] studied here, which [are] very close to cDC2.

To the best of our knowledge, the overall gene expression profiling of CD11c^{high} tDC had not been described before. The proposed proximity to cDC2 was based on the analysis of a few cell surface markers, transcription factors and genes. Moreover, it has been studied only at steady state and not in detail after activation. Hence, our study does bring novel information, including further convergence between CD11c^{high} tDC and cDC2 during their response to a viral infection in vivo, which could be seen as surprising since experts in the field were expecting tDC and cDC2 to exert different functions.

Moreover, we did study pDC-like cells which correspond to CD11c^{low} Ly6C⁺ tDC as discussed above. We showed that steady state pDC-like cells and CD11c^{high} tDC were positioned in close vicinity on the scRNA-seq UMAP, whereas their gene expression profiles diverged upon activation, with the pDC-like cells

converging towards activated pDC whereas the CD11c^{high} tDC converged towards activated cDC2. These results are also novel, and were not predictable considering the phenotypic and transcriptomic proximity between pDC-like cells and CD11c^{high} tDC at steady state.

14) Line 452-453 and 456: the genes mentioned here should be validated by qPCR and/or protein detection and checked how specific they really are to know how useful they really are for potential tDC targeting. Validation by qRT-PCR has now been achieved (Extended Data Fig. 10d).

Minor points.

a) Fig. 2a: very confusing and hard to see— use more space to show this.

We thank the reviewer for pointing this issue which we have now corrected by editing Fig. 2a.

b) Extended Fig. 2a: CDP should be gated Ly6D-negative to exclude any pDC progenitors (according to Dress et al. 31213723).

Negativity for Ly6D is not yet a consensus for CDP identification. We cannot remove the Ly6D⁺ cells from our CDP gate since two different panels were generated for the myeloid versus lymphoid precursor analyses, and Ly6D was not included in the “myeloid precursor” panel. Even as we defined it without excluding potential Ly6D⁺ pDC progenitor contaminants, CDPs are negative for tdTomato expression. Thus, our data support the conclusion that tdTomato is not expressed in CDPs but only late in the CD11c⁺ pre-pDC.

c) Extended Figure 2B: CD127 and SiglecH signals seem to be rather weak – this could be improved.

Although we agree that this could be improved, our data are robust enough to support our conclusions. Hence, we did not redo these experiments since this is not necessary for our conclusions and we had to prioritize the use of our currently limiting resource in money and personnel.

d) Line 262 and 264: discrepancy between CCR9 protein expression and *Ccr9* mRNA expression in pDC-like cells – how can this be explained? Are these cells downregulating CCR9 and protein is still there when mRNA has already been downregulated?

There can be discrepancies between mRNA and protein expression, including for key cell surface markers of DC types (see XCR1 case, answer to reviewer#1, point 6). This could be due to low copy number gene expression with high drop-out frequency, or to rapid downregulation of gene transcription upon activation whereas the half-life of the protein allows maintenance of its expression as happens in the IFN-I-producing pDC where CCR9 protein expression is maintained in the face of strong *Ccr9* gene downregulation (see Figure just below). Similar phenomena probably occur in pDC-like cells where CCR9 is better detected than *Ccr9*.

e) Line 403: mention IFNL2 as type III IFN is also mentioned in the title of this chapter.
Thank you for pointing out this omission. We have corrected it (line 414).

Reviewer #3:

We thank Reviewer #3 for pointing out the “elegance”, “specificity” and “strength” of our novel mutant mice, the “robustness” of the data we provide for its validation which “convincingly” demonstrates that “it is a bona fide pDC reporter mouse, far surpassing current models available”. We appreciate the statement that “the differential localization of pDC producing IFN or not during MCMV infection is novel data and of interest”.

Suggestions for improvement.

A) Although this group and others cite peak MCMV-induced Type I IFN responses in serum at approx 36hrs after infection, the data of Fig 7 show beautifully that pDC activation/relocalisation to the MZ has occurred (maximally) by 12 hrs after infection. Using this mouse model to examine very early kinetics of infection, from 0-12 hrs, would elevate the novelty of this work and provide a deeper understanding of the role of pDC in these early stages, something that has been missing until now. RNAseq of pDC from these early stages to ascertain whether they are already capable of IFN production would be of interest to the field, as would the phenotype of the pDC like cells, similar to Fig 8.

We have shown in previous papers that pDC IFN-I production cannot be detected by intracellular staining between 0h and 30h post-infection (Abbas *et al.*, 2020; Zucchini *et al.*, 2008). In our hands, this is also the case when looking at pDC EYFP expression in *Ifnb1^{EYFP}* mice (unpublished data), which becomes barely detectable at 33 hours after infection (Abbas *et al.*, 2020). Hence, we do not expect to be able to detect pDC expressing IFN-I genes between 0-12 hrs after MCMV infection, or their frequency would be so low (<0.1% of pDC) that having enough of these cells to robustly characterize them upon performing scRNA-seq of total pDC would require profiling over 100,000 pDC, which is not technically and financially feasible. It may still be that the pDC clustering at the marginal zone at 12 hours after MCMV infection start to diversify in terms of transcriptomic or epigenetic states, which would indeed be interested to investigate. However, performing such studies are beyond the scope of a Technical Report aiming at introducing the new mouse model to the immunology community, the strategy suggested by the editors and that we are happy to follow.

To illustrate original in vivo observations enabled for the first time to the best of our knowledge thanks to our novel reporter mouse models, we have added novel confocal microscopy data as 3D reconstructions showing, in the marginal zone of the spleen of MCMV-infected mice, the tight interactions that exist between pDC producing IFN-I and virus-infected cells, as well as between pDC producing or not IFN-I (movie S1-S3).

B) Moreover, although the spleen has provided useful data in the development of the MCMV infection model, from the perspective of the role and function of pDC during infection it is important that the total numbers are tracked within the spleen (not just percentages as shown in Fig 3 and extended data).

We agree. We have generated novel spectral flow cytometry experiments to reinforce the phenotypic characterization of the cells expressing tdTomato or GFP in ZeST mice. We have harnessed these experiments to measure the absolute numbers of the different DC types in the spleen at steady state and at 48h after MCMV infection, and in larger cohorts of mice than achieved previously (new Supplementary Table 1).

C) In addition, this model would allow the tracking of the efflux of pDC from the bone marrow, a missing 'link' from this work. Is pDC egress from bone marrow occurring at early time points? How does the reporter work on pDC in the bone marrow compartment, one of the most abundant sources of pDC and their immediate immature precursors?

In Fig. 2, we did show tdTomato expression in the pDC lineage in the BM, not only in CD11c⁺ pre-pDC (Fig. 2c) but also in terminally differentiated pDC (Fig. 2b). This shows that tdTomato is clearly and rather uniformly expressed in terminally differentiated BM pDC, and starts to be expressed at lower but detectable levels in their immediate precursors, the CD11c⁺ pre-pDC, but not in more upstream precursors.

We agree that it would be scientifically very interesting to address whether and how MCMV infection modulate pDC efflux from the BM, and whether the pDC that start clustering at the marginal zone around 12h after infection represent recent BM emigrants that circulated in the blood and just entered the spleen via the venous sinus of the marginal zone. However, performing such studies are beyond the scope of a Technical Report focused on introducing the new mouse model to the immunology community.

Specific comments of data.

1) In Ext Fig 3 it is unclear how the percentages in Supp Table I relate back to the plots of extended Fig 3, particularly the far-right dot plots.

In the previous version of the Supplementary Table 1, the percent for each population were given within the first gate shown in (Fig. 3b), namely within singlets, live (LiveDead⁻), non-autofluorescent, Lineage(CD19, CD3, Ly6G, NK1.1)⁻ cells expressing CD11c or SiglecH. In Extended Data Fig. 3, the far-right plots give the percent of each population (pDC, Zbtb46⁺ Ly6D⁺ cells, and pDC-like cells) within the upstream gate of (GFP⁺ or tdTomato⁺) non-cDC2, non-cDC1, BST2^{high} SiglecH⁺ cells (see gating strategy on Fig. 3b). Hence, the data were not calculated in the same gate for Supplementary Table 1 versus Extended data Fig. 3.

In any case, as explained in point (B) above, we have now generated new data by using spectral flow cytometry to identify DC types (Extended Data Fig. 5) and calculate their absolute numbers in the spleen at steady state and at 48h after MCMV infection. These new data were used to calculate absolute numbers of cells, for more mice per group, and hence to revise Supplementary Table 1 according to this reviewer's request.

2) Although line 197 states 'that proportions of the distinct DC populations were not 'significantly modified during MCMV infection', in the data of Supp Table I the percentages of pDC in the 48h infected spleen range from 6-32%; 4-12% at 36h. As stated above, cell numbers are required here- is there a massive loss in some animals of cDC- numbers here for all cell types would help to clarify. Data from additional animals may be required if large variations between animals are observed.

We agree. As stated just above, we have performed additional experiments to answer this request and revise Supplementary Table 1 accordingly.

3) Given the variation in cell populations noted in 2- it would seem inadequate to analyse cells from a single infected mouse (from each of 36 & 48h PI) in the data of Ext Fig 5 and Fig 8.

We agree. Actually, the statements that one single mouse was used for each time point for the scRNA-seq experiment is wrong, due to mis-communication on this specific point between the first and last authors. We deeply apologize for this mistake. Indeed, 2 to 3 individual mice were used as a source for the single cell for each time point, with 3 independent sorts performed with 2 or 3 animals each time (sorts for mice #56, 58 on 03/11/2020, for #52, 53 & 61 on 17/12/2020, for #81, 84, 86 on 11/02/2021), sorting plates frozen until all samples had been collected, and all libraries generated and sequenced simultaneously to avoid eventual batch effects. Additional figures have been added to the revised manuscript to allow tracing which cells come from which donor mice (Extended Data Fig. 7a), and to quantitate the contribution of each mouse to each combination of DC type and activation state (Supplementary Fig. 1). The cell types and states identified in our scRNA-seq experiment are robust, since they are shared across individual mice, with a similar distribution of cell types and states between individual mice studied at the same time point after infection.

References.

- Abbas, A., Vu Manh, T.P., Valente, M., Collinet, N., Attaf, N., Dong, C., Naciri, K., Chelbi, R., Brelurut, G., Cervera-Marzal, I., et al. (2020). The activation trajectory of plasmacytoid dendritic cells in vivo during a viral infection. *Nat Immunol* 21, 983-997. 10.1038/s41590-020-0731-4.
- Ardouin, L., Luche, H., Chelbi, R., Carpentier, S., Shawket, A., Montanana Sanchis, F., Santa Maria, C., Grenot, P., Alexandre, Y., Gregoire, C., et al. (2016). Broad and Largely Concordant Molecular Changes Characterize Tolerogenic and Immunogenic Dendritic Cell Maturation in Thymus and Periphery. *Immunity* 45, 305-318. 10.1016/j.immuni.2016.07.019.
- Baban, B., Chandler, P.R., Sharma, M.D., Pihkala, J., Koni, P.A., Munn, D.H., and Mellor, A.L. (2009). IDO activates regulatory T cells and blocks their conversion into Th17-like T cells. *J Immunol* 183, 2475-2483. 10.4049/jimmunol.0900986.
- Baban, B., Hansen, A.M., Chandler, P.R., Manlapat, A., Bingaman, A., Kahler, D.J., Munn, D.H., and Mellor, A.L. (2005). A minor population of splenic dendritic cells expressing CD19 mediates IDO-dependent T cell suppression via type I IFN signaling following B7 ligation. *Int Immunol* 17, 909-919. 10.1093/intimm/dxh271.
- Chauveau, A., Pirogova, G., Cheng, H.W., De Martin, A., Zhou, F.Y., Wideman, S., Rittscher, J., Ludewig, B., and Arnon, T.I. (2020). Visualization of T Cell Migration in the Spleen Reveals a Network of Perivascular Pathways that Guide Entry into T Zones. *Immunity* 52, 794-807 e797. 10.1016/j.immuni.2020.03.010.
- Chiang, C.F., Okou, D.T., Griffin, T.B., Verret, C.R., and Williams, M.N. (2001). Green fluorescent protein rendered susceptible to proteolysis: positions for protease-sensitive insertions. *Arch Biochem Biophys* 394, 229-235. 10.1006/abbi.2001.2537.
- Crozat, K., Tamoutounour, S., Vu Manh, T.P., Fossum, E., Luche, H., Ardouin, L., Guillems, M., Azukizawa, H., Bogen, B., Malissen, B., et al. (2011). Cutting edge: expression of XCR1 defines mouse lymphoid-tissue resident and migratory dendritic cells of the CD8alpha+ type. *J Immunol* 187, 4411-4415. 10.4049/jimmunol.1101717.
- Dalod, M., and Scheu, S. (2022). Dendritic cell functions in vivo: A user's guide to current and next-generation mutant mouse models. *Eur J Immunol*. 10.1002/eji.202149513.
- Dress, R.J., Dutertre, C.A., Giladi, A., Schlitzer, A., Low, I., Shadan, N.B., Tay, A., Lum, J., Kairi, M., Hwang, Y.Y., et al. (2019). Plasmacytoid dendritic cells develop from Ly6D(+) lymphoid progenitors distinct from the myeloid lineage. *Nat Immunol* 20, 852-864. 10.1038/s41590-019-0420-3.
- Ginhoux, F., Guillems, M., and Merad, M. (2022). Expanding dendritic cell nomenclature in the single-cell era. *Nat Rev Immunol* 22, 67-68. 10.1038/s41577-022-00675-7.
- Hentschel, E., Will, C., Mustafi, N., Burkovski, A., Rehm, N., and Frunzke, J. (2013). Destabilized eYFP variants for dynamic gene expression studies in *Corynebacterium glutamicum*. *Microb Biotechnol* 6, 196-201. 10.1111/j.1751-7915.2012.00360.x.
- Johnson, B.A., 3rd, Kahler, D.J., Baban, B., Chandler, P.R., Kang, B., Shimoda, M., Koni, P.A., Pihkala, J., Vilagos, B., Busslinger, M., et al. (2010). B-lymphoid cells with attributes of dendritic cells regulate T cells via indoleamine 2,3-dioxygenase. *Proc Natl Acad Sci U S A* 107, 10644-10648. 10.1073/pnas.0914347107.
- La Manno, G., Soldatov, R., Zeisel, A., Braun, E., Hochgerner, H., Petukhov, V., Lidschreiber, K., Kastrioti, M.E., Lonnerberg, P., Furlan, A., et al. (2018). RNA velocity of single cells. *Nature* 560, 494-498. 10.1038/s41586-018-0414-6.
- Leylek, R., Alcantara-Hernandez, M., Lanzar, Z., Ludtke, A., Perez, O.A., Reizis, B., and Idoyaga, J. (2019). Integrated Cross-Species Analysis Identifies a Conserved Transitional Dendritic Cell Population. *Cell Rep* 29, 3736-3750 e3738. 10.1016/j.celrep.2019.11.042.
- Maier, B., Leader, A.M., Chen, S.T., Tung, N., Chang, C., LeBerichel, J., Chudnovskiy, A., Maskey, S., Walker, L., Finnigan, J.P., et al. (2020). A conserved dendritic-cell regulatory program limits antitumour immunity. *Nature* 580, 257-262. 10.1038/s41586-020-2134-y.
- Manh, T.P., Alexandre, Y., Baranek, T., Crozat, K., and Dalod, M. (2013). Plasmacytoid, conventional, and monocyte-derived dendritic cells undergo a profound and convergent genetic reprogramming during their maturation. *Eur J Immunol* 43, 1706-1715. 10.1002/eji.201243106.
- Manlapat, A.K., Kahler, D.J., Chandler, P.R., Munn, D.H., and Mellor, A.L. (2007). Cell-autonomous control of interferon type I expression by indoleamine 2,3-dioxygenase in regulatory CD19+ dendritic cells. *Eur J Immunol* 37, 1064-1071. 10.1002/eji.200636690.
- Mellor, A.L., Baban, B., Chandler, P.R., Manlapat, A., Kahler, D.J., and Munn, D.H. (2005). Cutting edge: CpG oligonucleotides induce splenic CD19+ dendritic cells to acquire potent indoleamine 2,3-dioxygenase-dependent T cell regulatory functions via IFN Type 1 signaling. *J Immunol* 175, 5601-5605. 10.4049/jimmunol.175.9.5601.
- Munn, D.H., Sharma, M.D., Hou, D., Baban, B., Lee, J.R., Antonia, S.J., Messina, J.L., Chandler, P., Koni, P.A., and Mellor, A.L. (2004). Expression of indoleamine 2,3-dioxygenase by plasmacytoid dendritic cells in tumor-draining lymph nodes. *J Clin Invest* 114, 280-290. 10.1172/JCI21583.
- Rodrigues, P.F., Alberti-Servera, L., Eremin, A., Grajales-Reyes, G.E., Ivanek, R., and Tussiwand, R. (2018). Distinct progenitor lineages contribute to the heterogeneity of plasmacytoid dendritic cells. *Nat Immunol* 19, 711-722. 10.1038/s41590-018-0136-9.
- Schlitzer, A., Heiseke, A.F., Einwachter, H., Reindl, W., Schiemann, M., Manta, C.P., See, P., Niess, J.H., Suter, T., Ginhoux, F., and Krug, A.B. (2012). Tissue-specific differentiation of a circulating CCR9- pDC-like common dendritic cell precursor. *Blood* 119, 6063-6071. 10.1182/blood-2012-03-418400.
- Schlitzer, A., Loschko, J., Mair, K., Vogelmann, R., Henkel, L., Einwachter, H., Schiemann, M., Niess, J.H., Reindl, W., and Krug, A. (2011). Identification of CCR9- murine plasmacytoid DC precursors with plasticity to differentiate into conventional DCs. *Blood* 117, 6562-6570. 10.1182/blood-2010-12-326678.
- Shimizu, K., Morikawa, S., Kitahara, S., and Ezaki, T. (2009). Local lymphogenic migration pathway in normal mouse spleen. *Cell Tissue Res* 338, 423-432. 10.1007/s00441-009-0888-5.
- Tombolini, R., Unge, A., Davey, M.E., de Bruijn, F.J., and Jansson, J.K. (1997). Flow cytometric and microscopic analysis of GFP-tagged *Pseudomonas fluorescens* bacteria. *FEMS Microbiology Ecology* 22, 17-28. 10.1111/j.1574-6941.1997.tb00352.x.
- Zucchini, N., Bessou, G., Robbins, S.H., Chasson, L., Raper, A., Crocker, P.R., and Dalod, M. (2008). Individual plasmacytoid dendritic cells are major contributors to the production of multiple innate cytokines in an organ-specific manner during viral infection. *Int Immunol* 20, 45-56. 10.1093/intimm/dxm119.

Fig. R1.

Fig. R1, excerpts from (Abbas et al., 2020). The *Ifnb1*^{EYFP} mouse model allows fate-mapping the pDCs that have produced IFN-I. **a**, Flow cytometry analysis of the expression of IFN- α/β versus YFP in pDC isolated from 36h MCMV-infected *Ifnb1*^{EYFP} reporter mice. Numbers inside quadrants indicate the frequency of each pDC subpopulation. **b**, Flow cytometry kinetic analysis of the expression of IFN- α/β versus YFP in pDC isolated from *Ifnb1*^{EYFP} mice, at 0 (UN), 33, 36, 40, 44 and 48 hours after MCMV infection. **c**, Proportions of IFN- α/β ⁺YFP^{neg}, IFN- α/β ⁺YFP⁺ and of IFN- α/β ^{neg}YFP⁺ cells amongst pDC, at indicated time points. **d**, Pie charts recapitulating the proportions of cells expressing IFN-I and/or YFP (see color key) amongst pDC positive for either molecule at different time points during the course of MCMV infection in *Ifnb1*^{EYFP} mice. **e**, Dimensional reduction performed using UMAP algorithm and clustering of 264 bona fide pDC isolated from control or 36h MCMV-infected mice. **f**, Inverse hyperbolic arcsine (asinh) fluorescence intensity of YFP projected on the UMAP space. **g-h**, Expression of *Yfp* and *Ifnb1* on the UMAP space. **i** Violin plots showing mRNA expression profiles of *Eyfp*, *Ifnb1* and *Ifna4* across all individual cells and in comparison between clusters identified in (e), cluster 0= UN pDC, clusters 1-4= MCMV *Eyfp*^{neg}YFP^{neg} pDC, cluster 5= MCMV *Eyfp*⁺YFP^{neg} pDC, cluster 6= MCMV *Eyfp*⁺YFP⁺ pDC, cluster 7= MCMV *Eyfp*^{neg}YFP⁺ pDC. **j**, Violin plots showing inverse hyperbolic arcsine (asinh) fluorescence intensity of EYFP across Seurat clusters. **k**, Violin plots showing mRNA expression profiles of *Ii12b* and *Ccr7* across Seurat clusters. **l**, Pseudo-temporal inference of the pDC activation trajectory as assessed by Monocle 3 from the scRNA-seq pDC dataset. Pseudo-time was calculated for each cell, upon setting of the cells from the uninfected animal as the root of the activation trajectory consistent with their location at one end of it. **m**, Comparison of the expression of the EYFP protein vs the IFN-I meta-gene along pseudo-time. Individual cells are shown as dots and a 6-order polynomial curve was fit to the data to illustrate expression pattern along pseudo-time. **n**, Comparison of the expression along pseudo-time of the indicated genes vs the IFN-I meta-gene, each normalized to their maximal value. **o**, Reconstruction of the activation trajectory of pDCs by using RNA Velocity. The Velocityto software was used to compute the most probable next activation state of each pDC based on its relative expression of unspliced versus spliced mRNA for many genes in comparison with all other pDCs. Projections of the individual velocity vector of each pDC are represented as arrows on the UMAP space. **p**, Predicted induction (red) vs termination (blue) of the transcription of selected genes. This prediction is based for each cell and indicated gene on the difference between the observed and the expected abundance of unspliced mRNA, called the *u* residuals.

The IFN-I^{neg} EYFP^{neg} → IFN-I⁺ EYFP^{neg} → IFN-I⁺ EYFP⁺ → IFN-I^{neg} EYFP⁺ activation trajectory of the pDCs discovered in (Abbas et al., 2020) is supported by 3 approaches.

- 1) Intracellular staining for IFN-I in the pDC from *Ifnb1*^{EYFP} mice. It demonstrates the existence of 3 activation states of the pDCs producing IFN-I (Fig. R1a). IFN-I⁺YFP^{neg} cells are at an “early state” of IFN-I expression, where they express all the IFN-I

genes and the *Eyfp* gene (Fig. R1e, g-i), as well as detectable levels of the IFN-I proteins (Fig. R1a), but not yet sufficient levels of the EYFP protein to allow its detection by flow cytometry (Fig. R1f, j). The IFN-I⁺YFP⁺ cells are at the “peak state” of IFN-I production where they express high levels of the IFN-I and EYFP genes and proteins. The IFN-I^{neg} YFP^{+/high} cells are at a “late state” where they have secreted all of their IFN- I and stopped producing it (being negative for the IFN-I and *Eyfp* genes), but are fate mapped for this past function due to their maintenance of EYFP protein expression.

- 2) Kinetic analysis of the proportions of the 3 states described above (Fig. R1b-d). It shows that the first 2 activation states are transient and tightly correlated in time, whereas the third state appears later and accumulates over time, consistent with their proposed succession.
- 3) Bio-informatics analysis of scRNAseq data, with two different methods, Monocle (Fig. R1l-n) and RNA velocity (Fig. R1o,p). RNA velocity is completely independent from the other approaches. Indeed, it computes the next activation state of each single cell based on the integration of its precise transcriptional state for hundreds of intron-bearing genes, as assessed by their ratio of unspliced to spliced RNA. Being intron-less, the *Eyfp* reporter and IFN-I genes were not considered in this analysis. Neither EYFP protein expression. Hence, the RNA Velocity method was based on using completely different information than the other methods. Yet, it converged onto the same pDC activation trajectory: IFN-I^{neg} EYFP^{neg} → IFN-I⁺ EYFP^{neg} → IFN-I⁺ EYFP⁺ → IFN-I^{neg} EYFP⁺.

Decision Letter, first revision:

1st Dec 2022

Dear Marc,

Thank you for your patience. We have finally received all of the referee reports for your revised manuscript entitled "Novel mouse models based on intersectional genetics to identify and characterize plasmacytoid dendritic cells" (NI-RS33584A). It has now been seen by the original referees and both reviewers find that the paper has improved in revision, and are recommending publication. Therefore we'll be happy in principle to publish it in Nature Immunology, pending minor revisions to satisfy the referee #2 final requests (reporting requirements that we would have also requested) and to comply with our editorial and formatting guidelines.

We will now perform detailed checks on your paper and will send you a checklist detailing our editorial and formatting requirements in about a week. Please do not upload the final materials and make any revisions until you receive this additional information from us.

If you had not uploaded a Word file for the current version of the manuscript, we will need one before beginning the editing process; please email that to immunology@us.nature.com at your earliest convenience.

Thank you again for your interest in Nature Immunology Please do not hesitate to contact me if you have any questions.

Kind regards,

Laurie

Laurie A. Dempsey, Ph.D.
Senior Editor
Nature Immunology
l.dempsey@us.nature.com
ORCID: 0000-0002-3304-796X

Reviewer #1 (Remarks to the Author):

The authors have addressed all the requested points.

Reviewer #2 (Remarks to the Author):

The revised manuscript is improved in clarity and the reviewers' questions were addressed in sufficient detail in my opinion. The paper will be a valuable resource and technical report describing a new mouse model that will be useful for future functional studies of plasmacytoid DCs and related cell types.

Minor issues:

Please include a description of what can be seen in the videos that were added as supplemental data. It is not possible to understand what is shown here without an explanation.

The additional references 52-59 should be included in the main list of references, because they are mentioned in the main text of the paper.

For the sgCMAP algorithm that is used here and strongly recommended by the authors, only the See et al. paper is referenced, but I cannot find the algorithm anywhere. It would be important to provide the Github link (and the version) with a doi to actually be able to find the package and reproduce the data analysis. A marup fiel of how the analysis was done would also be great. It would be very good to upload not only the raw scRNAseq data but also the processed data in order to be able to use the data set as a resource.

Author Rebuttal, first revision:

Point-by-point response to comments remaining from reviewers.

Reviewer #1:

We thank Reviewer #1 for stating that we “have addressed all the requested points”.

Reviewer #2:

We thank Reviewer #2 for acknowledging that our revised manuscript “is improved in clarity and the reviewers’ questions were addressed in sufficient detail”. We are happy that this reviewer states that our “paper will be a valuable resource and technical report describing a new

mouse model that will be useful for future functional studies of plasmacytoid Dcs and related cell types”.

Minor issues.

A) Please include a description of what can be seen in the videos that were added as supplemental data. It is not possible to understand what is shown here without an explanation. We did include a caption online in the corresponding box during the submission. We are sorry that this information was apparently not accessible to the reviewers. It will be available to readers in the published version of the paper. It is pasted below.

Supplementary Video 1 to 3. Interactions between IFN-I-producing pDC and MCMV infected cells or pDC that do not produce IFN-I. Representative videos of IFN-I-producing pDC in direct contact with MCMV infected cells (IE1+) at 40h after MCMV infection in the marginal zone of the spleen of SCRIPT mice (pDC: magenta; Ifnb1-YFP reporter: yellow; IE1: cyan). 3D reconstructions were generated using the IMARIS software v9.9.1. n = 4 mice, thickness: 20-30 µm, distance between slices: 1 µm.

B) The additional references 52-59 should be included in the main list of references, because they are mentioned in the main text of the paper.

One reference has been removed upon shortening the main text of the manuscript as requested by the editors. References 51-59 are only cited in the methods” section of the manuscript, and are thus listed at the end of this section and not with the other references cited in the previous section of the manuscript, as per the style of nature Immunology.

C) For the sgCMAP algorithm that is used here and strongly recommended by the authors, only the See et al. paper is referenced, but I cannot find the algorithm anywhere. It would be important to provide the Github link (and the version) with a doi to actually be able to find the package and reproduce the data analysis.

We have now provided in the methods section a Github link for the algorithm (https://github.com/SlgN-Bioinformatics/sgCMAP_R_Scripts), as well as for an example and recommendations on how to use it (https://github.com/DalodLab/MDlab_cDC1_differentiation/blob/main/scRNAseq_pipeline.md#cell-annotation-using-cmap).

D) A mark-up file of how the [scRNA-seq] analysis was done would also be great.

We thank the reviewer for this suggestion. It has now been included as Supplementary Fig. 2.

E) It would be very good to upload not only the raw scRNAseq data but also the processed data in order to be able to use the data set as a resource.

This had already been done at the time we initially submitted to GEO. The data uploaded for our GEO series GSE196720 encompass (i) the raw data, (ii) the count matrix, and (iii) the log_n-normalized data.

Final Decision Letter:

Dear Marc,

I am delighted to accept your manuscript entitled "Novel mouse models based on intersectional genetics to identify and characterize plasmacytoid dendritic cells" for publication in an upcoming issue of Nature Immunology.

Over the next few weeks, your paper will be copyedited to ensure that it conforms to Nature Immunology style. Once your paper is typeset, you will receive an email with a link to choose the appropriate publishing options for your paper and our Author Services team will be in touch regarding any additional information that may be required.

Please note that *Nature Immunology* is a Transformative Journal (TJ). Authors may publish their research with us through the traditional subscription access route or make their paper immediately open access through payment of an article-processing charge (APC). Authors will not be required to make a final decision about access to their article until it has been accepted. [Find out more about Transformative Journals](https://www.springernature.com/gp/open-research/transformative-journals).

Your paper will be published online soon after we receive your corrections and will appear in print in the next available issue. Content is published online weekly on Mondays and Thursdays, and the

embargo is set at 16:00 London time (GMT)/11:00 am US Eastern time (EST) on the day of publication. Now is the time to inform your Public Relations or Press Office about your paper, as they might be interested in promoting its publication. This will allow them time to prepare an accurate and satisfactory press release. Include your manuscript tracking number (NI-TR33584B) and the name of the journal, which they will need when they contact our office.

About one week before your paper is published online, we shall be distributing a press release to news organizations worldwide, which may very well include details of your work. We are happy for your institution or funding agency to prepare its own press release, but it must mention the embargo date and Nature Immunology. Our Press Office will contact you closer to the time of publication, but if you or your Press Office have any enquiries in the meantime, please contact press@nature.com.

Also, if you have any spectacular or outstanding figures or graphics associated with your manuscript - though not necessarily included with your submission - we'd be delighted to consider them as candidates for our cover. Simply send an electronic version (accompanied by a hard copy) to us with a possible cover caption enclosed.

Please note that we encourage the authors to self-archive their manuscript (the accepted version before copy editing) in their institutional repository, and in their funders' archives, six months after publication. Nature Portfolio recognizes the efforts of funding bodies to increase access of the research they fund, and strongly encourages authors to participate in such efforts. For information about our editorial policy, including license agreement and author copyright, please visit www.nature.com/ni/about/ed_policies/index.html

An online order form for reprints of your paper is available at <https://www.nature.com/reprints/author-reprints.html>. Please let your coauthors and your institutions' public affairs office know that they are also welcome to order reprints by this

method.

Kind regards,

Laurie

Laurie A. Dempsey, Ph.D.
Senior Editor
Nature Immunology
l.dempsey@us.nature.com
ORCID: 0000-0002-3304-796X